

**Advancements in the Aerosol Robotic Network (AERONET)**
**Version 3 Database – Automated Near Real-Time Quality**
**Control Algorithm with Improved Cloud Screening for Sun**
**Photometer Aerosol Optical Depth (AOD) Measurements**
David M. Giles[1,2,] Alexander Sinyuk[1,2,] Mikhail S. Sorokin[1,2,] Joel S. Schafer[1,2,] Alexander
Smirnov[1,2,] Ilya Slutsker[1,2,] Thomas F. Eck[2,3,] Brent N. Holben[2,] Jasper Lewis[2,4,] James Campbell[5,]
Ellsworth J. Welton[2,] Sergey Korkin[2,3,] and Alexei Lyapustin[2]
[1]Science Systems and Applications Inc. (SSAI), Lanham, MD 20706, USA
[2]NASA Goddard Space Flight Center (GSFC), Greenbelt, MD 20771, USA
[3]Universities Space Research Association (USRA), Columbia, MD 21046, USA
[4]Joint Center for Earth Systems Technology, University of Maryland, Baltimore County, Baltimore, MD 21250,
USA
[5]Marine Meteorology Division, Naval Research Laboratory (NRL), Monterey, CA 93943, USA
**Correspondence:** David M. Giles (David.M.Giles@nasa.gov)





**Abstract.** The Aerosol Robotic Network (AERONET) provides highly accurate, ground-truth measurements of the aerosol optical depth (AOD) using Cimel Electronique Sun/Sky radiometers for more than 25 years. In Version 2 (V2) of the AERONET database, the near real-time AOD was semi-automatically quality controlled utilizing mainly cloud screening methodology, while additional AOD data contaminated by clouds or affected by instrument anomalies were removed manually before attaining quality assured status (Level 2.0). The large growth in the number of AERONET sites over the past 25 years resulted in significant burden to manually quality control millions of measurements in a consistent manner. The AERONET Version 3 (V3) algorithm provides fully automatic cloud screening and instrument anomaly quality controls. All of these new algorithm updates apply to near real-time data as well as post-field deployment processed data, and AERONET reprocessed the database in 2018. A full algorithm redevelopment provided the opportunity to improve data inputs and corrections such as unique filter specific temperature characterizations for all visible and near-infrared wavelengths, updated gaseous and water vapor absorption coefficients, and ancillary data sets. The Level 2.0 AOD quality assured data set is now available within a month after post-field calibration, reducing the lag time from up to several months. Near real-time estimated uncertainty is determined using data qualified as V3 Level 2.0 AOD and considering the difference between the AOD computed with the pre-field calibration and AOD computed with pre-field and post-field calibration. This assessment provides a near real-time uncertainty estimate where average differences of AOD suggest a +0.02 bias and one sigma uncertainty of 0.02, spectrally, but the bias and uncertainty can be significantly larger for specific instrument deployments. Long-term monthly averages analyzed for the entire V3 and V2 databases produced average differences (V3−V2) of +0.002 with a ±0.02 standard deviation, yet monthly averages calculated using time-matched observations in both databases were analyzed to compute an average difference of −0.002 with a ±0.004 standard deviation. The high statistical agreement in multi-year monthly averaged AOD validates the advanced automatic data quality control algorithms and suggests that migrating research to the V3 database will corroborate most V2 research conclusions and likely lead to more accurate results in some cases.



## 1 Introduction

Space-based, airborne, and surface-based Earth observing platforms can remotely retrieve or measure aerosol
abundance. Each method has its own assumptions and dependencies in which the aerosol total column abundance
quantified by aerosol optical depth (AOD) introduces uncertainty in the retrieval or measurement. At the forefront,
ground based Sun photometry has been considered the ground truth in the measurement of AOD given minimal
assumptions, reliable calibration, and weak dependency on trace gases at carefully selected wavelength bands thus
resulting in highly accurate data (Holben et al., 1998). Meanwhile, AOD inferred from other observing platforms
such as satellite retrievals provide quantitative AOD but with significantly higher uncertainty (Remer et al., 2005; Li
et al., 2009; Levy et al., 2010; Sayer et al., 2013). Further, in situ measurements lack the ability to provide a reliable
columnar AOD due to the requirement of measuring aerosols vertically in each layer while not perturbing or
modifying the particle properties during the measurement (Redemann et al., 2003; Andrews et al., 2017). Light
Detection and Ranging (LIDAR) is fundamental in the determination of the vertical aerosol extinction distribution
(Welton et al., 2000; Omar et al., 2013). Quantification of columnar AOD from ground-based LIDAR, for example,
may be less reliable due to low signal to noise ratio during the daylight hours at high altitudes and below the overlap
region in which the aerosols very near the surface are poorly observed by LIDAR. Satellite retrieval issues include
determining the AOD for very high aerosol loading episodes, cloud adjacency effects, land/water mask depiction,
surface reflectance, highly varying topography, and aerosol type assumptions (Levy et al., 2010; Levy et al., 2013;
Omar et al., 2013). With each of these measurement platforms, uncertainties exist with AOD; however, these
concerns are minimized with AOD measurements from surface based Sun photometry such as from the federated
Aerosol Robotic Network (AERONET). Ground-based Sun photometry, a passive remote sensing technique, is
robust in measuring collimated direct sunlight routinely during the daytime in mainly cloud-free conditions (Shaw
1983; Holben et al., 1998). While these surface-based measurements are only point measurements, the federated
AERONET provides measurements of columnar AOD and aerosol characteristics over an expansive and diverse
geographic area of the Earth's surface at high temporal resolution.
Standardization of Sun photometer instrumentation, calibration, and freely available data dissemination of AOD and
related aerosol databases highlights the success of the federated AERONET. For more than 25 years, the
AERONET federation has expanded due to the investments and efforts of NASA (GSFC), University of Lille
(PHOTONS/ACTRIS), and University of Valladolid (RIMA/ACTRIS) and other subnetworks (e.g., AEROCAN,
AeroSpan, AeroSibnet, CARSNET) and collaborators at agencies, institutes, universities, and individual scientists
worldwide. Conceived in the late 1980s, AERONET's primary objective was to provide an aerosol database for
validation of Earth Observing System (EOS) satellite retrievals of AOD and atmospheric correction. In addition to
columnar direct Sun AOD, sky radiances were used to infer aerosol characteristics initially from Nakajima et al.
(1996) (SkyRad.PAK) and later by the Dubovik and King (2000) inversion algorithm to obtain products such as
aerosol volume size distribution, complex index of refraction, single scattering albedo, and phase functions.



AERONET is a network of autonomously operated Cimel Electronique Sun/sky photometers used to measure Sun
collimated direct beam irradiance and directional sky radiance and provide scientific quality column integrated
aerosol properties of AOD and aerosol microphysical and radiative properties (Holben et al., 1998).  The
development and growth of the program relies on imposing standardization of instrumentation, measurement
protocols, calibration, data distribution and processing algorithms derived from the best scientific knowledge
available.  This instrument network design has led to a growth from two instruments in 1993 to over 600 in 2018.
During that time, improvements were made to the Cimel instruments to provide weather-hardy, robust
measurements in a variety of extreme conditions.  While the basic optical technology has evolved progressively
from analog to digital processing over the past 25 years, the most recent Sun/sky/lunar CE318 Model T instruments
provide a number of new capabilities in measurement protocols, integrity, and customizability.

All of the slightly varying models of the Cimel instruments can have measurement anomalies affecting direct Sun
measurements which include measurements in the presence of clouds, various obstructions in the instrument's field
of view, or systematic instrumental issues such as electrical connections, high dark currents, and clock shifts to
name a few.  Some of these issues depend on instrument model and, for more than a decade, these anomalies have
been removed semi-automatically utilizing the cloud screening method developed by Smirnov et al. (2000) and
further quality controlled by an analyst to remove additional cloud contaminated data and instrument artifacts from
the database.  Chew et al. (2011) identified up to 0.03 AOD bias at Singapore due to optically thin cirrus clouds for
Version 2 Level 2.0 data.  Coincidentally, Huang et al. (2011) examined how cirrus clouds could contaminate AOD
measurements up to 25% (on average) of the data in April at Phimai, Thailand, in the Version 2 Level 2.0 data set.
The number of AERONET sites has increased to more than 600 sites in the network as of 2018 and the labor
intensive effort of quality controlling hundreds of thousands of measurements manually had resulted in a significant
delay of quality assured data (Level 2.0) in the AERONET Version 2 database.

With these issues at hand, the cloud screening quality control procedure was reassessed as well as all other aspects
of the AERONET processing algorithm including instrument temperature characterization, ancillary data set
updates, and further quality control automation.  Utilizing these improvements, the Version 3 Level 2.0 quality
controlled dataset requires only the pre-field and post-field calibrations to be applied to the data so these data can
now be released within a month of the final post-field instrument calibration instead being of delayed up to several
months.  As encouraged by the AERONET community, automatic quality controls in Version 3 are now also applied
to near real-time Level 1.5 AOD products allowing for improved data quality necessary for numerous applications
such as numerical weather prediction, atmospheric transport models, satellite evaluation, data synergism, and air
quality.

The AERONET Version 3 processing algorithm marks a significant improvement in the quality controls of the Sun
photometer AOD measurements particularly in near real-time.  The revised AERONET algorithm is introduced by
first reviewing the calculations made to compute the AOD plus changes in the input data sets and the resulting





calculation of optical depth components. Next, the preprocessing steps and data prescreening are discussed for the
Version 3 quality control algorithm. Cloud screening and instrument quality control algorithm changes are
discussed with reference to Smirnov et al. (2000), and the solar aureole cirrus cloud screening quality control is
introduced for the first time. The automation of instrument anomaly quality controls and additional cloud screening
is described in the subsequent sections. Lastly, the AERONET Version 2 and Version 3 database results are
analyzed for the entire data set as well as for selected sites.
**2    Aerosol Optical Depth Computation**
Sun photometry is a passive remote sensing measurement technique in which mainly collimated light generally not
scattered or absorbed by the atmosphere illuminates a photodiode detector and this light energy is converted to a
digital signal. The digital signal (V) measured by the instrument is proportional to the solar irradiance. The relative
solar calibration is derived from the Langley method (Ångstrom 1970; Shaw et al., 1973) utilizing the digital counts
from the instrument versus the optical air mass to obtain the calibration coefficient ($V_o$) by choosing the intercept
where optical air mass is zero at the top of the atmosphere (Shaw, 1983). The relative extraterrestrial solar
irradiance is proportional to $V_o$. As shown by Holben et al. (1998) and for completeness in this discussion, the Beer-
Lambert-Bouguer law converted to instrument digital counts is shown in Eq. (1):

$$V(\lambda) = V_o(\lambda) * d^2 * \exp[-\tau(\lambda)_{Total} * m], \tag{1}$$

where $V(\lambda)$ is the measured spectral voltage of the instrument dependent on the wavelength ($\lambda$), $V_o(\lambda)$ is the relative
extraterrestrial spectral calibration coefficient dependent on $\lambda$, $d$ is the ratio of the average to the actual Earth-Sun
distance (Michalsky, 1988; USNO, 2018), $\tau(\lambda)_{Total}$ is the total optical depth, and $m$ is the optical air mass, which is
strongly dependent on the secant of the solar zenith angle (Kasten and Young, 1989). For the Cimel Sun
photometer, the voltage signal is expressed as integer digital counts or digital number (DN). The error in the $\tau(\lambda)_{Total}$
is dependent on the optical air mass ($m$) by $\delta\tau$ proportional to $m^{-1}$ and hence the AOD computation error will be
maximum at $m$=1 (Hamonou et al., 1999). The absolute uncertainty in the AOD measurement can be described as
Eq. (2), with calibration uncertainty of $V_o$ being the overwhelmingly dominant error source:

$$\delta\tau = \frac{1}{m} * \left(\frac{\delta V}{V} + \frac{\delta V_o}{V_o} + \tau * \delta m\right) \cong \frac{1}{m} * \frac{\delta V_o}{V_o} \tag{2}$$


The spectral aerosol optical depth (AOD; $\tau(\lambda)_{Aerosol}$) should be computed from the cloud-free spectral total optical
depth ($\tau(\lambda)_{Total}$) and the subtraction of the contributions of Rayleigh scattering optical depth and spectrally dependent
atmospheric trace gases as shown in Eq. (3).

$$\tau(\lambda)_{Aerosol} = \tau(\lambda)_{Total} - \tau(\lambda)_{Rayleigh} - \tau(\lambda)_{H_2O} - \tau(\lambda)_{O_3} - \tau(\lambda)_{NO_2} - \tau(\lambda)_{CO_2} - \tau(\lambda)_{CH_4} \tag{3}$$






Table 1 provides a list of the spectral corrections used in the calculation of AOD and precipitable water from
935nm. The nominal standard aerosol wavelengths are 340nm, 380nm, 440nm, 500nm, 675nm, 870nm, 1020nm,
and 1640nm. For wavelengths shorter than and equal to 1020nm, these channels are measured using a Silicon
photodiode detector with a spectral range from 320nm to 1100nm. If the Cimel instrument has an InGaAs detector
with a 900nm to 1700nm spectral range, then the 1640nm wavelength is measured along with a redundant 1020nm
measurement used to compare instrument optical characteristics between detectors, lenses, and collimator tubes.
The Cimel SEAPRISM instrument models, which are deployed on ocean or lake platforms as part of the
AERONET-Ocean Color component to retrieve normalized water leaving radiances at 8–12 additional visible band
wavelengths for ocean and lake remote sensing studies, are similarly corrected for atmospheric effects (Zibordi et
al., 2010).

Rayleigh optical depth calculations require the use of the station pressure (Bodhaine et al., 1999) as well as the
optical air mass (Kasten and Young 1989). To determine AERONET site station pressure ($P_S$), the NCEP/NCAR
reanalysis mean sea level pressure and geopotential heights at standard levels (1000hPa, 925hPa, 850hPa, 700hPa,
and 600hPa) are fitted by a quadratic function in logarithmic space to infer the station pressure at the corresponding
interpolated geopotential height. The NCEP/NCAR reanalysis data are available routinely at six hourly temporal
resolution and 2.5 degrees spatial resolution (Kalnay et al., 1996). Errors in the station pressure are generally less
than 2hPa when the station elevation is accurate and the weather conditions are benign (i.e., atmospheric pressure
tends to be stable), since aerosol measurements are typically performed in mainly cloud-free conditions.

The 935nm wavelength is used to determine the water vapor optical depth contribution, which is consequently
subtracted from the longer aerosol wavelengths (i.e., 709nm SEAPRISM, 1020nm, and 1640nm). The AOD at
935nm is extrapolated based on the Ångstrom exponent (AE) computed from the linear regression of the AOD and
wavelengths in logarithmic space within the range of 440–870nm excluding channels affected by water vapor
absorption (Eck et al., 1999). To extract the precipitable water (PW) in cm from the 935nm measurements, the
Rayleigh optical depth and the AOD components need to be subtracted from the total optical depth at 935nm. As a
result, the dimensionless column water vapor abundance ($u$) is obtained using the following equations:

$$T_W = \ln\left[T_{935nm[Measured]}\right] - \ln\left[T_{935nm[Extrapolated]}\right] \tag{4}$$

$$-\ln[T_W] = \ln\left[V_{o\ 935nm} * d^{-2}\right] - \ln[V_{935nm}] - m * \left(\tau_{935nm\ AOD} + \tau_{935nm\ Rayleigh}\right) \tag{5}$$

$$\ln\left[\frac{T_W}{C}\right] = -A * (m_W * u)^B \tag{6}$$



$$u = \frac{\left[\frac{\ln T_W}{-A}\right]^{1/B}}{m_W} \qquad (7)$$


where $T_W$ is the water vapor transmission and constants $A$ and $B$ are absorption constants unique to the particular
935nm filter, $C$ is an absorption constant assumed to be equal to one (Ingold et al., 2000), $d$ and $m$ are defined in Eq.
(1), $m_W$ is the water vapor optical air mass (Kasten et al., 1965), and $u$ is the total column water vapor abundance
(Schmid et al., 2001; Smirnov et al., 2004). The total column water vapor abundance ($u$) is converted to total
column water content or PW by using the normalization factor ($u_o$=10 kg/m$^2$) and dividing it by the mean value of
water density ($p_o$=1000 kg/m$^3$) to obtain water column height units of cm (Bruegge et al., 1992; Ingold et al., 2000).

In the calculation of the filter dependent $A$ and $B$ constants, the water vapor absorption optical thickness is
determined by the integration of water vapor extinction coefficient over height from the bottom to the top of the
atmosphere. This calculation requires the following inputs to determine the extinction at each height: HITRAN
spectral lines with assumed US1976 model standard atmosphere temperature and pressure profiles; the absorption
continuum look up table from the Atmospheric and Environmental Research (AER) Radiative Transfer Working
Group (Clough et al., 1989; Mlawer et al., 2012); and Total Internal Partition Sums that define the shape and
position of lines dependent on temperature (Gamache et al., 2017). Nine defined total column water vapor amounts
(0.5 cm, 1.0 cm, 1.5 cm, 2.0 cm, 2.5 cm, 3.0 cm, 4.0 cm, 5.0 cm, and 6.5 cm) are used to generate water vapor
absorption optical depth lookup tables. From these lookup tables, transmittances are calculated based on the
bandpass and averaged spectral solar irradiance for the quiet Sun obtained from the University of Colorado
LASP/NRL2 model (Coddington et al., 2016) to generate filter-specific $A$ and $B$ coefficients. The one sigma
uncertainty in the calculation of PW in cm is expected to be less than 10% compared to GPS precipitable water
retrievals (Halthore et al., 1997). The spectral water vapor optical thickness ($\tau_{H2O}(\lambda)$) is determined by computing
the average of all $A$ and $B$ constants from the suite of filters affected by water vapor absorption (i.e., 709nm
SEAPRISM, 935nm, 1020nm, and 1640nm) in the AERONET database. The $\tau_{H2O}(\lambda)$ is also dependent on the
dimensionless total column water vapor abundance (Michalsky et al., 1995; Schmid et al., 1996):

$$\tau_{H_2O}(\lambda) = \bar{A}(\lambda) + \bar{B}(\lambda) * u \qquad (8)$$


The contribution of ozone ($O_3$) optical depth is determined utilizing the total column Total Ozone Mapping
Spectrometer (TOMS) monthly average climatology (1978–2004) of $O_3$ concentration at 1.00° x 1.25° spatial
resolution, the $O_3$ optical air mass using $O_3$ scale height adjustment by latitude (Komhyr et al., 1989), and the $O_3$
absorption coefficient (Burrows et al., 1999). The OMI $O_3$ data set is not used here due to instrument sampling
anomalies (McPeters et al., 2015). While the TOMS $O_3$ data set is extensive and generally characterizes the
distribution of $O_3$, recent changes in concentration could introduce some minor uncertainty in AOD. Similarly, the
nitrogen dioxide ($NO_2$) optical depth is calculated using the total column OMI monthly average climatology (2004–
2013) of $NO_2$ concentration at 0.25° x 0.25° spatial resolution and the $NO_2$ absorption coefficient (Burrows et al.,



1998). Tropospheric $NO_2$ is highly variable spatially due to various source emissions and stratospheric $NO_2$
concentrations are more stable spatially than the tropospheric $NO_2$ (Boersma et al., 2004), therefore, regions with
high tropospheric $NO_2$ emission will tend to have greater proclivity for deviating from climatological means.
Further, $NO_2$ can vary significantly on the diurnal scale (Boersma et al., 2008). Improved satellite observations,
models, or collocation with surface-based PANDORA instruments measuring temporal total column $O_3$ and $NO_2$
may assist in reducing the uncertainty and determination of the total column $NO_2$ optical depth contribution in later
versions of the algorithm (Herman et al., 2009; Tzortziou et al. 2012). Concentrations for carbon dioxide ($CO_2$) and
methane ($CH_4$) are assumed constant and optical depths are computed based on the HITRAN-derived absorption
coefficients of 0.0087 and 0.0047 for the 1640nm filter, respectively, and adjusted to the station elevation.

The calibration of the AOD measurements is traced to a Langley measurement performed by a reference instrument.
The reference instruments obtain a calibration based on the Langley method morning only analyses based on
typically 4 to 20 days of data performed at a mountaintop calibration sites. The primary mountaintop calibration
sites in AERONET are located at Mauna Loa Observatory (19.536° N, 155.576° W, 3402 m) on the island of
Hawaii and Izana Observatory (28.309° N, 16.499° W, 2401 m) on the island of Tenerife in the Canary Islands
(Toledano et al., 2018). These reference instruments are routinely monitored for stability and typically recalibrated
every three to eight months. Reference instruments rotate between mountaintop calibration sites and inter-calibration
facilities at NASA GSFC (38.993° N, 76.839° W, 87 m) in Maryland, Carpentras (44.083° N, 5.058° E, 107 m) in
France, and Valladolid (41.664° N, 4.706° W, 705 m) in Spain, where reference instruments operate simultaneously
with field instruments to obtain pre-field and post-field deployment calibrations. For periods when the AOD is low
($\tau_{440nm}$<0.2), optical air mass is low ($m$<2), and aerosol loading is stable, the reference Cimel calibration may be
transferred to field instruments (Holben et al., 1998).

The Version 2 processing used default temperature corrections based on three sensor head temperature ($T_S$) ranges
($T_S$<21°C, 21°C≤$T_S$≤32°C, and $T_S$>32°C) using a constant nominal temperature sensitivity only for the 1020nm filter
direct Sun measurements. In Version 3, measurement temperature sensitivity has been updated for all wavelengths
≥400nm and all measurement types (i.e., direct solar, sky, water, and lunar viewing measurements). Beginning in
2010, the temperature sensitivity was characterized for almost all wavelengths uniquely for each Cimel instrument.
The temperature effect on signal is a function of the combined sensitivity of the detector and the filter material itself.
If any Cimel data relying on a filter was in use prior to 2010 and the filter was not temperature characterized, then
the default values for the filter and manufacturer type are applied, if established. Filters in the ultraviolet (i.e.,
340nm and 380nm) are not measured for temperature dependence because of low integrating sphere radiance output
at these wavelengths. Due to temperature dependence of the field instrument and the reference instrument, the Sun
and sky calibration transfer needs to be adjusted by computing the ratio of the Cimel temperature coefficients for
each wavelength and for the temperature observed at the time of the calibration. In addition, when the AOD is
computed for field instruments, the sensor head temperature is measured for each direct Sun measurement so these



data can be adjusted to the temperature response of the instrument optics (i.e., combined effect of the detector and
filters) and electronics.

The temperature response is measured at the AERONET calibration facilities using an integrating sphere and a
temperature chamber where the temperature is varied from −40°C to +50°C.  The wavelength dependent
temperature coefficient is typically determined from the slope of ordinary least squares regression fit of the digital
voltage counts versus the sensor head temperature reading.  For this relationship, the second order polynomial fit is
computed for 1020nm, while other filters use either a linear or second order polynomial fit (depending on the larger
correlation coefficient).  For Cimel Model 4 and some Model 5 instruments with two Silicon photodiode detectors,
the digital counts for solar aureole and sky instrument gains are used to determine temperature coefficients for each
detector.  Some Model 5 and all Model T instruments perform the direct Sun and sky measurements on the same
detector (Silicon or InGaAs) and typically utilize the solar aureole gain digital counts.

According to Holben et al. (1998), all instruments generally perform measurements sequentially from longer
wavelength to the shortest wavelength filters on a rotating filter wheel inside the sensor head, which positions each
filter in front of the photodiode detector and behind the sensor head lenses and collimator tube.  The robotically
controlled sensor head points automatically at the Sun based on the time and geolocation of the instrument.  The
laboratory tuned 4-quadrant detector provides nearly perfect solar and lunar tracking to one motor step or ~0.1°
immediately following the geographic pointing.  A dual tube external collimator with internal baffles attached to the
top of the sensor head reduces stray light effects into the sensor head 1.2° field of view optical train.

The instrument performs measurements of the Sun using measurement triplets, that is, performing the series of
measurements of all filters at time hh:m0:00 (time notation for hours, minutes, seconds), where for duration of about
eight seconds, and then repeating these measurements at hh:m0:30 and hh:m1:00.  The resulting one-minute
averaged measurement sequence is defined as a triplet measurement and the maximum to minimum range of these
measurements is termed the triplet variability.  The triplet measurement advantageously allows for separation of
homogeneously dispersed aerosols versus highly temporally variable clouds.  The triplet measurements are
performed either every 15 minutes for older Model 4 instruments or every three minutes for newer Model 5 and
Model T instruments increasing the temporal availability of the AOD measurements in the AERONET database.
**3    Automatic Quality Controls of Sun Photometrically Measured Aerosol Optical Depth**
The AERONET database has provided three distinct levels for data quality: Level 1.0, Level 1.5, and Level 2.0.  In
Version 2, Level 1.0 was defined as prescreened data, Level 1.5 represented near real-time automatically cloud-
cleared data, and Level 2.0 signified automatically cloud-cleared, manually quality controlled data set with pre and
post-field calibrations applied.  In Version 3, the definitions have been modified substantially for Level 1.5 and
Level 2.0.  Version 3 Level 1.5 now represents near real-time automatic cloud screening and automatic instrument
anomaly quality controls and Level 2.0 additionally applies pre-field and post-field calibrations. The Version 3 fully





automated cloud screening and quality control checks eliminate the need for manual quality control and cloud
screening by an analyst and increases the timeliness of quality assured data.  Note that in all cases each subsequent
data quality level requires the previous data level to be available as input (e.g., Level 1.5 requires Level 1.0 and
Level 2.0 requires Level 1.5).  The following sections will describe these new definitions and automatic quality
controls in detail and the impact these new quality assurance measures have on the AERONET database.

**3.1    Preprocessing Steps and Prescreening**
Most preprocessing data quality criteria operate on voltage (V, expressed as the integer digital number (DN)) or
sensor head temperature ($T_S$).  The impact of these conditions may immediately remove data from Level 1.0
consideration or later only impact Level 1.5 and Level 2.0 AOD.  Each quality control section describes the
reasoning for the screening at the specified data quality level.  Digital count anomalies typically result from
anomalous electronic issues such as very low or high battery voltages, malfunctioning amplifiers, or loose
connections of internal control box components.  These digital count anomalies mostly affect older instruments
(Cimel Models 4 (CE318-1) and 5 (CE318N)), while several of these connection issues have been mitigated in the
newest instruments (Cimel Model T (CE318-T)).
**3.1.1    Electronic Instability**
Cimel Model 4 instruments use a 16-bit analog/digital (A/D) converter in the processing unit in which the analog
signal from the sensor head detector to the control box is subject to electronic noise. Cimel Model 5 instruments use
a 16-bit A/D converter inside the sensor head and the instrument invokes electronic chopping to reduce electronic
noise.  Cimel Model T instruments utilize an increased quantization from 16 bits to 24 bits, which significantly
reduces noise effects. Cimel Model 5 and Model T instruments internally adjust for the dark current ($V_D$) with each
measurement and no separate record is logged. Cimel Model 4 instruments perform $V_D$ measurements after each
sky scan (approximately hourly) for each spectrally dependent instrument gain parameter (i.e., Sun, aureole, and
sky). Large $V_D$ values generally represent significant instrument electronic instability. Quality controls applied to the
$V_D$ will remove the entire day for Model 4 instrument data from all of the quality levels for either of the following
conditions: 1) a single dark current measurement is greater than 100 counts for greater than N-1 wavelengths, where
N is the total number of wavelengths or 2) more than three dark current measurements are greater than 100 counts
for three or more wavelengths.

Amplifiers in the Cimel Model 4 instruments can produce unphysical increases in the digital counts or decreases in
the AOD for the 340nm and 380nm wavelengths at large optical air mass (Fig. 1). These instability issues are
evaluated simply using a relative threshold with respect to the available visible wavelength AOD measurements.  If
the $\tau_{380}$ is greater than $0.5 * \tau_{340}$ and $(\tau_{440} + \tau_{500\ or\ 675} < \tau_{380} + \tau_{340} - 2.0)$, then the triplet measurements for 340nm and
380nm are removed from the database for Level 1.5 and subsequent levels. These quality controls are limited to
Model 4 instruments that were not manufactured after 2001; however, the early AERONET database (1993–2005)



contains much of these data.  New Cimel Model T instruments are replacing Model 4 instruments but over 40 Model
4 instruments remain active in 2018.

The instrument may rarely malfunction by producing constant digital voltages for triplet measurements and the
result of keeping these data in the database leads to unphysical variations in the AOD. A frequency analysis is
performed to determine if any digital number (DN) values occur more than 10 times in a day.  If more than 50% of
the DNs are from the same triplet measurement, then this measurement is identified as an anomalous measurement.
If more than 50% of the triplet measurements in the day are considered anomalous, then the entire day will be
removed from Levels 1.5 and 2.0.
**3.1.2    Radiometer Sensitivity Evaluation**
The Cimel 4-quadrant solar near infrared detector requires enough sensitivity to track the Sun and a DN threshold of
100 in the near infrared is needed to have sufficient signal.  Near infrared wavelengths (e.g., 1020nm) typically have
a higher measured solar DN(V) due to higher atmospheric transmission in the presence of fine mode dominated
aerosols even in very high aerosol loading conditions.  When the DN ($V_{870nm}$ or $V_{1020nm}$) is less than 100 counts for
any measurement of the solar triplet, then the entire solar triplet AOD will be removed for all wavelengths from
Level 1.0 and subsequent levels due to potential solar tracking accuracy issues.

Version 2 data processing assessed the instrument electronic and diffuse light sensitivity by defining a digital
number (DN) of 10 to remove solar AOD triplet measurements.  Electronic issues impact Cimel Model 4
instruments in the UV and short visible wavelengths due to high $DN(V_D)$. Scattered diffuse light into the collimated
field of view can affect all instruments and produce unusual AOD changes with optical air mass especially when the
aerosol loading is high and optical air mass is large. The signal to noise ratio of the Cimel instrument requires setting
a minimum threshold for the determination of the solar measured DN(V) to limit the effect of diffuse radiance in the
instrument field of view (Sinyuk et al., 2012).  When a dark current $DN(V_D)$ ( e.g., ~50–100) is nearly equal to or
larger than the measured solar DN(V) (e.g., ~25–50) will result in V and $\tau$ decreasing with increasing optical air
mass.  All wavelengths are evaluated to determine if the measured solar DN(V) (subtracted from the closest
temporal dark current $DN(V_D)$ for Model 4 instruments only) is less than $DN(V_O)/1500$, then the identified
wavelength will be removed from all AOD levels.  A threshold of 1500 is calculated from a DN of 15000, a typical
average $DN(V_O)$ for Cimel Models 4 and 5, normalized to a minimum signal DN of 10.  The maximum product of
AOD times optical air mass ($\tau_m = \tau * m$) of approximately 7.3 is computed by the natural logarithm of 1500 (i.e., ln
(15000/10)) for Cimel Model T instruments.  For non-Model T instruments, the 100 DN threshold for 870nm and
1020nm limits the $\tau_m$ to approximately 5.0 (i.e., ln (15000/100)) for only those two wavelengths.  The $\tau_m$ maximum
threshold applies to all channels; however, the signal count can decrease significantly with optical air mass and
depend on the wavelength dependence of $V_O$.  For values exceeding the $\tau_m$ maximum threshold, the diffuse radiation
increases the signal and, as a result, unfiltered AODs show a decrease in magnitude as optical air mass increases for



high AOD even when DN($V_D$) equals zero. A measured solar DN(V) lower than the ratio DN($V_O$)/1500 threshold
will result in the removal of the solar triplet AOD for the specific wavelength (Fig. 2).

### 3.1.3    Digital Voltage Triplet Variance

As mentioned in Sect. 2, the Cimel instrument performs a direct Sun triplet measurement at regular intervals
throughout the day. A variance threshold is applied based on the root mean square (RMS) differences of the triplet
measurements relative to the mean of these three values. If the (RMS/mean)*100% of the triplet values is greater
than 16%, then these data are not qualified as Level 1.0 AOD (Eck et al., 2014). The temporal variance threshold is
sensitive to clouds with large spatial-temporal variance in cloud optical depth and optically thick clouds such as
cumulus clouds as well as issues due to poor tracking of the instrument.

### 3.1.4    Sensor Head Temperature Anomaly Identification

Each Cimel instrument has a fixed resistance (Model 4) or band gap (Models 5 and T) temperature sensor inside the
optical head within 0.5 cm of the detector, filter wheel, and optical train assembly. As discussed in Sect. 2, the
instrument optics and digital counts can have dependence to the sensor head temperature ($T_S$) which is saved with
each measurement triplet. Sensor head temperatures may be erroneous due to instrument electronic instability or
communication issues. These potentially unphysical values of $T_S$ are evaluated by a number of algorithm steps such
as checks for 1) constant $T_S$ values, 3) unphysical extreme high or low $T_S$, 4) potentially physical yet anomalously
low $T_S$ with respect to the NCEP/NCAR reanalysis ambient temperatures, and 5) unphysical $T_S$ decreases or
increases (dips). When the algorithm removes a $T_S$ reading or the $T_S$ measurement is missing, an assessment is made
on the instrument temperature response based on ±15°C of the NCEP/NCAR reanalysis temperature for the date and
location to determine whether the temperature characterization coefficient for a specific wavelength would result in
a change of AOD by more than 0.02. If this condition is met for a specific wavelength, then data associated with
this wavelength-specific triplet measurement will be removed at Level 1.5 and subsequent levels while preserving
other less temperature dependent spectral triplet measurements.

### 3.1.5    Eclipse Circumstance Screening

During episodic solar or lunar eclipses, AOD will increase to the maximum obscuration of the eclipse at a particular
location on the Earth's surface. The AOD increases due to the reduction of the irradiance due to the celestial body
(Moon or Earth) obscuring the calibrated light source (Sun or Moon). While any one point on Earth infrequently
experiences an eclipse, when an eclipse episode does occur, the eclipse can affect many locations nearly
simultaneously making manual removal tedious at sites distributed globally. To automate the removal of eclipse
episodes, the NASA solar and lunar eclipse databases are queried for eclipse circumstances based on geographic
position of the site to produce a table of eclipse episodes starting from 1992. The eclipse tool utilizes established
Besselian elements based on the Five Millennium Canon of Solar Eclipses: −1999 to +3000 (Espenak and Meeus
2006) to quantify the geometric and temporal position of the celestial bodies (Sun, Earth, and Moon), determine the
type of eclipse (e.g., partial, annular, total), and predict times of the various stages of the solar or lunar eclipse. For





the Version 3 database, the eclipse site-specific tables are used to discretely remove triplet measurements affected by
any stage of the eclipse circumstance.  For example, during a solar eclipse, solar triplets will be removed between
the partial eclipse first contact to the partial eclipse last contact regardless of the eclipse obscuration or magnitude
for Level 1.5 data and subsequent levels (Fig. 3). The partial eclipse first contact is defined as the time at which the
penumbral shadow is visible at a point on the Earth's surface and the partial eclipse last contact is defined as the
time at which the penumbral shadow is no longer visible a point on the Earth's surface. Efforts to retain AOD during
solar eclipse episodes have been attempted by the authors in which up to 95% of the AOD can be corrected based on
adjusting calibration coefficients by the eclipse obscuration.  However, spectral calibration coefficients also need to
be adjusted to account for the solar atmosphere spectral irradiance, which becomes more dominant during the solar
eclipse episode and is a topic of further investigation.

### 3.1.6   Very High AOD Retention

Cloud screening procedures in the next section may inadvertently remove aerosol in very high aerosol loading cases
due to biomass burning smoke and urban pollution as discussed by Smirnov et al. (2000).  For Version 3, each triplet
reaching Level 1.0 is evaluated for possible retention in the event that a specific Level 1.5 cloud screening procedure
removes the triplet.  When the AOD measurement for 870nm is >0.5 and AOD 1020nm >0.0, these conditions will
potentially qualify the triplet for very high AOD retention.  Further analysis is performed on those qualified triplets
to remove the effect of heavily cloud-contaminated data using the AE for the wavelength ranges of 675–1020nm or
870–1020nm (Eck et al., 1999).  If the $AE_{675-1020nm}$>1.2 (or $AE_{870-1020nm}$>1.3, if $AOD_{675nm}$ is not available), and the
AE for the same range is less than 3.0, then the triplet qualifies for very high AOD retention and the triplet can be
retained at Level 1.5 even if the measurement does not pass Level 1.5 cloud screening quality control steps in Sect.

402   3.2.

### 3.1.7   Total Potential Daily Measurements

Cloud screening methods in Sect. 3.2 may incompletely remove all cloud-contaminated points and leave data
fragments. To mitigate this issue, a methodology was developed based on the total number of potential
measurements in the day and calculated AE values. The total number of potential measurements in the day is
defined as the number of triplet measurements plus the number of humidity status reports (i.e., wet sensor
activations). If the number of remaining measurements after all screening steps in Sect. 3.2 are performed is less
than three measurements or 10% of the potential measurements (whichever is greater), then the algorithm will
remove the remaining measurements. This condition is repeated after each cloud screening step in Sect. 3.2 and will
only be activated when the very high AOD restoration is not triggered (see Sect. 3.1.6) or when the $AE_{440-870nm}$ is
less than 1.0 for a triplet measurement indicating large particles such as clouds may contaminate the remaining
measurements.



**3.1.8    Optical Air Mass Range**
The basic Cimel Sun photometer Sun and sky measurement protocols were specified to NASA requirements in
Hoblen et al. (1992, 1998, and 2006), and have only been slightly modified since that time for improved
measurement capability of the Model 5 and Model T instruments.  All instruments systematically perform direct Sun
measurements between the optical air mass ($m$) of 7.0 in the morning and $m$ of 7.0 in the evening.  In Version 2 and
earlier databases, AERONET data processing limited the Level 1.5 and Level 2.0 AOD computation from $m$ of 5.0
in the morning to $m$ of 5.0 in the evening.  The $m$ limitation may avoid potential error in the computation of the
optical air mass at large solar zenith angles (Russell et al., 1993) and possible increased cloud contamination
(Smirnov et al., 2000).  For Version 2 and 3 processing, the Kasten and Young 1989 formulation was used to
account for very small differences in the optical air mass calculations at high solar zenith angles.  Noting that the
AOD error ($\delta\tau/m$) has a minimum at large $m$ (conversely a maximum at solar noon), the maximum $m$ of 5.0 was
extended to $m$ of 7.0 in Version 3 processing.  The larger optical air mass range leads to an increase in the number of
solar measurements occurring in the early morning and the early evening contributing to additional AOD
measurements used for input for almucantar and hybrid inversions plus an increase in AOD measurement at high
latitude sites when solar zenith angles may be large even at solar noon.  The impact on the cloud screening
performance appears to be minimal for measurements closer to the horizon.  The fidelity of the Version 3 cloud
screening (see Sect. 3.2) AODs supports the extended optical air mass range for Level 2.0.
**3.2    Level 1.5 AOD Cloud Screening Quality Controls**
As discussed in Sect. 3.1, several preprocessed criteria and parameters are necessary to quality control the AOD data
quality in near real-time (NRT).  Cloud screening procedures proposed by Smirnov et al. (2000) were designated to
remove or reduce cloud contaminated AOD measurements.  However, these procedures also had the effect of
surreptitiously removing occasionally other non-cloud anomalies such as repeated AOD diurnal dependence when
AOD had a large maximum at midday and minimum at high optical air masses due to environmental impacts on the
optical characteristics of the instrument (e.g., moisture on the sensor head lens or spider webs in the collimator
tube). While these cloud screening methods have been implemented for about 25 years, the state of knowledge has
progressed over this period and thus necessitates review and modification of cloud screening quality control
procedures (Kaufman et al. 2005, Chew et al., 2011; Huang et al., 2011).  The calculation of the AOD at Level 1.0
essentially represents the following in Eq. (9):

$$\tau_{app\,Total} = \frac{1}{\Gamma_{anomaly}}\left(\tau_{aerosol} + \frac{\tau_{cirrus}}{C_{cirrus}} + \tau_{liquid\,cloud} + \tau_{eclipse}\right)\qquad(9)$$


where $\tau_{app\,Total}$ is the apparent total optical depth, which at this point in the data processing, may be affected by the
contributions of liquid cloud droplets ($\tau_{liquid\,cloud}$), cirrus amplification factor ($C_{cirrus}$) applied to the cirrus crystal
optical depth ($\tau_{cirrus}$) due to strong forward scattering into the field of view of the instrument, solar or lunar eclipses
($\tau_{eclipse}$), and instrument anomalies ($\Gamma_{anomaly}$ adjustment factor).  Given cloud free conditions and perfect instrument





operation, the additional non-aerosol $\tau$ components would be zero and $C_{cirrus}$ and $\Gamma_{anomaly}$ would be one. However,
the Cimel Sun photometer always attempts to measure the Sun if it can be tracked regardless of the total optical
depth magnitude.

Clouds are a major factor in the effort to quality control remotely sensed aerosol data (Smirnov et al. 2000; Martins
et al. 2002; Kaufman et al., 2005; Chew et al., 2011; Kahn and Gaitley 2015). A significant portion of the liquid
cloud contribution is removed by the prescreening prior to Level 1.0 as discussed in Sect. 3.1.3. The $\tau_{app\ Total}$ should
be adjusted based on a multiplier dependent on the cirrus crystal size ($\tau_{correct}=C_{cirrus}*\tau_{app\ Total}$) according to Kinne et
al. (1999). While this cirrus coefficient ($C_{cirrus}$) is not specifically modelled by Kinne et al. (1999) for the Cimel
instrument field of view half angle of 0.6°, this multiplier is likely to be close to one for small cirrus crystals (e.g.,
$r_{eff}$=6µm–16µm), but near two for larger cirrus crystal sizes (e.g., $r_{eff}$=25µm–177µm). These adjustment factors
would result in the reduction of the $\tau_{app\ Total}$ due to forward scattering in the presence of cirrus. On the other hand,
liquid water cloud droplets would significantly increase the $\tau_{app\ Total}$ in a manner similar to large dust particles.

Cimel instruments also may have internal and external anomalous conditions that modify the optical characteristics
or response of the instrument resulting in amplification or dampening impacts ($\Gamma_{anomaly}$) of varying magnitudes on
the computation of the $\tau_{app\ Total}$. These anomaly adjustments can be difficult to quantify and can have strong
dependence on optical air mass ($m$) or the sensor head temperature ($T_S$). As a result, the following sections will
describe the mechanisms in which these additional cloud and anomaly components are automatically eliminated or
reduced as close to zero as possible to provide a quality assured AOD ($\tau_{aerosol}$) after final calibration is applied (see
Sect. 4) across the global AERONET database.
**3.2.1    Cloud Screening Quality Controls**
As Level 1.0 AOD data may have cloud contamination, these data should be considered as potentially cloud
contaminated where the triplet measurement represents the apparent AOD ($\tau_{app\ aerosol}$) as defined in the previous
section. Table 2 provides a summary of the cloud screening quality control changes from Version 2 to Version 3
and these changes are discussed in detail below and Sect. 3.2.2.

Cimel triplet measurements are performed typically every three minutes (every 15 minutes for older instrument
types) and these triplet measurements can detect rapid changes in the $\tau_{app\ aerosol}$ by analyzing the maximum to
minimum variability (i.e., the $\Delta\tau_{app\ aerosol}\{MAX-MIN\}$). Assuming that spatial and temporal variance of aerosols
plus clouds is much greater than aerosols alone, in many cases, $\Delta\tau_{aerosol}$ would be near zero and $\Delta\tau_{cloud}$ should be
much larger than zero when especially liquid phase cloud droplets exist. For Version 2 and earlier databases,
Smirnov et al. (2000) methodology utilized all available wavelengths to perform $\tau_{app\ aerosol}$ triplet screening for cloud
contamination. Therefore, large triplet variability would indicate the presence of clouds due to large $\Delta\tau_{cloud}$.
Analyses (e.g., Eck et al., 2018) have shown that removing the entire triplet measurement when only one or more of
the shorter wavelengths indicates a large variation ($\Delta\tau_{aerosol}(\lambda)$ much greater than zero) may not be the most robust





approach.  For example, in cases of highly variable fine mode aerosols such as smoke can produce large triplet
variability as a result of the inhomogeneous nature of the aerosol plume especially for shorter wavelengths (e.g.,
340nm, 380nm, 440nm) where fine mode dominated aerosol particles can have radii similar to short wavelength
measurements.

Considering these factors, several potential techniques were explored utilizing various wavelength combinations and
utilizing the Spectral Deconvolution algorithm (SDA) fine and coarse mode triplet separation (O'Neill et al., 2001,
2003).  While the SDA algorithm derived triplets for coarse mode AODs relative change tended to show utility in
cloud removal, the SDA algorithm itself could not be applied universally to the AERONET database to due
anomalous results in which fine and coarse mode AODs can have a negative relationship when the number of
available wavelengths or wavelength range is not satisfied.  Anomalies in SDA retrievals can occur when the
uncertainty in AOD is relatively large near solar noon compared to the magnitude of AOD as is sometimes the case
when only the pre-field deployment calibration has been applied.  Upon further consideration of the triplet
variability technique, analyses indicated that using the three longest standard AERONET wavelengths (i.e., 675nm,
870nm, 1020nm) could be used to remove a triplet measurement when they have high triplet variability that exceeds
0.01 or 0.015*AOD (whichever is greater).  The reduction in the threshold of the triplet variability criterion is
proportional to the magnitude decrease AOD uncertainty compared to UV wavelengths (0.02) to those of visible and
near infrared wavelengths (0.01).

While Smirnov et al. (2000) did not impose an Ångstrom exponent limitation; Version 3 processing constrains the
$AE_{440–870nm}$ of Level 1.5 data to be within −1.0 and +3.0.  In general, the $AE_{440–870nm}$ values outside this range are
unphysical and should not be used due to the inconsistency of the AOD spectral dependence.  These inconsistencies
typically occur at very low optical depth (<0.05) where the uncertainty of the AOD may be up to 100% of the actual
value thus producing AE values that are invalid.

The AOD time series smoothness uses a number of numerical methods and fits dependent on the application.  For an
AOD time series, rapid and large increases are usually the result of cloud contamination.  In Version 2 and prior
versions, a technique proposed by Smirnov et al. (2000) to implement a smoothness methodology similar to
Dubovik et al. (1995). In this scheme, the triplet measurements were considered as discrete points and differences in
logarithm of $\tau_{app\ aerosol}$ and relative difference in times between those measurements were utilized to calculate the
first derivative differences in which an arbitrary parameter D (similar to the norm of the second derivative) is
calculated.  In Version 2 and earlier versions, when the value of D was greater than 16 for an AOD measurement
time sequence for 500 nm or 440nm, then this triplet was removed from the data set. Further, the smoothness
procedure was repeated or measurements were rejected for the day if less than three triplets remained for the day as
discussed in Smirnov et al. (2000). While the D=16 threshold was empirically derived, the smoothness parameter is
somewhat arbitrary in origin and operates in logarithmic coordinates rather than natural ones.  For example, the
distribution of aerosol measurements in a single day is typically normally distributed rather than logarithmically





distributed. Further, the D parameter smoothness procedure was not always successful at removing cloud-
contaminated data and this may be related to the fact that the empirically derived D parameter was tuned for 15-
minute triplet measurement intervals rather than three-minute intervals now commonly observed in the network.
Therefore, an approach adhering to the relative change in the total optical depth with time is feasible and a more
straightforward physical quantification of the change in $\tau_{app\ aerosol}$ with time.

The AOD time series smoothness in Version 3 evaluates the same $\tau_{app\ aerosol}$ 500nm wavelength (or 440nm if 500nm
is not available). The Version 3 smoothness method computes the relative rate of change of $\tau_{app\ aerosol}$ per minute and
if $\Delta\tau_{app\ aerosol}/\Delta t > 0.01$ per minute, then the larger triplet measurement in the pair is removed and the smoothness
procedure will continue to remove triplets until measurement pairs in the day do not surpass the smoothness
threshold. The selection of this threshold of 0.01 per minute hinges on the premise that the triplet average does not
change rapidly within one minute. The Version 3 smoothness procedure could be affected by extreme changes in
AOD due to anomalous aerosol plumes (e.g., biomass burning or desert dust plumes) where no temporal gradient
exists.

After the cirrus cloud screening quality control (to be discussed in the Sect. 3.2.2), triplets are evaluated for spurious
or isolated measurements remaining during the day after applying the cloud screening quality control procedures.
So-called "standalone points" may be relevant given the ability of the instrument to perform measurements in cloud
breaks or gaps. Here, the definition of a standalone triplet is when no triplets are available within 1 hour of the
measurement. If the $AE_{440-870nm}$ is greater than 1.0, the algorithm retains the triplet measurement; otherwise, the
measurement will be removed from the data set. Finally, daily averaged data are evaluated for temporal stability
using the AOD stability during the day at 500nm (or 440nm) and daily outlier triplets using the 3-sigma check for
AOD at 500nm (or 440nm) and $AE_{440-870nm}$ to be within ±3 standard deviations (Smirnov et al. 2000). At this point
in the quality control algorithm, the remaining triplet measurements are not expected to have a major component of
$\tau_{cloud}$ or $\tau_{cirrus}$.

**3.2.2    Novel Cirrus Removal Method Utilizing Solar Aureole Curvature**
Utilizing satellite and surface-based LIDAR, studies have shown the AERONET Version 2 Level 2.0 AOD data are
impacted by homogeneous optically thin cirrus clouds with a bias up to 0.03 in AOD (DeVore et al., 2009; Chew et
al., 2011; Huang et al., 2011). The optically thin cirrus bias can influence radiative forcing calculations and satellite
validation when clouds contaminate the measurement (DeVore et al., 2012). In addressing the shortcoming of
Smirnov et al. (2000) and manual checks in which the identification of optically thin cirrus clouds give relatively
weak signal in the AOD or AE, the authors leveraged high angular resolution radiance measurements routinely
performed in the solar aureole region (3.2°–6.0° scattering angle range). While cirrus detection may be possible with
other scattering angle ranges, Cimel Sun photometer radiance measurements do not presently have high enough
angular resolution from 6.0°–35.0° to reliably and consistency detect cirrus induced atmospheric phenomena (e.g.,



solar halos and sun dogs), since these events depend on cirrus crystal shape and orientation and are not always detectable beyond levels of cloud optical depth variability.

The use of the solar aureole radiance ($L_A$; $\mu W/cm^2/sr/nm$) with respect to the scattering angle ($\varphi$; in radians) has been demonstrated using the Sun and Aureole Measurement (SAM) aureolegraph instrument to indicate the presence of large particles such as cirrus crystals (DeVore et al., 2009, 2012; Haapanala et al., 2017). The effect of the surface reflectance is much less than the radiance of the solar aureole so it is ignored; however, this may become important at very large solar zenith angles and bright surfaces such as snow (Eiden 1968). All Cimel instrument models perform solar aureole measurements at the nominal 1020nm wavelength. The Cimel performs solar triplet measurements directly on the solar disk, while solar aureole radiances are measured mainly during the almucantar, principal plane, and hybrid sky scans. These solar aureole measurements are performed hourly for Models 4 and 5 instruments during sky scan scenarios and for Model T instruments before each solar triplet as well as for the hourly almucantar and hybrid sky scan measurements.

The AERONET measurements of the solar aureole directional radiances ($L_A$) depend on the absolute calibration of the integrating sphere. The integrating spheres at the AERONET calibration centers provide an absolute calibration traceable to a NIST standard lamp hosted at the NASA GSFC calibration facility. The uncertainty in the radiance calibration is typically less than 3% due to systematic degradation in the lamp levels, changes in integrating sphere characteristics, and instrument spectral signal response. The solar aureole radiance magnitudes also depend on the instrument Sun sensitivity gain settings for each wavelength for Cimel Model 4 and 5 instruments, while the Model T instruments use an internal instrument gain switch applying to all wavelengths. The $L_A$ measurements have calibration and temperature correction applied and are measured by all Cimel instruments at the 440nm, 675nm, 870nm, and 1020nm wavelengths. Due to lower AOD in fine mode aerosol loading situations, less Rayleigh scattering, and lower calibration uncertainty, the $L_A$ measurements at 1020nm have less noise for evaluating cirrus cloud presence.

Given that the $L_A$ measurements are performed at discrete $\varphi$, we calculate the ordinary least squares linear regression fit on logarithmic scale when more than three scattering angles are available to determine the intercept ($a$), slope ($b$), and the correlation coefficient ($R$). If $R$ is less than or equal to 0.99, then we do not proceed to check for cirrus contamination. When $R$ is greater than 0.99, the curvature ($k_o$) for the first available scattering angle ($\varphi_o$) in the 3.2°–6.0° scattering angle range is calculated using the equation of curvature of the signed planar curve, which gives the rate of turning of the tangent vector in Eq. (10) (Kline 1998):

$$k = \frac{y''}{(1 + y'^2)^{\frac{3}{2}}}$$
(10)a



The curvature ($k$) can be formulated by assuming the Power Law function and its derivatives, and, in our
application, using the first scattering angle ($\varphi_o$) in radians for $\varphi$ below:

$$y = a * \varphi^b \tag{10)b}$$
$$y' = a * b * \varphi^{b-1} \tag{10)c}$$
$$y'' = a * b * (b-1) * \varphi^{b-2} \tag{10)d}$$


According to the $k$ formulation, the stronger the forward scattering peak, then the smaller the value of curvature
since the second derivative is small and the first derivative is large due to the steepness of the solar aureole
radiances. Further, the overall slope of curvature for all of the scattering angles (3.2°–6.0°) can be calculated using
the assumption that $y'^2 \gg 1$ rendering the addition of 1 in the denominator of Eq. (10)a insignificant. The slope of
the logarithm of curvature versus logarithm of scattering angle is desired and this slope can be calculated using $a$
and $b$ from the linear regression above by converting from logarithmic coordinates. Therefore, we derive the Eq.
(11 to determine the slope of curvature dependent only on the slope of the linear regression fit of $L_A$ and $\varphi$ on
logarithmic scale as follows:

$$\ln k = a + (1 - 2b) * \ln \varphi \tag{11}$$


Here, the slope of curvature ($M$) is defined as ($1-2b$). The value of $M$ will typically be positive since $b$ will tend to
be negative due to the dimming of the solar aureole with increasing scattering angle. Alternatively, $M$ can be
calculated numerically for each $k$ and $\varphi$ to obtain similar results. A small value of curvature ($k_o$) at the smallest
scattering angle available represents the possible existence of large particles producing a forward scattering peak.
The slope of curvature ($M$) represents the average characterization of the solar aureole shape across the scattering
angle 3.2°–6.0° range where a large magnitude signifies the potential presence of large particles as curvature
increases with increasing scattering angle across the forward scattering peak.

The Micropulse LIDAR Network (MPLNET) is a global network of LIDARs monitoring the vertical distribution of
aerosols and clouds (Welton et al., 2000, 2002; Campbell et al., 2002). To determine the thresholds for these Sun
photometer solar aureole curvature parameters for different surface types and aerosol environments, the MPLNET
LIDAR cloud identification database was used at eight collocated AERONET sites as shown in Table 3. Multi-year
MPLNET LIDAR deployment data were analyzed and matched with AERONET observations when the solar zenith
angle was less than 30° to minimize the spatio-temporal differences of the zenith pointing LIDAR versus the
slantwise pointing of the Sun photometer in which sky condition can be quite different at large solar zenith angles.
The MPLNET cloud base height data product was matched with MERRA reanalysis vertical temperature profile
corresponding to the geopotential height pressure surface. When a cloud top temperature is less than −37°C, a cloud
is designated to be cirrus, while other non-cirrus clouds may contain liquid or mixed phase particles (Sassen and
Campbell, 2001; Campbell et al., 2015; Lewis et al., 2016). The partitioning the AERONET data set of solar



aureole radiances in terms cirrus clouds, non-cirrus clouds, all clouds, and clear (no cloud base detected) sky
condition categories allowed for the empirical determination of potential thresholds for the curvature parameters.
For each site, AERONET curvature parameters ($k$ and $M$) were computed for almucantar and principal plane solar
aureole ($L_A$) measurements (i.e., left and right scans separately) and further categorized based on the coincident
LIDAR detected sky condition. These solar aureole radiances have calibration and temperature characterization
applied for the 1020nm channel and these $L_A$ measurements were only quality controlled based on the correlation
threshold of 0.99 discussed above.

Figure 4a shows the number distribution of the $k$ at NASA GSFC (38.99° N, 76.84° W) for each of the four LIDAR
sky condition categories. The number of the potential clouds is large for magnitudes of $k$ less than 2.0E−5.
Similarly, Fig. 4b and Fig. 4c show the number distributions of the $M$ at NASA GSFC for each LIDAR sky
condition category. In Fig. 4b, the number of potential clouds generally dominates when the $M$ is greater than 4.3
with generally clear or possibly cloudy conditions when $M$ is less than or equal to 4.3. Some overlapping of the
categories for $M$ may be related to the differences in the viewing geometry of the sky between the Sun photometer
and the LIDAR or inhomogeneous cloud conditions.

Algorithmically combining the two thresholds of $k$ and $M$ produces a defined distribution of clear versus cloudy sky
condition categories. When the threshold of $k<2.0E−5$ is applied first, then the distribution of mainly cloudy
conditions becomes more distinct as shown for NASA GSFC in Fig. 4c. The maximum in the number distribution
for cirrus is near $M=4.6$ and the maximum in the number distribution of clear sky condition is at $M=4.3$ (Fig. 4c). At
Singapore (1.29° N, 103.78° E), Fig. 5c suggests that the distinction of small aerosol particles and larger cirrus cloud
ice crystals allows for adequate separation to identify an observation as cloud contaminated using a threshold of $M$
greater than 4.3. Figure 6a shows the number distribution of the curvature at the first scattering angle for coincident
AERONET and MPLNET observations at the SEDE BOKER (30.85° N, 34.78° E). Figure 6c shows the distinction
is similarly distributed as GSFC and Singapore to potentially identified cirrus contaminated observations. For Fig.
6a, the clear sky condition category is much higher in number than other sky condition categories; however, the $k$
values less than the first scattering angle threshold of 2E−5 (shown by the orange vertical line) indicates a
significant presence of dust particles rather than cirrus clouds due to forward scattering of dust. Note that as for Fig.
4 and Fig. 5, the x-axis of Fig. 6a is truncated to 1E−4 but the number distribution continues at values near zero for
larger first point curvatures. SEDE BOKER data in Fig. 6c exhibits a significant contribution of clear conditions are
preserved indicating that this method does not appear to misidentify dust as cirrus at this mixed dust and urban
pollution site.

When evaluating all of the collocated AERONET/MPLNET sites in Table 3 (Fig. 7), the maximum in the number
distribution for cirrus is at $M=4.3$ after the $k<2.0E−5$ threshold is applied with a relative minimum for the clear
conditions for $M>4.3$. Given this information, an empirical threshold of $M>4.3$ can be established for maximizing
the removal of cirrus clouds and minimizing removal of potentially clear data points. As mentioned previously, the



almucantar and principal plane sky scans are performed on an hourly basis. If cirrus clouds are homogeneously
distributed in the sky, then this assumption allows for the application of the temporal screening of triplet
measurements within 30 minutes of the solar aureole measurement time. As a result, a significant number of cirrus
contaminated measurements for $M \leq 4.3$ are likely removed with this procedure given the normally distributed
number distribution of cirrus identified solar aureole measurements around $M=4.3$. For the Cimel Model T
instruments, sky scan aureole measurements are superseded by a special solar aureole scan (CCS) performed from
3.0° to 7.5° scattering angle range at 0.3° increments (left and right) after each triplet solar measurement; therefore,
temporal screening for these triplet measurements is applied within two minutes of the CCS scan. Overall, the
aureole curvature cirrus cloud screening quality control decreases the probability of a cirrus bias in the AOD data set
globally by using this standard procedure. However, the Version 3 Level 1.5 AOD data set may still be influenced
by optically thin or sub-visible cirrus clouds with ice crystals similar in diameter to coarse mode aerosols such as
those found at polar latitudes or when solar aureole measurements are not available due to instrument malfunction or
incomplete data transfer.

Figure 8 shows solar aureole radiances have significant nonlinearity with scattering angle when impacted by cirrus
clouds while measurements without cirrus are more linear. The SEDE BOKER site is influenced by desert dust.
Dust particles can affect the calculation of the $k$ parameter to be close to the threshold of 2E−5 even when cirrus
clouds are not present (SEDE BOKER case 1); however, the overall slope is more linear for the non-cirrus case
compared to the cirrus case (SEDE BOKER case 2). As a result, the $M$ parameter is much lower and the algorithm
action would be to preserve the SEDE BOKER Case 1 data and remove data for SEDE BOKER case 2. Note that
the $k$ parameter is quite low for SEDE BOKER Case 1 and in general dusty sites may frequently have $k$ less than
2E−5; therefore, the $M$ curvature parameter is needed to prevent inadvertent removal of aerosol data. For fine mode
at GSFC case 1 and Singapore, small values of $k$ and large values of $M$ result in removal of the cirrus-contaminated
data. For comparison, the GSFC case 2 shows significant linearity when cirrus clouds are not present. The GSFC
case 3 and Trinidad Head case show the variation in these curvature parameters at low optical depths in which only
one of the curvature parameters indicates the possibility of cirrus clouds. While these two curvature parameters may
be used independently in certain conditions, the current algorithm must employ both curvature parameter thresholds
to avoid inadvertently identifying aerosols as clouds in dust and low aerosol loading conditions.
**3.3**   Level 1.5 Quality Controls to Screen Instrument Anomalies
While cloud-screening quality controls remove a significant portion of data impacted by cloud contamination and
some instrument anomalies, a portion of the remaining AOD data set can be impacted by internal or external
instrument anomalies. Most instrument anomalies can be removed utilizing the prescreening steps outlined in the
Sect. 3.1, but a number of issues still exist which are more evident after the cloud screening quality controls have
been applied to the data set. A data set with some clouds can mask or offset patterns in the AOD spectra that can
clearly identify data anomalies dependent on optical air mass. For AERONET instruments, data anomalies either
dependent on the optical air mass, the sensor head temperature, or leakage, degradation, or looseness of the optical





interference filter. Section 3.1 addresses the quality control procedure with respect to the instrument temperature
dependence. Some instrument anomalies dependent on the optical air mass include deviations of the measurement
time to the true time (i.e., time shift) and obstruction of light into the silicon or InGaAs detector (e.g,. dust, moisture,
spider webs). Measurements performed at high latitudes have a slowly varying optical air mass and thus optical air
mass pattern recognition is more difficult. The AOD spectra may have optical air mass dependence for out of band
leakage or degradation of transmittance due to irregularities in the optical filter composition or the AOD may have
significant variability due to a loose filter inside the sensor head.

The retained spectral AOD measurements passing the quality controls from Sect. 3.1 and Sect. 3.2 are evaluated as
input for the quality controls in the present section. The removal of nearly all of the clouds and most instrument
anomalies from the previous steps allow for more defined pattern recognition. This section will discuss the pattern
recognition techniques utilized for the time shift and AOD diurnal dependence, provide a description of the detector
consistency, and AOD spectral dependence quality controls. Further, the AOD diurnal dependence algorithm can be
used jointly with the detector consistency and AOD spectral dependence quality controls to remove anomalous data
with more certainty. These quality controls can be applied for multiple days to remove data impacted by anomalies
for more than one day even when clouds interrupt the day-to-day AOD pattern. The final data set is evaluated for
the remaining number of observations in a day and deployment period.

### 3.3.1 Time Shift Screening

AERONET data are transferred by satellite Data Collection Platform (DCP), PC, or SIM card data transfer. The
older Vitel satellite transmitters provided a handshake between the instrument and transmitter allowing for time
adjustment and newer Sutron Satlink transmitters provide a GPS time stamp to each message. While time shift is
not an issue for satellite transmissions, the time shift can become more significant for PC data transfer and even
some instruments using SIM card data transfer. AERONET has developed a program called cimel_https_connect
that can update the processing unit clock of Cimel Model 5 instruments. Older instruments (Model 4) and old non-
AERONET data transfer software (e.g., Cimel ASTPwin) do not have the capability to synchronize the Cimel
control box with the time-synced AERONET server. Most non-AERONET software requires the PC time to be
updated from a timeserver or GPS system to provide accurate clock synchronization. Even some newer Model T
instruments transferring data by PC or SIM can have faulty GPS modules in which the clock deviated significantly.
Cimel Model T instruments may allow for the PC software (e.g., cimelTS_https_connect) updating the time and
overriding the GPS module.

A Cimel clock that deviates from true time can result in an optical air mass calculation not appropriate for the actual
time especially when the optical air mass varies relatively rapidly diurnally. This instrument anomaly can result in
significant changes in the AOD, which affects all wavelengths but most greatly shorter wavelengths (e.g., 340nm,
380nm, and 440nm) at large optical air mass when it changes rapidly. In general, longer wavelength AODs (675nm,
870nm, and 1020nm) have less impact from erroneous optical air mass calculations due to less influence of



molecular (Rayleigh) scattering.  As a result, AODs from the longer wavelengths tend to be more stable and AODs
from the shorter wavelengths will tend to crossover the longer wavelengths only at one end of the day (near sunrise
or near sunset).  The timing of the wavelength crossover depends on whether the Cimel clock is too fast or too slow
with respect to the actual time.  For example, if the time is slow (fast) relative to the actual time, the temporally
deviated optical air mass magnitude will be larger (smaller) than the actual optical air mass and thus the short
wavelength AODs will be lower (higher) and possibly cross the longer wavelength AODs (significantly increase
spectral dependence).  In general, Cimel clock temporal deviations in AOD data can be identified using the
following:

1.        When the shortest available wavelength AOD crosses neighboring UV, visible, and NIR channel

AODs near sunset and the short wavelength AOD is decreasing significantly relative a longer stable

wavelength (e.g., 870nm) AOD, this condition indicates the Cimel clock is too fast (Fig. 9a).

2.        When the shortest available wavelength AOD crosses neighboring UV, visible and NIR channel

AODs near sunrise and the short wavelength AOD is increasing significantly relative to a longer stable

wavelength (e.g., 870nm) AOD, this condition indicates the Cimel clock is too slow (Fig. 9b).

The AOD differences and trends are used for a specific optical air mass interval (2.5–7.0), where the temporal clock
deviation amplifies the error in optical air mass calculations.  Individual day screening is limited to mainly cloud
free periods with low AOD in areas with significant variation in optical air mass from ~1.0–7.0.
The time shift algorithm is applied over a multi-day period.  The algorithm scans the current day plus 19 days in the
past (~3 week period) to determine if three or more days indicate the occurrence of a time shift.  If the multi-day
time shift criteria of three or more days are met, then data between the current day and the last occurrence of the
time shift are removed from the field deployment.  Although the Cimel clock could possibly be adjusted
periodically, most time shift issues tend to occur at remote sites and this approach will maximize the removal of data
over the multi-day period to minimize the negative impact on the data from the clock-shifted anomalies.  Moderate
to high aerosol loading can partly mask the temporal AOD time shift pattern and these data periods may not be
removed completely unless they occur between periods of lower aerosol loading when the clock shift spectral AOD
pattern is more defined.
**3.3.2    Detector Consistency Quality Control**
The instrument external collimator on the sensor head avoids stray light and reduces front lens contamination, while
the internal sensor head defines the field of view of the instrument (nominally 1.2°) by the achromatic front lens,
filter, and field stop before each detector.  The external collimator is composed of two tubes and the aperture design
varies slightly by instrument type.  The Cimel Model 4 instrument type has two Silicon photodiode detectors in the
sensor head to measure the Sun and sky while newer model instruments have one Silicon photodiode and one
InGaAs photodiode detector to measure the Sun and sky on both detectors.  One of the detectors could be impacted



by an obstruction such as a spider web, insect debris, or moisture. For Cimel Model 4 and some Model 5
instruments, the sky scan scenario performs two measurements at the 6° azimuth angle for the almucantar and 6°
scattering angle for the principal plane at each wavelength over both detectors. For these older instruments, the
solar aureole gain is used for the solar Silicon diode detector and the sky gain is used for the sky Silicon diode
detector. These redundant measurements can allow for detection of the change in the relative signal but this method
is currently more appropriate to use for quality controlling the inversion products due to uncertainty in sky
calibration. Newer Model 5 and Model T instruments (with the solar and sky measurements performed on both
detectors) do not have the redundant sky measurement; instead, these instruments have a redundant solar
measurement at 1020nm in both collimator tubes, where each solar measurement of the triplet is performed within
eight seconds of each other. The AOD 1020nm measurements on Silicon and InGaAs detectors can be compared
directly to determine if an obstruction exists in front of either of the detectors. Applying a similar approach to Giles
et al. (2012), the difference limit ($\Delta\tau_{Limit}$) can be computed using the optical air mass and AOD magnitude dependent
formulation (Eq. (12)):

$$\Delta\tau_{Limit} = \frac{(0.04 + (0.02 * MIN[\tau_{1020nm}]))}{m} \tag{12}$$


where $MIN[\tau 1020nm]$ is the minimum of the AOD at 1020nm obtained from the redundant AOD 1020nm
measurements on Silicon and InGaAs detectors and $m$ is the optical air mass. The difference limit for an AOD
1020nm minimum of 1.0 will result in the 0.06/m 1020nm difference limit described in Giles et al. (2012). A more
lenient approach is used here based on the AOD magnitude to prevent removal of data for low AOD at 1020nm. At
low AOD, the average field instrument uncertainty (up to 0.01) becomes more significant while the maximum AOD
error occurs at midday and differences due to their temperature dependency can contribute up to 0.02 AOD bias.
Given the relative difference in the AOD 1020nm measurements, the maximum uncertainties in both 1020nm
measurements must be considered. Therefore, the 0.02 threshold is derived from the average uncertainty (up to
0.01) and the 0.04 limit is derived from the maximum midday error in AOD and temperature dependency (up to
0.02). When more than 10% of the total measurements for the day exceed the $\Delta\tau_{Limit}$, data are removed in the
following manner:
1.   If the AOD 1020nm Silicon subtracted by the AOD 1020nm InGaAs detector is greater than $\Delta\tau_{Limit}$, then the

Silicon side has an obstruction and the entire measurement is removed for both Silicon and InGaAs AOD

data.

2.   If AOD 1020nm Silicon subtracted by the AOD 1020nm InGaAs is less than $-\Delta\tau_{Limit}$, then the InGaAs

detector has an obstruction and only the InGaAs AOD for 1020nm and 1640nm measurements are

removed.

3.   If the redundant AOD 1020nm values are nearly the same ($-\Delta\tau_{Limit} \geq \Delta\tau \geq \Delta\tau_{Limit}$), then an obstruction could

possibly exist in the event that a substance (e.g., spider webs, dust, moisture) similarly obstruct both

detectors.

For condition (3), this case is further evaluated by the AOD diurnal dependence quality control in the next section.



### 3.3.3    Aerosol Optical Depth Diurnal Dependence

The AERONET instrument has spectral calibrations made and typically applied both before and after field deployment. When the instrument operates in the field, the pre-field spectral calibration applied to the near real-time data is constant.  If the calibration changes significantly during the instrument deployment, the error in the computation of the AOD increases with decreasing optical air mass where the maximum error occurs when optical air mass approaches one ($\delta\tau*m$; Hamonou et al., 1999).  As a result, an apparent diurnal dependence in the AOD can occur depending on the magnitude of the deviation from the pre-field calibration.  When both the pre-field and post-field calibrations are applied and data still show a diurnal dependence in the AOD, then the deviation in the field measurements is due to a non-linear change in the calibration coefficient since Level 2.0 data utilize a linear interpolation between the pre-field and post-field calibration coefficients.

Midday maximum (concave pattern) or midday minimum (convex pattern) of AOD diurnal dependence can be observed at any AOD magnitude but are typically more pronounced at lower aerosol loading due to calibration offset (Cachorro et al., 2004) or instrument anomalies.  Quality controls developed for the analysis of the AOD diurnal dependence need to consider the impact of clouds and missing data to assess whether to remove these data while minimizing the removal of data exhibiting true diurnal dependence.  For example, one cloud-free day may show diurnal dependence, but on another day, the morning or afternoon data may not be available due to missing data during cloudy or rainy periods.  The algorithm must have a sufficient number of observations to perform a robust assessment of the AOD diurnal dependence.

Variation in the number of available measurements in a day due to clouds or instrument issues can limit the application of a single day only approach.  As a result, the morning and afternoon periods must have at least five measurements separately and the analysis of the full day must have at least 10 measurements.  To analyze the diurnal dependence and reduce the impact of outliers, the GNU Scientific Library robust least squares (RLS) linear regression fit is performed for AOD versus the inverse optical air mass ($m^{-1}$, where $m$ is approximately the cosine of the solar zenith angle).  The slope and correlation coefficient ($R$) values derived from the linear fit are used as thresholds to determine the magnitude and strength of the diurnal dependence (Table 4).

The nominal AERONET 440nm, 675nm, 870nm, and 1020nm wavelengths for the Silicon detector and 1640nm for the InGaAs detector are assessed for diurnal dependence and potential removal of all spectral channels. An example of the AOD diurnal dependence of 1020nm wavelength is shown in Fig. 10 at the Rio Branco (9.96° S, 67.87° W) AERONET site where the site manager indicated spider webs were obstructing measurements.  If data are removed for the InGaAs detector, then only InGaAs detector data are removed, while removal of the Silicon detector data will remove all data including InGaAs detector data, if any.  The AOD diurnal dependence is classified as two categories: independent and dependent.  If the algorithm meets the strict thresholds for "independent" diurnal dependence, then all channels exhibiting diurnal dependence can remove data for a day, except the 1020nm channel since some old data with temperature defaults may exhibit false diurnal dependence.  Otherwise, all of the above



channels are used for the "dependent" diurnal dependence quality control. The dependent diurnal quality control
relies on more lenient thresholds for the slope and $R$; however, the removal of data generally requires that another
quality control flag is set such as the detector consistency quality control (Sect. 3.3.2), where an obstruction was
identified in front of one of the detectors or at least one additional qualified wavelength meeting the slope and $R$
thresholds. When a qualified wavelength indicates dependent AOD diurnal dependence for Day or both AM and
PM and AM and PM slopes are positive, then the entire day can qualify for independent removal. This methodology
allows for a more skilled approach in removing only data affected by instrumental anomalies while minimizing the
removal of data coincidently producing a true diurnal dependence signature.

The AOD diurnal dependence identification can be complicated by changes in aerosol loading during the day, cloud
artifacts, and missing data. A multi-day scan must be performed to maximize the removal of data impacted by
instrument anomalies. A multi-day assessment example is provided in Fig. 11 for Rio Branco. Figure 11a shows
that the spectral AOD varies significantly diurnally for the period from 26 August to 5 September 2011, especially
for the 870nm and 1020nm near infrared wavelengths. Figure 11b shows evaluation of the slope and correlation
coefficient (R) for the AOD 1020nm daily variation, which shows 7 of the 10 days exceeding the thresholds (slope >
0.1 and R>0.94) and wavelengths established in Table 4. For these data to qualify for dependent AOD diurnal
dependence removal, additional information is needed such as another qualified wavelength with slope and R
exceeding the thresholds. For this case, the AOD 870nm daily slope and correlation parameters (not shown) also
exceed the thresholds, which lead to the elimination of these data from Levels 1.5 and 2.0. Similar to the time shift
screening in Sect. 3.3.1, the AOD diurnal dependence algorithm scans the last 19 days including the current day to
determine the first occurrence and last occurrence of the dependent and independent AOD diurnal dependence.
When three or more days are identified, data are removed from the first occurrence to the last occurrence of AOD
diurnal dependence during the 20-day period. The multi-day screening allows for the elimination of data affected by
an obstruction in the instrument field of view even with moderately high aerosol loading in the NIR wavelengths
and when days with incomplete number of measurements from the established protocol due to clouds.

### 3.3.4   Reverse Spectral Dependence

While the majority of the cloud screening quality controls remove aerosol measurements contaminated by clouds,
some spurious points or slowly varying changes in cloud properties may still affect the data set at this point in the
algorithm. A new method (Fig. 12) utilizing the Ångstrom exponent (AE) is applied to the remaining data set for
evaluation of cloud contamination. Ångstrom exponents derived from anomalous AOD measurements due to
instrument artifacts may produce a similar signature. The spectral dependence among the wavelengths is now much
improved compared to Version 2 by removing temperature dependencies that influenced the calculation of the AE at
low AODs reducing the effect of improper spectral dependence due to temperature anomalies.

The AE is computed utilizing the ordinary least squares fit of the logarithms of AOD and wavelength for the ranges
of 440–870nm, 870–1640nm (if 1640nm is available), and the 870–1020nm (for Silicon detectors only) range (Eck





et al., 1999). The reverse spectral dependence algorithm in Fig. 12 removes cloud contaminated points utilizing
these AE ranges depending on the instrument model.
Figure 13 shows the removal of the anomalously high AOD at the Bratts Lake (50.20° N, 104.71° W) AERONET
site in southwest Canada. In Fig. 13b, all negative and a few positive AE values are identified and the algorithm
removes nearly all of the residual cloud contamination in this case. However, the penultimate and final
measurements in Fig. 13c have slightly higher AOD than the previous hour of data, which may be due to marginal
contamination by optically thin cirrus clouds. Additional algorithm development is still needed to further enhance
the removal cloud contaminated data with small ice crystals while not removing dust aerosols.

### 3.3.5    Aerosol Optical Depth Spectral Dependence

The wavelength dependence of AOD typically is strong for fine mode aerosols (e.g., pollution or smoke) and weak
for coarse mode aerosols (e.g., dust or sea salt). The AE provides an index of the strength of the spectral
dependence related to the estimation of the possible aerosol size (Eck et al., 1999). In general, the $AE_{440–870nm}$ will
typically provide values between approximately 0.0 and 3.0. These prospective values indicate no spectral
dependence at $AE_{440–870nm}$ of 0.0 and very strong spectral dependence with an $AE_{440–870nm}$ near 3.0 (AE values of 3.0
have not been observed in good quality data with sufficiently high AOD). The spectral dependence can be used to
evaluate the quality of each channel given that most channels in the measurement suite adhere to the stated AOD
uncertainty of 0.01 for wavelengths ≥400nm and 0.02 for wavelengths <400nm (Eck et al., 1999). The fit of the
AOD with wavelength on logarithmic scale should generally be linear for coarse mode dominated or fine/coarse
mode particle mixtures. However, in moderate to high aerosol loading cases (especially when fine mode
dominated), a quadratic or cubic assumption is needed to fit the data depending on the wavelength range under
evaluation (Eck et al., 1999; O'Neill et al., 2008). The ordinary least squares (OLS) methodology is perturbed by
the presence of outliers and therefore skews the fit towards outliers. If the boundary wavelengths are impacted by
anomalies, the ordinary least squares can poorly fit other intermediate wavelengths.

In an effort to reduce the influence of outliers, the GNU Scientific Library (GSL Version 2.2.1 C compilation)
robust least squares (RLS) technique is utilized to improve the removal of spectral AOD outliers. In general, the
OLS technique is sensitive to the endpoints and to the number of points used in the regression. For example, the
outlier detection will have less skill with a few points or anomalous endpoints. The RLS scheme uses an iterative
approach with up to 100 passes using the Tukey biweight function and assigning the outliers a lower weight with
each pass. The RLS approach allows for the more meticulous removal of wavelengths out of spectral dependence
and more importantly preserves mid-visible wavelengths that could be removed incorrectly when utilizing the
ordinary least squares method.

Outlier detection is performed utilizing the uncertainty of the AOD measurement and providing an allowable
tolerance in the fit given potential irregular nature of the uncertainty (0.01 to 0.02). For wavelengths ≥400nm and
<1600nm, the allowable AOD difference between the measurements and fit for a candidate wavelength is



(0.02*AOD)+0.02, based on the stated AOD uncertainty for these wavelengths (Holben et al., 1998; Eck et al.,
1999). For wavelengths <400nm and 1640nm, the allowable AOD difference between the measurements and fit for
a candidate wavelength is (0.02*AOD)+0.04, which is adjusted for greater uncertainty at the UV wavelengths and
greater uncertainty in the larger spectral range to fit the 1640nm wavelength.

The spectral outlier procedure begins by identifying and removing any negative AOD values that are not within the
allowable AOD difference from the RLS linear fit. Negative AOD due to slight calibration drift can be observed at
very clean locations; otherwise, these negative values may be anomalous. The algorithm will evaluate each
wavelength separately and compute the RLS linear fit based on the remaining wavelengths producing the slope,
intercept, and $R^2$ values, where the slope and intercept are used to compute the AOD fit at the wavelength under
evaluation. If the algorithm does not identify any wavelengths for removal, then the procedure is complete. If AOD
is low ($AOD_{440nm}$<0.1) and one wavelength AOD exceeds the maximum allowable difference, then the wavelength
will be removed due to the linear fit deviation. However, if more than one wavelength has AOD marked for removal
for the low AOD condition, then the wavelength with the largest departure from the linear fit to the measurement
and largest $R^2$ will qualify for removal.

In the case of higher AOD ($AOD_{440nm}$≥0.1), the algorithm stores the information from the RLS linear fit and
continues to perform a RLS quadratic fit (400nm≤λ≤1020nm) or a RLS cubic fit (λ =1640nm). If the candidate
wavelength deviates from the allowable difference in fit to the measurements for the higher order fits, then the
wavelength will be removed if it is identified as a wavelength that corresponds to the maximum deviation for the
RLS linear fit. Figure 14 provides an example of this condition at the Osaka (34.65° N, 135.59° E) AERONET site.
After each wavelength removal regardless of order of the fit, the algorithm repeats until no wavelength removals
occur or when less than three wavelengths remain.
**3.3.6    Large Aerosol Optical Depth Triplet Variability**
In addition to growth of hygroscopic aerosols near cumulus cloud boundaries and large triplet variability at short
wavelengths in highly variable fine mode plumes, a misaligned filter due to improper filter wheel movement or dust
on the filter may produce large AOD triplet variability (AOD Max – AOD Min). The cloud screening triplet
variability quality control removes the entire measurement when 675nm, 870nm, and 1020nm AOD triplets have
large triplet variability exceeding the threshold (0.01 or 0.015 * AOD, whichever is greater). A situation may exist
where one of those wavelengths or shorter wavelengths are impacted by a filter anomaly making it necessary to
assess the large AOD triplet variability. If the triplet measurement is identified for high AOD retention (Sect. 3.1.6),
then the following large adjacent triplet quality control is not performed because very high aerosol loading in fine
mode events can lead to large triplet variability naturally. Occasionally, if the triplet is very large and exceeds the
limit of 0.03+0.2*AOD, then the wavelength is removed independently of the next longer wavelength.



To further screen anomalous triplets individually or the entire day, each triplet and wavelength is evaluated using the
triplet variability from the shortest wavelength (e.g., 340nm) and the next longer wavelength (e.g., 380nm).  The
allowable triplet variability limit is computed based on the aerosol loading and the AOD triplet variability of the
next longer wavelength: 0.03+0.02*AOD+triplet_variability_of_next_longer_wave. If the total number of triplets
for a wavelength exceeding the large triplet variability threshold is more than 25%, then the AOD measurements for
the wavelength are removed completely for the entire day.  Figure 15 shows the large triplet variability removal at
the PEARL (80.05° N, 86.42° W) AERONET site in northern Canada.  The triplets at shorter wavelengths may
naturally exhibit relatively large triplet variability hence it is necessary to check the shorter wavelength in
comparison to the next longer wavelength which typically will be more stable if clouds do not impact the
measurements.
**3.3.7    Remaining Measurements Evaluation**
After the previous quality control algorithms have been applied, extraneous data points may remain and are
identified for possible removal.  A number of conditions have been implemented based on the total data removed for
the day, number of wavelengths remaining for the day, and number of measurements for a wavelength for a
deployment.  These "cleanup" conditions below will remove all wavelengths in a day for any of the following
conditions dependent on the "retain high AOD" from Sect. 3.1.6 and the number of wavelengths in a day:

1.   If retain high AOD and less than two wavelengths remain in a day

2.   If retain high AOD and two wavelengths but are not 870nm and 1020nm in a day

3.   If not retain high AOD and less than three wavelengths remain in a day

4.   If not retain high AOD and less than half of the wavelengths remain in a day


Each wavelength must be evaluated for remnant data artifacts. If greater than 50% of the total cloud screened AOD
data for a wavelength in a day are removed, then AOD measurements for the candidate wavelength will be removed
for the day.  Further, a condition is implemented to remove specific wavelengths for an entire deployment.  For
example, if the number of measurements for a wavelength is less than 20% of the total cloud screened data set for a
deployment, then all of the measurements for the specified wavelength will be removed for the deployment.  These
removal conditions are necessary to fully quality control the spectral AOD data set and avoid unphysically irregular
and fragmented data sets.
**3.4    Algorithm Performance Assessment**
Data quality controls applied to the quality controlled Level 1.0 data set are evaluated for removal performance for
each part of the Level 1.0 prescreening and Level 1.5 algorithm.  The Level 1.0 prescreening is applied to about 84
million solar triplet measurements from 1993–2018.  The radiometric sensitivity screening (see Sect. 3.1.2) for the
DN of 1020nm removes about 36% and the digital voltage triplet variance greater than 0.16 (see Sect. 3.1.3)
removes nearly 11% of the Level 1.0 data.  The remaining Level 1.0 prescreening that check for radiometric
sensitivity screening for DN of 870nm, extreme temperatures ($T_S \leq -40°C$ or $T_S > 100°C$), and bad measurement





configuration conditions remove approximately 0.5% of the Level 1.0 data. Therefore, nearly half (48%) of the
initial 84 million solar triplet measurements are removed by the Level 1.0 prescreening steps due to the presence of
clouds in the solar measurements that greatly reduce the signal (e.g., stratus clouds) or exhibit significant temporal
variability within the one minute triplet measurement sequence (e.g., cumulus clouds).

The Level 1.5 quality control algorithm is divided into the two main steps for cloud screening and instrument data
anomaly removal.
Figure 16 shows the percentage of the Level 1.0 data removed by the Level 1.5 cloud screening quality control.
Over 23% of the removal in the cloud screening algorithm was due to the large triplets at the long wavelengths
(675nm, 870nm and 1020nm). Nearly 5% of the removal of the Level 1.0 data was due to the presence of cirrus
clouds as detected by the solar aureole curvature algorithm and is significant since a cirrus contamination bias is
evident in the AOD in Version 2 Level 2.0 data set. The "Unqualified" category indicates data that are negative
AOD or lack the sufficient channels to participate in the cloud screening part of the algorithm and these
measurements are rejected from Level 1.5. After all of the data are cloud screened, about 66% of the Level 1.0 data
are passed to the second part of the Level 1.5 instrument quality control algorithm for examination of the instrument
anomalies and other spurious clouds and artifacts.

The second stage of the Level 1.5 quality control algorithm utilizes measurements passed from the cloud screening
algorithm. While the cloud screening algorithm rejects the entire measurement in the presence of clouds, the
instrument quality controls can also reject the entire measurement or remove data by wavelength depending on the
anomalous condition.
Figure 17 shows the removal of Level 1.5 cloud screened data due to mainly instrument anomalies for each
wavelength. More than 2.5% of the data are removed due to the AOD diurnal dependence screening, about 2% for
the time shift screening, and 1.5% for the AOD 1020nm difference screening. These three instrument quality
control algorithms remove in general the most across all wavelengths. Some removal occurs significantly spectrally
for the InGaAs channel (1640nm). The InGaAs channels can be affected in some instruments more significantly by
water contamination as the InGaAs side of the collimator is facing away from the Sun when in the parked or resting
position. Further, when the algorithm removes all of the Silicon channels, the remaining InGaAs channels are also
removed since no other independent method exists to check the InGaAs channel data quality. The "Remaining"
measurements removal shows that nearly 4% of the cloud screened data are removed from the InGaAs data set. The
AOD spectral dependence removes more than 2% of the 340nm wavelength data, which tends to be the most
unstable wavelength (due to filter degradation), and about 0.5% for all other wavelengths. The temperature
screening removal of missing or anomalous temperatures mostly affects the Silicon 1020nm wavelength with nearly
1% of the cloud-screened data removed due to its large temperature dependence compared to the other wavelengths.





## 4    Assessment of the Quality Assurance Data Set

The aerosol optical depth (AOD) data will be qualified for consideration of Level 2.0 once it passes the Level 1.5 checks. To reach Level 2.0, these data must meet the following conditions:

1. Data must have pre-field and post-field calibration applied; or in some cases, the pre-field deployment or post-field deployment calibration may be made constant for the deployment after evaluation of the best calibration values.

2. Temperature characterization must be applied utilizing the temperature correction for the instrument or default values for each wavelength.

3. Instrument must be designated as the primary instrument for the site.

Once the above conditions are met, these data are considered to reach Level 2.0.  These Level 2.0 data are recommended for publication and use in various atmospheric applications.  The automated algorithm attempts to preserve aerosol data while removing data artifacts.  Some unusual atmospheric conditions (e.g., small cirrus particles r<5µm) or rare instrument anomalies (e.g., loose filters or partially removed multi-da AOD diurnal dependence) affecting the AOD may rarely pass through the algorithm and users are advised to consider inspecting these data carefully when using them for detailed studies. Further, optical air mass dependent anomalies such as the time shift and AOD diurnal dependence quality controls may allow data to pass when aerosol loading is high or too few data exist to make an assessment.  These quality controls can determine patterns more skillfully at lower aerosol loading which could result in retaining potentially contaminated high aerosol loading periods when the pattern may be less defined and does not meet the quality control thresholds.

The subsequent sections discuss the impact of the temperature characterization on the Version 3 Level 2.0 AOD data to quantify the change in regards to the Version 2 Level 2.0 data set.  Further, the assessment of the Version 3 near real-time product is made to determine the average bias of the AOD based on the applied calibration.  Finally, an analysis is made of the Version 3 Level 2.0 AOD long-term averages for select AERONET sites and these are compared to the Version 2 Level 2.0 AOD long-term averages.

### 4.1    Temperature Characterization Evaluation

The accurate measurement of the spectral direct-beam Sun intensity (from which AOD is computed) depends on the sensor head temperature of the instrument as discussed in Sect. 2.  The sensor head temperature can vary significantly since the optical head canister is heated by the Sun and can be much higher (>10°C) than the ambient temperature especially near solar noon.  The temperature sensitivity of the Silicon detector is more significant for the 1020nm filter due to the proximity to the edge of the spectral range of the detector in which temperature dependence becomes more significant.  The temperature dependence for all wavelengths may vary due to the composition and/or manufacturing quality of the filters and/or detectors.  Due to technical difficulty, the ultraviolet wavelength (λ<400nm) filters have not been temperature characterized in Version 3; however, UV filters may have



a temperature dependence. Figure 18 shows the difference in the AOD temperature coefficients for Version 3 temperature correction applied to Version 3 data and Version 2 temperature correction applied to Version 3 AOD data from 1993–2018. The AOD varies most significantly for the Silicon 1020nm channel with a full range of ~0.02 for sensor head temperatures between −25°C and +55°C. Notably, the shorter wavelength channels and the InGaAs wavelengths (i.e., 1020nm and 1640nm) do not show significant change in AOD less than 40°C. All of the wavelengths, except the Silicon 1020nm, show an AOD difference decrease from −0.005 to −0.010 for temperatures greater than 40°C, which may be due to changes in instrument characteristics (e.g., electronic instability in the instrument) at high temperatures. The decreasing AOD difference with increasing temperature may be related to the smaller number of observations at high temperatures and contribution by instruments with temperature characterization measurements that did not reach temperatures greater than 40°C. Temperature characterization has proven to be small yet necessary adjustment to the AOD computation and this improvement is especially exhibited in arctic regions or sites with very low aerosol loading in which the Version 3 AOD spectra have much less crossover allowing for the computation of more accurate Ångstrom exponents than in the Version 2 data set.

## 4.2    Level 1.5 Near Real-time Aerosol Optical Depth Bias and Uncertainty

The Version 3 near real-time data set provides improved data quality compared to Version 2 since the algorithm has improved cloud screening and instrument quality controls applied to the data. The data set can vary in the near real-time interval from current day up to one month as ancillary data sets are received and processed, hence, these database changes invoke reprocessing of the AOD throughout the near real-time phase. Once AOD data have been pre-field and post-field calibrated, then these data may be raised to Level 2.0 as described in Sect. 4. The near real-time data using only constant pre-field calibration is compared to the quality assured data set that uses both the pre-field and post-field calibrations applied to the data with the assumption of linear interpolation. Figure 19 shows the distribution by wavelength for this comparison of the near real-time and quality assured data set for the entire database of Level 2.0 qualified data excluding calibration site data and deployments using a copied pre-field or post-field calibration. These results are based on the Version 3 Level 2.0 data set in which the Level 1.5 algorithm scans the entire deployment. The AOD difference histograms were computed for optical air mass ranges ($1.0 \leq m < 7.0$ and $1.0 \leq m < 1.5$). The optical air mass $1.0 \leq m < 7.0$ range includes all of the data; however, these AOD difference magnitudes will be constrained by the improved AOD measurements at large optical air mass and influenced toward Northern hemisphere winter mid-latitude sites when AOD tends to be low. The optical air mass $1.0 \leq m < 1.5$ range includes data will provide AOD measurements near solar noon and these measurements are generally less accurate ($\delta\tau * m$) than at larger optical air mass. In addition, optical air mass $1.0 \leq m < 1.5$ range data include a greater influence of tropical locations and data from the mid-latitude summer when AOD tends to be moderate to high.

Figure 19 shows the AOD average differences for the $1.0 \leq m < 7.0$ range indicate a positive bias in which the AOD for the pre-field only calibration tends to be on average +0.003 to +0.009 higher than the AOD using the interpolated calibration. Similarly, AOD average differences for the $1.0 \leq m < 1.5$ range show a positive bias and similar wavelength variations but up to two times larger differences than for the $1.0 \leq m < 7.0$ range. The largest



average differences and standard deviations are for the UV wavelengths, which have greater uncertainty as
discussed in Sect. 2. The AOD differences for the wavelengths longer than 500nm have about less than half the bias
of the UV wavelengths. The Level 1.5 algorithm performance improves with increased data availability such as a
greater number of wavelength or number of days. When an instrument deployment begins, some of the Level 1.5
algorithm steps such as multi-day removal schemes are not available until several days into the deployment
producing larger differences in the near real-time AOD with respect to the final product. While wavelength
dependent biases of +0.003 to +0.009 for the $1.0 \leq m < 7.0$ range and +0.006 to +0.015 for the $1.0 \leq m < 1.5$ range exist
when only the pre-field calibration is applied, the difference can vary significantly depending on each instrument
deployment necessitating continued post-field calibration and maintenance effort.

When an instrument is deployed in the field, the pre-field calibration is used constantly until the post-field
calibration is assessed and applied to the data using linear interpolation. The difference of pre-field calibration AOD
minus the post-field calibration AOD average difference and standard deviation are computed in day bins for the
number of days since the pre-field calibration. Figure 20 shows the AOD 500nm average difference for the optical
air mass ranges: $1.0 \leq m < 7.0$ and $1.0 \leq m < 1.5$. Instruments typically operate in the field between 12 and 18 months
from the pre-field calibration date; however, the instrument deployment may be delayed and the instrument may not
begin operation for a few months after the pre-field calibration. Thus, the number of AOD measurements in the
days since pre-field calibration bins increase to a maximum at about 100 days. Some instruments may operate longer
in the field to support field campaigns and other scientific priorities. Figure 20 shows that the AOD average
difference and the standard deviation slowly but steadily increase for each optical air mass range. At about 1.5 years
after pre-field calibration (~550 days), the AOD average difference is about +0.010 with a standard deviation of
0.015 for optical air mass $1.0 \leq m < 7.0$ range and +0.017 with a standard deviation of 0.021 for $1.0 \leq m < 1.5$. For the
UV wavelengths, the average differences and standard deviations tend to increase slightly while the longer visible
and near infrared wavelengths tend to decrease slightly. Therefore, the quality of the Level 1.5 near real-time AOD
changes with time with high quality data at the start of the deployment but up to a +0.02 bias and 0.02 uncertainty
for data collected more than 1.5 years since pre-field calibration.

**4.3    Multi-year Monthly Comparisons of Version 3 Level 2.0 to Version 2 Level 2.0 Databases**
Long-term average differences between the Version 3 and Version 2 Level 2.0 data sets provide insight into the
changes to be expected across most AERONET sites. The analysis of the Version 3 and Version 2 data sets shows
mainly the differences in the AOD, $AE_{440-870nm}$, precipitable water (PW) in cm, and the number of days are clustered
near zero (Fig. 21). Note that precipitable water data quality depends on the quality of the input wavelengths
(675nm and 870nm) and no further quality control is made on the 935nm wavelength. The increases in the Version 3
Level 2.0 multi-year monthly average AOD are often due to the increased presence of fine mode particles from high
aerosol loading events as well as aerosols in near cloud environments (Eck et al., 2018). The decrease in the multi-
year monthly average AOD is due to the improved removal of clouds in the Version 3 quality control algorithm.



Generally, the results should be very similar between Version 3 and Version 2 in AOD calculation since the temperature characterizations as well as NO2 absorption contributions typically have relatively minor contributions.

Other factors affecting the AOD calculation include the adjustment of site coordinates and elevation information for about 100 AERONET sites utilizing GPS or digital elevation model. A few rare extreme coordinate adjustments of more than 25 km included Petrolina_SONDA (9.0691° S, 40.3201° W), Ilorin (8.4841° N, 4.6745° E), and Ouagadougou (12.4241° N, 1.4872° W). A large site coordinate adjustment can complicate satellite matchups for these few cases but the review of all AERONET sites showed that less than a 5 km distance adjustment and less than 100-meter elevation adjustment was needed for most of these 100 suspected sites.

Figure 22 shows similar plots to Fig. 21 except that the observations used for the multi-year monthly averages in both data sets the instantaneous observations are time matched, hence, each data set has the same number of observations and number of days. The time matched long-term average comparison provides insight into the AOD calculation differences rather than impacts due to cloud screening and instrument quality controls applied in Level 1.5. Table 5 shows the multi-year monthly overall standard deviation and AOD maximum to minimum range is significantly reduced compared to the data set without time-matched observations. Figure 22a shows a slight decreasing trend of Version 3 AOD for increasing Version 2 AOD and most of the larger AOD deviations are for sites in Asia where the impact of the OMI NO2 corrections may be contributing to the slight shift of up to 0.02 for a few months and sites.

For unmatched or time matched data sets in Table 5, the precipitable water climatology changed on average insignificantly. The multi-year monthly overall days difference (Table 5) for the unmatched precipitable water data set was near zero and the standard deviation was near 25 days while the maximum of +150 and minimum of −130 days indicate significant variability due to the differences in quality controls between the algorithms. Overall, the changes from Version 2 to Version 3 in precipitable water are generally negligible in terms of the contribution to the calculation of the AOD.

Overall, the multi-year monthly overall average difference between Version 3 and Version 2 for unmatched data is +0.002 and time matched data is −0.002 indicating remarkable consistency between the long-term average quality assured data sets. For example, the NASA GSFC AERONET site multi-year monthly average (Fig. 23) located 20 km north of Washington, D.C., shows minor variations in the AOD and increase in AE due to removal of cirrus clouds during the winter months and increasing AOD in the summer months due to the greater abundance of cloud processed or near cloud aerosols (Eck et al., 2014).

Comparison of $AE_{440-870nm}$ in Fig. 21b and Fig. 22b show significantly lower values for Version 3 than Version 2 Level 2.0 at low optical depth. An analysis of long-term average data at Lulin, Taiwan (23.47° N, 120.87° E) identified significant reduction of Version 3 AE relative to Version 2 AE at very low AOD due to temperature



characterization that resulted in improved AOD spectral dependence (Fig. 24). The Lulin site is a high altitude mountain station located in south central Taiwan, and this site is affected episodically by trans-boundary aerosol plumes from East and Southeast Asia (Lin et al., 2013; Wang et al., 2013). In eastern China, multi-year monthly averages from the XiangHe site (39.75° N, 116.96° E) show a significant Version 3 AOD increase of 0.2, while maintaining nearly the same AE and increasing the number of days up to near 40% for the multi-year monthly average in July and August (Fig. 25). The XiangHe site is located to the east of Beijing and is routinely impacted by urban pollution and episodically by biomass burning and desert dust events (Li et al., 2007). The significant increase in the AOD for XiangHe is likely due to the retention of highly variable fine mode aerosol events particularly at very high AOD, which were removed by the Version 2 cloud screening wavelengths utilizing large triplets less than 675nm (Eck et al., 2018). Additionally, some very high AOD events at XiangHe were previously removed by the Version 2 mid-visible low signal threshold but are now retained in Version 3, but often only for wavelengths longer than 675nm, so the statistics for these days are not accounted for in the 500nm data shown in Fig. 25.

At the Mongu (15.25° S, 23.15° E) site (Fig. 26), the biomass burning smoke typically occurs during the dry season from April through November due to biomass fuel cooking and agricultural burning (Eck et al., 2003). Comparisons of multi-year monthly averages for the Mongu site shows small deviations for AOD up to ±0.01 with slight increases in Version 3 AE during December through March due to enhanced cirrus cloud removal from the solar aureole check. Notably, the number of days for the Mongu multi-year monthly averages significantly decreased by 10% to 25% in Version 3 due to improved cloud screening and sensor head temperature anomalies affecting instrument performance. In Cinzana, Mali (Fig. 27), the aerosol loading is dominated by background dust aerosol with episodic contributions to the aerosol loading from biomass burning smoke from November to March (Cavalieri et al., 2010). The AERONET IER-Cinzana site (13.28° N, 5.93° W) multi-year monthly averages show generally 0.03 lower AOD for Version 3 than Version 2 and nearly the same AE for both versions. The number of days for each month is 7% to 25% lower in Version 3 when compared to Version 2 mainly due to improved cirrus cloud screening.

## 5   Summary

The Aerosol Robotic Network (AERONET) has adopted a new automated quality assurance algorithm called Version 3. The significant impacts of the Version 3 algorithm are updated and improved cloud screening and quality control methods, which are powerful tools in quality assuring the Sun photometer AOD data. Comparisons between the quality assured data sets of Version 3 and Version 2 show excellent agreement. Deviations can be explained by known algorithm differences such as changes in the cloud screening triplet variability, cirrus cloud detection and removal, implementation of temperature characterization, updates to $NO_2$ climatology, modification of site coordinates and elevation, and identification of instrument anomalies such as aerosol optical depth (AOD) diurnal dependence, AOD spectral dependence, and instrument electrical and temperature stability.



Major highlights of this work include (not listed in priority):

1.  An automatic quality control algorithm significantly reduces the necessity of analysts to inspect millions of
AERONET measurements. The AERONET Version 3 algorithm applied in near real-time provides high
quality AOD for data assimilation applications. The Version 3 Level 2.0 data is provided within 30 days of
the post-field calibration evaluation after the instrument deployment, improving the timeliness of quality
assured data.

2.  Improvements to the total AERONET database cloud screening results in about 60% removal of clouds
from the complete Sun photometer database and this value is similar to the coverage of clouds globally of
about 68% (Rossow and Schiffer 1999). Autonomous Cimel Sun photometers can view gaps and nearby
regions of the clouds and become inactive during rain periods due to wet sensor activation and AERONET
sites are dominated by land locations which generally have lower cloud cover on average; therefore, these
factors would reduce the difference between total AERONET cloud removal percentage and global satellite
observations. Over 36% of the total data were removed by the 4-quadrant solar tracker sensitivity check
due to less accuracy in tracking the Sun in cloudy conditions, while about 23% of the removal was due to
the variability of clouds with respect to more homogeneous aerosol loading.

3.  Utilizing the shape of the solar aureole radiances with scattering angle, a cirrus detection algorithm was
developed by leveraging MPLNET LIDAR cloud detection capabilities. The solar aureole cirrus algorithm
eliminates ~5% of the Level 1.0 AOD data to reduce the bias of optically thin cirrus clouds in AERONET
database.

4.  Spectral temperature correction has been implemented for all AERONET instruments using the sensor head
temperature sensor reading. The temperature characterization shows significant AOD deviation ±0.01
variation between −25°C and +50°C for the Silicon 1020nm, since this wavelength is on the edge of the
Silicon detector sensitivity range. Other wavelengths in the 440nm to 1640nm range have weak
temperature dependence from −25°C and +30°C with a few wavelengths having greater temperature
dependence at higher temperatures.

5.  New automated instrument anomaly screening provides a systematic and objective scheme to remove entire
measurements or individual wavelengths from the AERONET AOD database. Importantly, obstructions to
the instrument optics are now removed automatically using an AOD diurnal dependence algorithm based
on the optical air mass. The AOD diurnal dependence technique employs several conditions that were
developed to mitigate the removal of true diurnal dependence conditions while maximizing the removal of
data significantly impacted by anomalies affecting the instrument optics.

6.  Bias and uncertainty estimates for near real-time AOD are computed by using the difference of the pre-
field calibration AOD minus the interpolated calibration AOD. The near-real time AERONET data have
an estimated bias up to +0.02 and one-sigma uncertainty up to 0.02; these values have slightly higher
uncertainty for shorter wavelengths and slightly lower uncertainty for longer wavelengths.





7.  The AERONET Version 3 and Version 2 AOD quality controlled databases are analyzed to have a long-term monthly average difference of +0.002 with ±0.02 standard deviation and greater agreement for time-matched observations with average difference of −0.002 with ±0.004 standard deviation. The high statistical agreement in multi-year monthly averaged AOD validates the advanced automatic data quality control algorithms and suggests that migrating research to the Version 3 database will corroborate most Version 2 research results and likely lead to some more accurate results.

8.  Examination of long-term sites in various aerosol source regions indicates mainly subtle changes in AOD, AE and the number of days available; however, in some months, improved cloud screening, high aerosol loading retention, and improved instrument anomaly screening not attained by Version 2 explain larger deviations in these parameters.

AERONET Version 3 has evolved into a database with unparalleled presence in Sun photometry. Future algorithms could include improvements to the detection of cirrus clouds in polar environments, where the ice crystal size is approaching the size of large non-cloud aerosols, the determination of anomalies in high aerosol loading conditions, and the identification of true AOD diurnal dependence versus one generated by an instrument anomaly. Cimel radiometers will also measure the moon to derive lunar AOD (Berkoff et al., 2011). For example, current lunar measurement protocols do not include lunar aureole measurements analogous to the solar aureole measurements, hence the lack of these measurements potentially reduces the ability of the algorithm to remove cirrus clouds at night, and thus a variation of the quality control methodology may need to be developed. Other surface-based remote sensing networks such as MAN (Smirnov et al., 2009), SKYNET (Takamura and Nakajima 2004), and PANDORA (Herman et al., 2009) may benefit by implementing applicable quality control methods established by AERONET.

*Data Availability*. Version 3 AOD data are available from the AERONET web site (https://aeronet.gsfc.nasa.gov) and the web site provides these data freely to the public. Data may be acquired by utilizing several download mechanisms including site-by-site download tools and web service options for near real-time data acquisition.

*Author contributions*. For five years, the AERONET staff (listed from DG to BH) worked individually and collaboratively drawing on their decades of project scientific, engineering and programming expertise to develop and assess the Version 3 AOD processing system presented herein. Traditional assignment of co-authorship is not possible. Aside from the first author, contributing AERONET staff is listed in reverse chronological order based on their start date with the project. JL, JC, and EW provided LIDAR data for development of the cirrus curvature methodology. SK and AL provided gaseous and water vapor absorption coefficients based on radiative transfer models.

*Competing interests:* The authors declare that they have no conflict of interest.



*Acknowledgements*. The AERONET and MPLNET projects at NASA GSFC are supported by the Earth Observing
System Program Science Office Cal-Val, Radiation Science program at NASA headquarters, and various field
campaigns.  NCEP Reanalysis data are obtained routinely from the U.S. National Weather Service Climate
Prediction Center.  We would like to thank Edward Celarier for several discussions and providing the OMI $NO_2$
monthly climatology.  Fred Espenak and Chris O'Byrne (NASA GSFC) provided solar and lunar eclipse predictions
and the Eclipse Explorer software.

We thank the MPLNET PIs for their effort in establishing and maintaining the sites: Arnon Karnieli
(SEDE_BOKER); Sachi Tripathi (Kanpur); Greg Schuster (COVE); Margarita Yela Gonzalez (Santa Cruz
Tenerife); and John Barnes (Trinidad Head).

The authors thank the AERONET calibration facilities in the USA (NASA GSFC, NOAA Mauna Loa Observatory,
and NEON), France (PHOTONS), and Spain (RIMA and Izana).  We thank the following AERONET PIs and their
staff for maintaining the sites and contributing aerosol data: Norm O'Neill, Ihab Abboud and Vitali Fioletov
(PEARL, Toronto, Bratts Lake); Itaru Sano (Osaka); Paulo Artaxo (Rio Branco); Neng-Huei Lin (Lulin); Pucai
Wang and Xiangao Xia (XiangHe); Mikhail Panchenko (Ussurisyk); Arnon Karnieli (SEDE BOKER); Emilio
Cuevas-Agullo (Santa Cruz Tenerife); Joseph Prospero (Ragged Point); Soo-Chin Liew and Santo Salinas Cortijo
(Singapore); S. N. Tripathi (Kanpur); Francisco Reyes (Malaga); and Jean Rajot and Beatrice Marticorena (IER-
Cinzana). A special acknowledgement is given to the AERONET principal investigators and their site staff around
the world who participate in monitoring aerosols to expand our scientific understanding of the Earth.





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



**Table 1.** Nominal AERONET wavelengths for ion assisted deposition filters used for aerosol remote sensing and spectral
corrections or components for each channel.

| Nominal Central Wavelengths (nm) | Filter Bandpass (nm) | Spectral Corrections/ Components |
|---|---|---|
| 340 | 2 | Rayleigh, $NO_2$, $O_3$ |
| 380 | 2 | Rayleigh, $NO_2$ |
| 440 | 10 | Rayleigh, $NO_2$ |
| 500 | 10 | Rayleigh, $NO_2$, $O_3$ |
| 675 | 10 | Rayleigh, $O_3$ |
| 870 | 10 | Rayleigh |
| 935 | 10 | Rayleigh, Aerosol |
| 1020 | 10 | Rayleigh, $H_2O$ |
| 1640 | 25 | Rayleigh, $H_2O$, $CO_2$, $CH_4$ |







**Table 2.** Summary of Cloud Screening Related Quality Control Changes from Version 2 to Version 3.

| Algorithm/Parameter | Version 2 | Version 3 |
|---|---|---|
| Very High AOD Restoration | N/A | $\tau870 >0.5$; $\alpha675\text{-}1020>1.2$ or $\alpha870\text{-}1020>1.3$, restore if eliminated by cloud screening |
| Optical Air Mass Range | Maximum of 5.0 | Maximum of 7.0 |
| Number of Potential Measurements | $N_{remain}<3$, reject all measurements in the day | After all checks applied, reject all measurements in the day if $N_{remain}<MAX\{3$ or 10% of N$\}$ |
| Triplet Criterion | All wavelengths checked; AOD Triplet Variability $> MAX\{0.02$ or $0.03 *\tau_{aerosol}\}$ | Check only wavelengths 675, 870, and 1020nm; AOD Triplet Variability $>MAX\{0.01$ or $0.015 *\tau_{aerosol}\}$ |
| Ångstrom Exponent (AE) Limitation | N/A | If $AE_{440-870nm} <-1.0$ or $AE_{440-870nm} >3.0$, then eliminate triplet measurement. |
| Smoothness Check | D<16 | For AOD500nm (or 440nm) $\Delta\tau_{aerosol}>0.01$ per minute, then remove larger $\tau_{aerosol}$ in pair. Repeat condition for each pair until points are not removed. |
| Solar Aureole radiance Curvature Check (Sect. 3.2.2) | N/A | Using 1020nm solar aureole radiances, compute the curvature ($k$) between 3.2° and 6.0° scattering angle ($\varphi$) at the smallest scattering angle. If $k<2.0E-5$ $\varphi$ and if slope of curvature ($M$) is greater than 4.3 (empirically determined), then radiances are cloud contaminated. For sky scan measurements, all $\tau_{aerosol}$ measurements are removed within 30 minutes of the sky measurement. For Model T, special aureole scan measurements will remove all $\tau_{aerosol}$ within a two minute period superseding any sky scan aureole measurements. |
| Standalone Measurements | N/A | If no data exists within 1 hour of a measurement, then reject it unless AE440-870nm>1.0. |
| AOD Stability Check | Same as Version 3 | Daily averaged AOD 500nm (or 440nm) has $\sigma$ less than 0.015, then do not perform 3-$\sigma$ check. |
| 3-$\sigma$ Check | Same as Version 3 | AOD 500nm and AE440-870nm should be within the MEAN$\pm3\sigma$; otherwise, the points are rejected. |





**Table 3.** AERONET and MPLNET sites and date ranges used for assessing cirrus and non-cirrus cloud presence

| Site | Latitude | Longitude | Elevation (meters) | Date Range |
|------|----------|-----------|--------------------|------------|
| GSFC | 38.9925° N | 76.8398° W | 87 | May 2001–Jan 2013 |
| COVE | 36.9000° N | 75.7100° W | 37 | May 2004–Jan 2008 |
| Kanpur | 26.5128° N | 80.2316° E | 123 | May 2009–Jan 2013 |
| SEDE_BOKER | 30.8550° N | 34.7822° E | 480 | Nov 2007–Apr 2013 |
| Santa_Cruz_Tenerife | 28.4725° N | 16.2473° W | 52 | Nov 2005–Jan 2013 |
| Singapore | 1.2977° N | 103.7804° E | 30 | Aug 2009–Jan 2013 |
| Ragged_Point | 13.1650° N | 59.4320° W | 40 | Jun 2008–Jan 2013 |
| Trinidad_Head | 41.0539° N | 124.1510° W | 105 | May 2005–Feb 2013 |







**Table 4.** Thresholds used to determine the independent and dependent AOD diurnal dependence. Satisfying both the slope and
correlation coefficient (*R*) conditions would constitute the possible removal of all measurements for a day.

| Day Removal Type | AOD Diurnal Shape | Analyzed Period | Slope Threshold | R Threshold |
|---|---|---|---|---|
| Independent | Concave | AM, PM, Day | >0.25 | >0.974 |
| Dependent | Concave | AM, PM | >0.04 | >0.94 |
| Dependent | Concave | Day | >0.1 | >0.94 |
| Dependent | Convex | AM, PM, Day | <−0.02 | <−0.94 |
| Dependent – $\tau_{avg}$<0.1 | Convex | AM, PM, Day | <−0.1 | <−0.94 |
| Independent – 2 or more Silicon wavelengths (440, 675, 870, 1020nm) or 1640nm InGaAs | Concave | AM, PM, Day | >0.1 Day or AM & PM > 0.02 | >0.94 |





**Table 5**. Statistics corresponding to Fig. 21 and Fig. 22 for AOD interpolated to 500nm, Ångstrom exponent 440–870nm, precipitable water (cm), and the number of days. Version 3 Level 2.0 and Version 2 Level 2.0 data are compared for the same multi-year monthly averages when sites have a total of more than 1000 days for all months and more than 30 days in each month. Data represented as "Matched" indicates the further condition that the exact observations were matched in Version 2 and Version 3 Level 2.0 multi-year monthly average data sets. Note that PW values for the "Matched" data set are approximately the same as the unmatched data set.

| Parameter | $AOD_{500nm}$ (V3−V2) Unmatched | $AE_{440-870nm}$ (V3−V2) Unmatched | PW (cm) (V3−V2) Unmatched | Days (V3−V2) Unmatched | $AOD_{500nm}$ (V3−V2) Matched | $AE_{440-870nm}$ (V3−V2) Matched |
|---|---|---|---|---|---|---|
| Average | 0.002 | −0.01 | −0.02 | −0.4 | −0.002 | −0.03 |
| Standard Deviation | 0.022 | 0.10 | 0.06 | 24.8 | 0.004 | 0.10 |
| Maximum | 0.247 | 0.29 | 0.34 | 150 | 0.015 | 0.35 |
| Minimum | −0.166 | −1.54 | −0.45 | −130 | −0.029 | −1.63 |
| Number of Months | 2953 | 2953 | 2953 | 2953 | 2514 | 2514 |





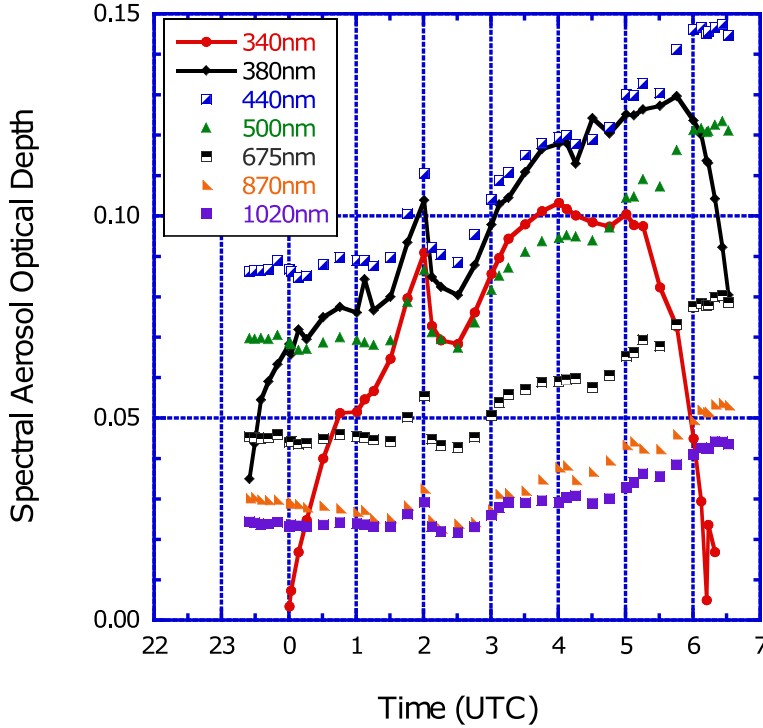

1573

**Figure 1.** Aerosol optical depth (AOD) data from AERONET Ussuriysk site (43.70° N, 132.16° E) on 30 November 2005 shows electronic instability. For the Cimel Model 4 instruments, the electronic sensitivity of the UV AOD data (340nm and 380nm) can be high due to a bad amplifier. The resulting AOD data for the UV channels are out of spectral dependence the entire day with a maximum error for large optical air mass due to large dark current values. The UV channels (identified by line plots) are removed by the quality control while preserving other wavelengths that are not affected by this condition.

1579





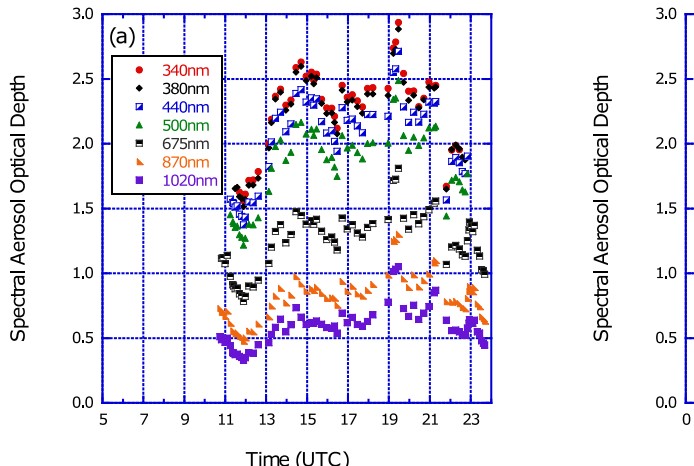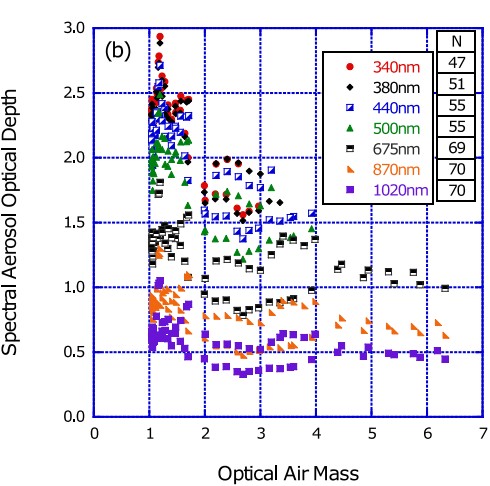

**Figure 2.** Spectral dependent low digital number removal at NASA Goddard Space Flight Center (GSFC; 38.99°N, 76.84°W). **(a)** Level 1.0 AOD data from GSFC on 8 July 2002 are plotted for the Quebec forest fire smoke event. Significantly fewer Level 1.0 AOD data are available for the shorter wavelengths near local sunrise (~11 UTC) and sunset (~23:30 UTC). **(b)** The distribution of the AOD measurements with respect to optical air mass clearly shows the removal of short wavelengths for large air mass in this fine mode aerosol event. The high aerosol loading due to smoke and haze results in significant extinction at UV and visible wavelengths, which corresponds to low digital counts. The low digital count quality control removes AOD measurements impacted by diffuse radiation scattered into the instrument field of view (Sinyuk et al., 2012).





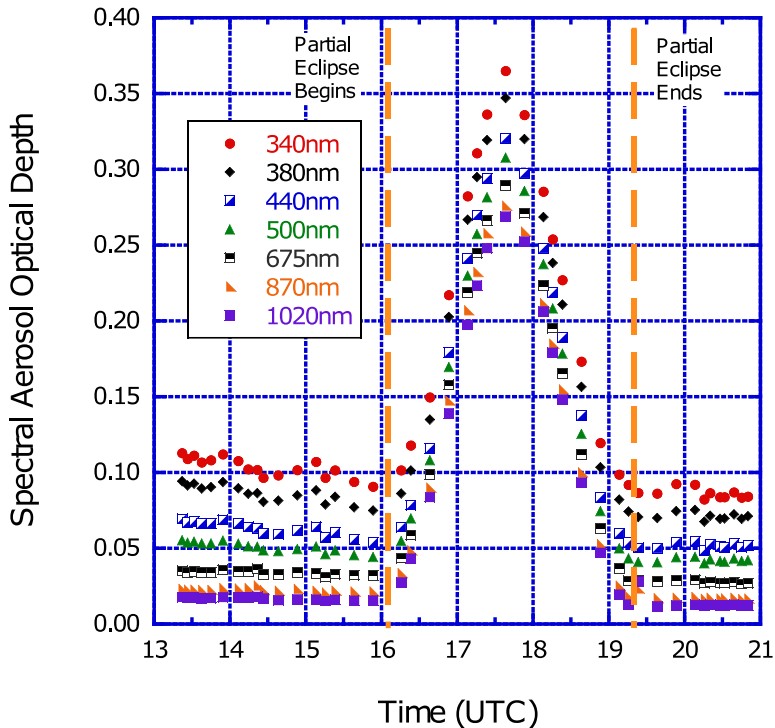

**Figure 3.** Eclipse circumstance at the NASA Goddard Space Flight Center (GSFC; 38.99° N, 76.84° W) on 25 December 2000 between 16:04:13 UTC and 19:16:25 UTC. The maximum AOD during the eclipse occurs at the maximum obscuration of 0.42, which results in a change of ~0.28 for AOD 500nm compared to data before and after the solar eclipse. Utilizing the NASA Solar Eclipse database, the AOD measurements are removed between the partial eclipse first contact and partial eclipse last contact as denoted by the vertical dashed lines.





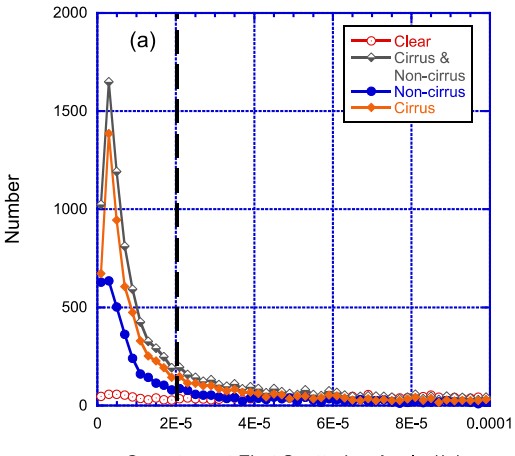

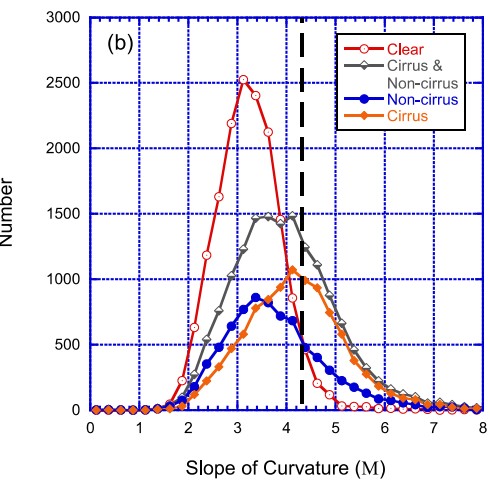

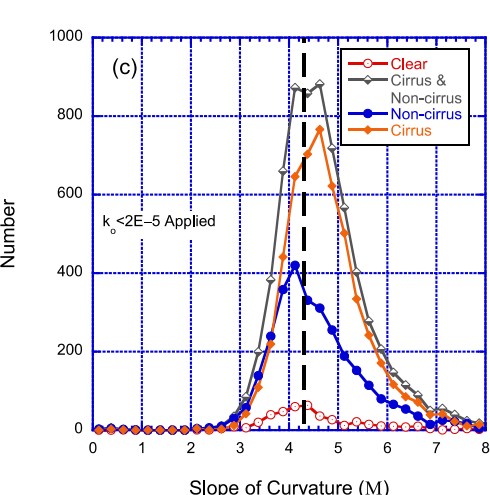

**Figure 4.** NASA Goddard Space Flight Center (GSFC; 38.99° N, 76.84° W) AERONET data coincident with MPLNET LIDAR derived sky condition categories (Clear, both Cirrus and Non-cirrus clouds, Non-cirrus clouds, and Cirrus clouds) from 2001–2013. The AERONET solar aureole 1020nm radiances are used to calculate the curvature at the first scattering angle ($k_o$) and the slope of curvature (M) between 3.2° and 6.0° scattering angles. **(a)** The number distribution of $k_o$ is shown and the dashed vertical line at $k_o$ equals 2E−5 indicates the threshold where values less than 2E−5 are considered possibly cirrus cloud contaminated (the x-axis is truncated at 1E−4 for viewing purposes). **(b)** The number distribution of M is shown and M greater than 4.3 are considered to be possibly cirrus cloud contaminated (the dashed vertical line indicates the threshold of 4.3). **(c)** Similar to panel (b) except that the $k_o$ threshold ($k_o$<2E−5) is applied first and, as a result, data greater than 4.3 in this panel are considered to be cirrus cloud contaminated.





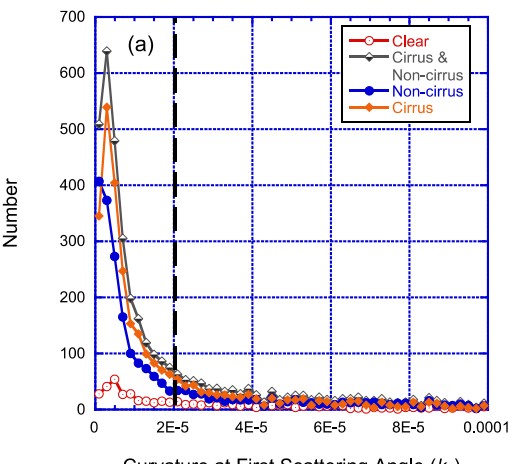

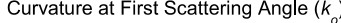

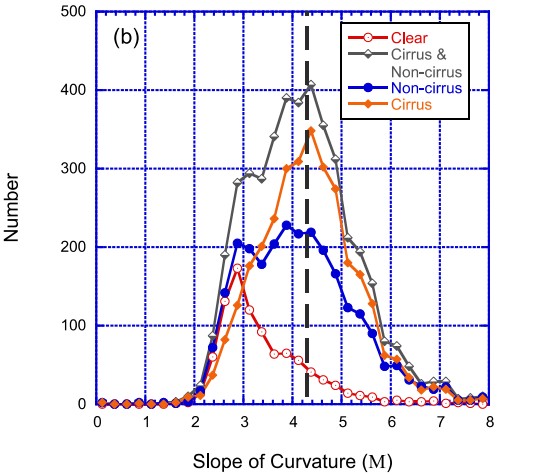
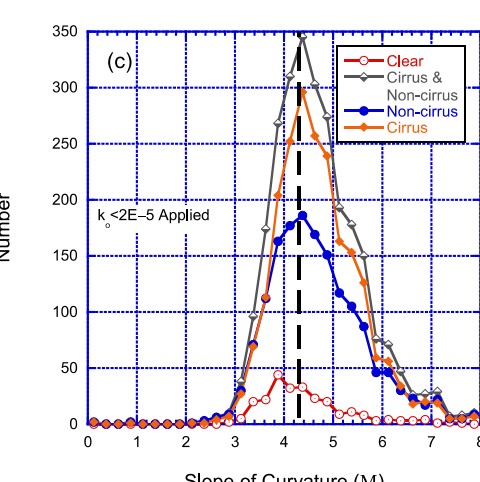


**Figure 5.** Similar to Fig. 4, except for Singapore (1.29° N, 103.78° E) from 2009–2013.



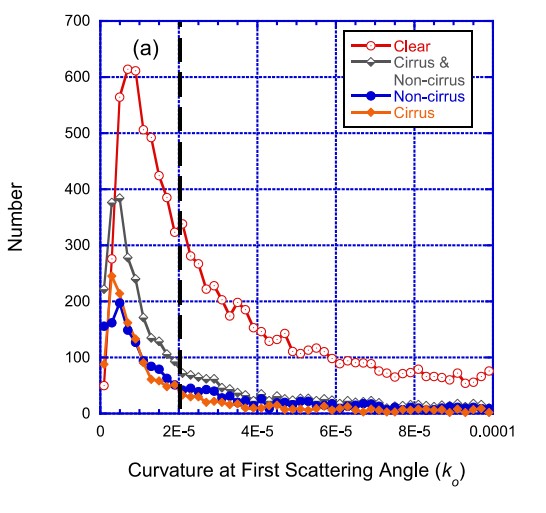

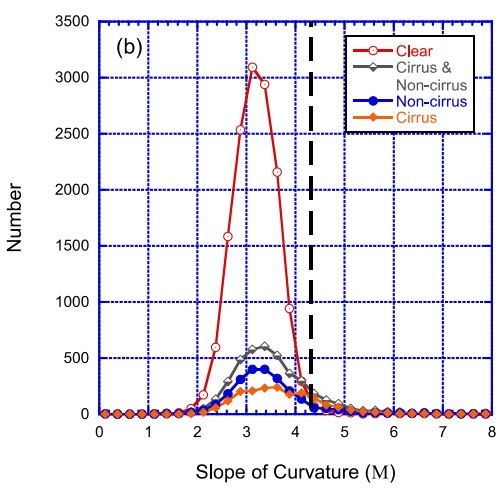

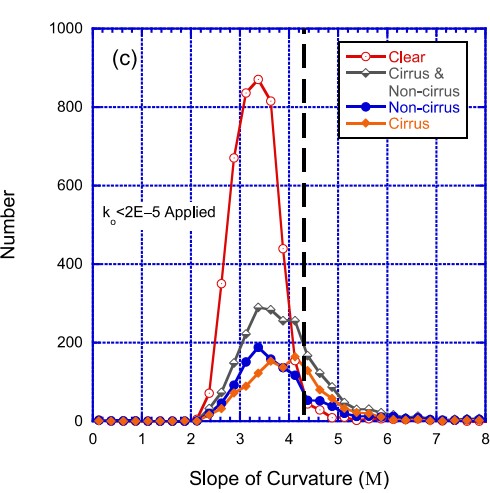


**Figure 6.** Similar to Fig. 4, except for SEDE BOKER (30.85° N, 34.78° E) from 2007–2013.






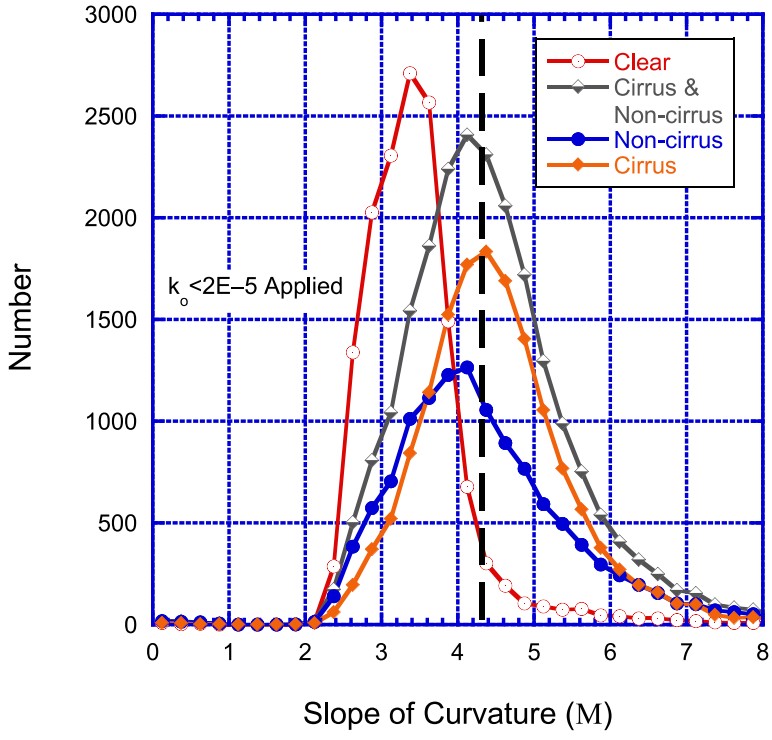

**Figure 7.** Similar to Fig. 4c including all analyzed sites in Table 3.





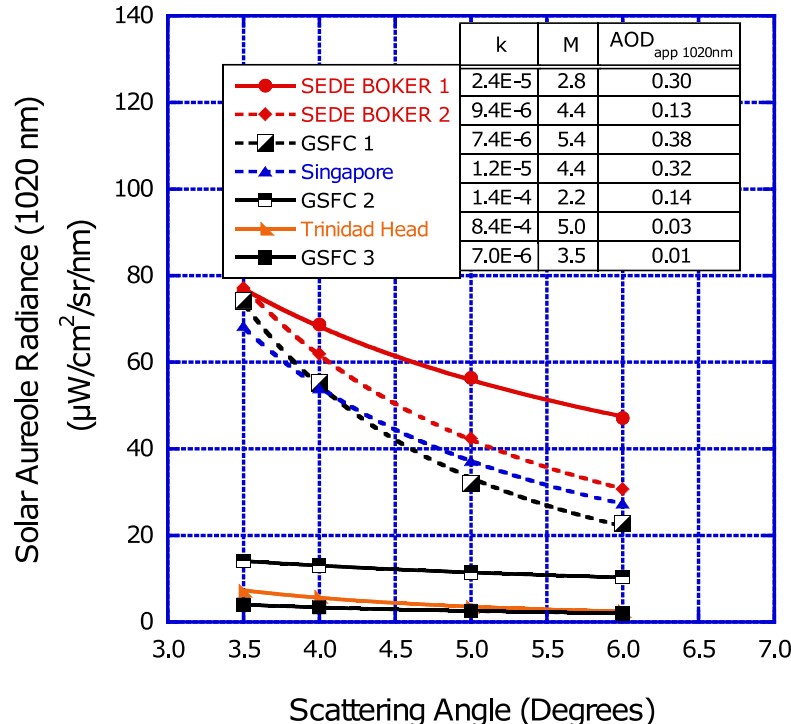


**Figure 8.** The solar aureole 1020nm radiance versus the scattering angle in degrees for selected sites. Data plots with the dashed lines (i.e., SEDE BOKER 2, GSFC 1, and Singapore) all qualify for the removal of data due to optically thin homogeneous cloud contamination.







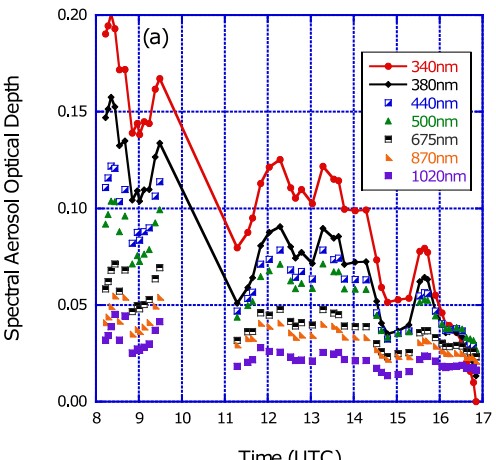
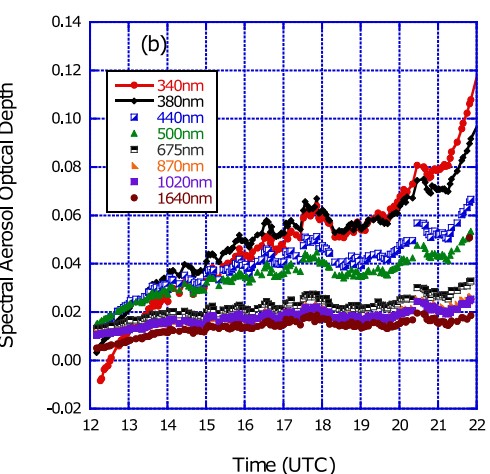


**Figure 9.** Time shifted aerosol optical depth (AOD) data examples at Malaga (36.72° N, 4.48° W) and Toronto (43.79° N, 79.47° W). Note the line plot is used to emphasize the 340nm and 380nm AOD impact for the time shift. **(a)** The Level 1.5 AOD cloud screened only data measured at the Malaga site on 30 January 2014. These data show the time shifted AOD especially at short wavelengths represent the instrument clock is too fast. **(b)** The Level 1.5 AOD cloud screened only data measured at the Toronto site on 24 September 2013. The time shifted aerosol optical depth especially at short wavelengths represent when the instrument clock was too slow. Panel (a) also shows the algorithm can be used with data gaps and lower temporal resolution measurement interval compared to panel (b).

1630





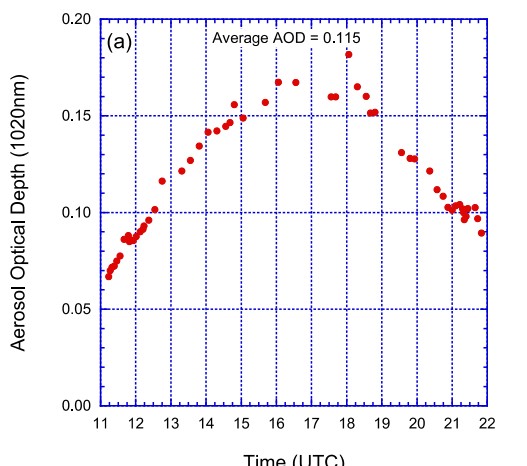
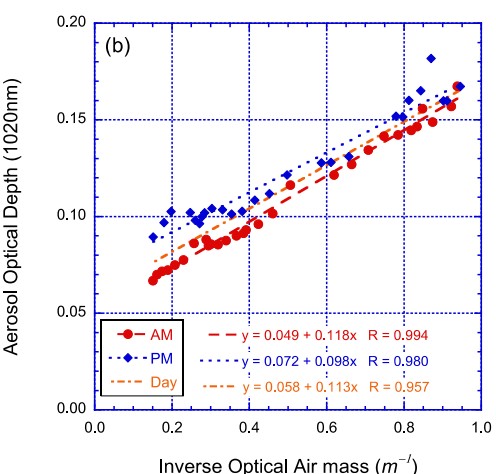

**Figure 10**. AERONET data collected at Rio Branco (9.96 °S, 67.87° W) on 30 August 2011. The AOD 1020nm Level 1.5 with only the cloud screening algorithm applied to the data. **(a)** The AOD diurnal dependence presents a concave shape during the solar day. **(b)** The AOD 1020nm and the inverse optical air mass show a highly correlated linear fit and the slope is significant for the full day (day) and morning (AM), and afternoon (PM). Data separation for AM and PM is defined by the local solar noon, which is 16:31:28 UTC at Rio Branco.





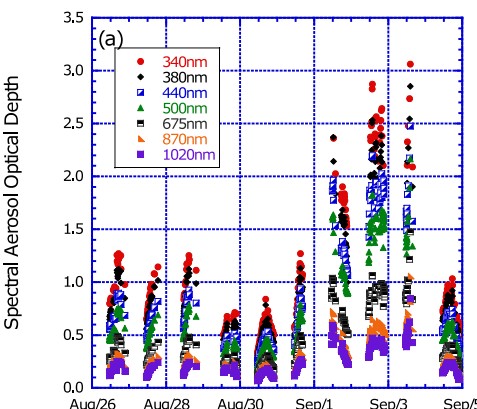
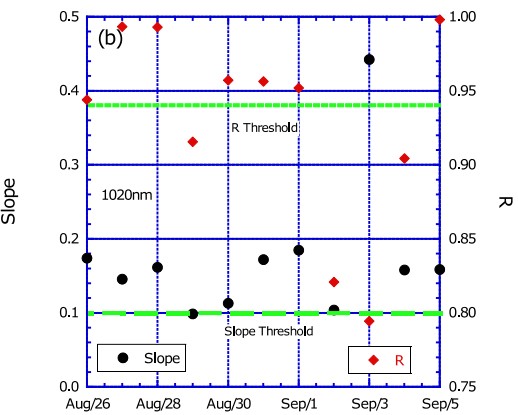

**Figure 11.** AERONET data collected at Rio Branco (9.96° S, 67.87° W) from 15 August to 30 September 2011. **(a)** The time series of Level 1.5 spectral AOD (cloud screened only) data is plotted from 26 August to 5 September 2011 and shows repeated diurnal dependence for varying magnitudes of AOD. **(b)** The robust linear fit slope and correlation coefficient (R) is calculated from the AOD 1020nm versus the inverse of the optical air mass (m$^{-1}$). For the full day evaluation, the green dashed line indicates the threshold for the slope parameter at 0.1 and the solid green line indicates the threshold for the correlation coefficient (R = 0.94). Both the slope and R must exceed these thresholds for at least three days scanning from the current day to the last occurrence within the 20-day period to remove the spectral AOD, and in this circumstance, all of the data are removed for the period for Levels 1.5 and 2.0.





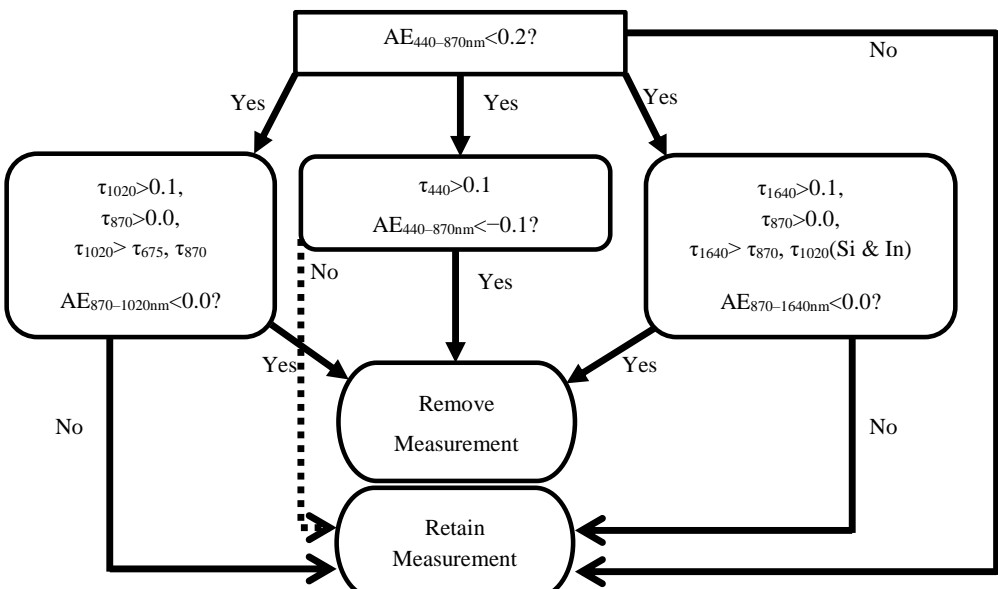


**Figure 12.** Flowchart of the reverse spectral dependence algorithm used to remove cloud contamination artifacts and instrument
anomalies. The 1640nm wavelength is available on some Cimel Model 5 instruments and all Model T instruments.





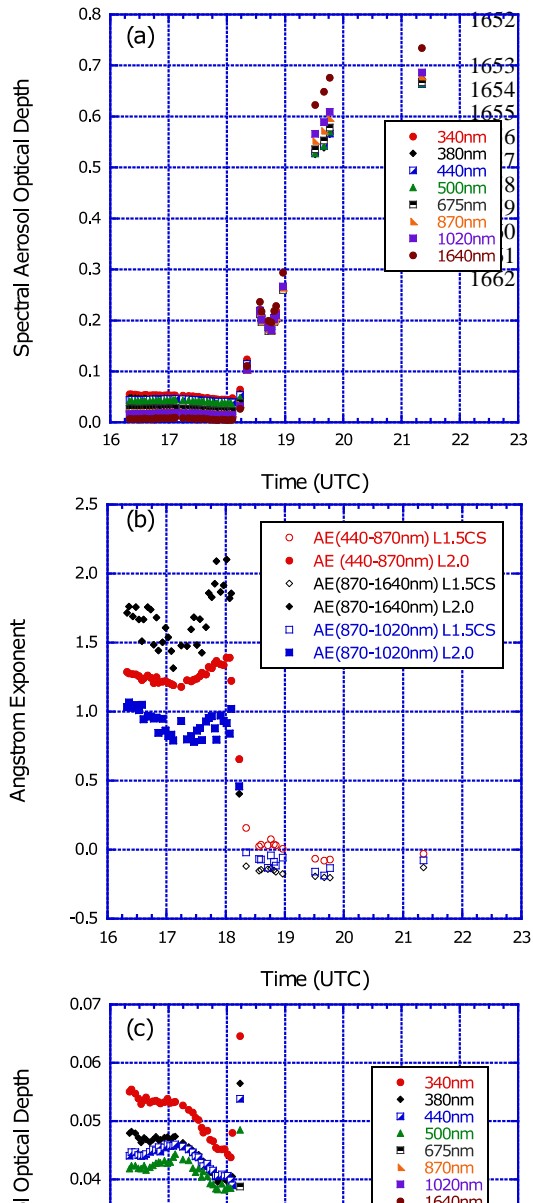

**Figure 13.** Data from Bratts Lake (50.20° N, 104.71° W) on 7 January 2007. **(a)** The Level 1.5 data with only the cloud screening (CS) algorithm applied shows cloud contaminated data remain after 18:10 UTC. **(b)** For the same period as (a), the Ångstrom exponent values decreased significantly to a level where coarse mode aerosol particles are not expected. **(c)** The final Level 1.5 and Level 2.0 data series after the reverse spectral dependence quality control or additional cloud screening method has been applied to the standalone Level 1.5 CS data.



1663

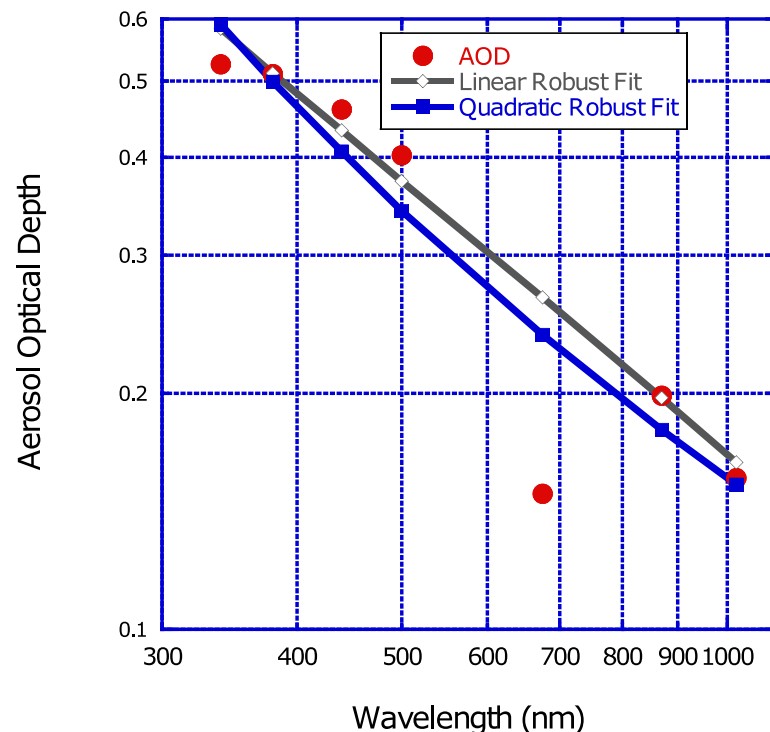

**Figure 14.** AERONET data from the Osaka (34.65° N, 135.59° E) site on 16 October 2006 at 22:02:11 UTC. The plot shows AOD versus the wavelength with lines identifying the linear and quadratic robust regression fits on logarithmic scale used by the AOD spectral dependence algorithm. The 675nm channel is clearly anomalous with fits differing by 0.12 for linear and 0.09 for quadratic. In addition, the AOD 340nm appears anomalous with deviations of 0.06 from linear fit and 0.07 from quadratic fit. While both wavelengths exceed their respective AOD thresholds (0.023 for 675nm and 0.051 for 340nm), the algorithm determines the maximum deviation for linear and quadratic fits and removes the AOD 675nm measurement. A subsequent scan by the algorithm determined that the remaining AOD measurements from 340nm to 1020nm were within the established fit deviation thresholds.

1672



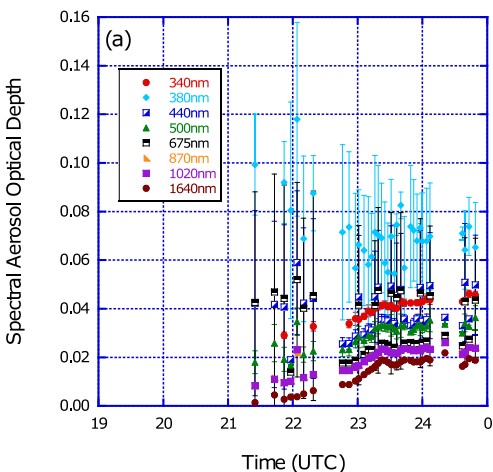 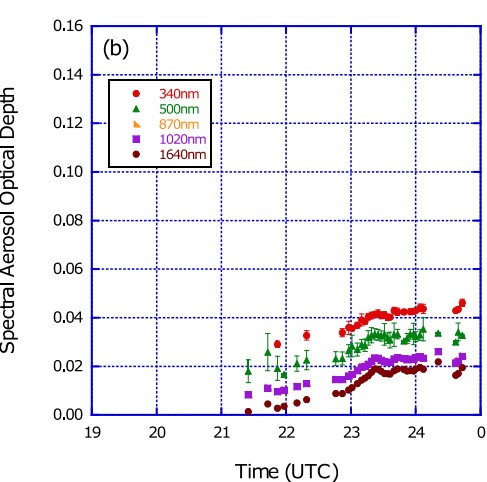

1673

**Figure 15.** Spectral AOD exhibiting large triplet variability at PEARL (80.05° N, 86.42° W) on 25 August 2013. **(a)** Version 3 Level 1.5 cloud screened only data is plotted with large triplet variability and these data were not removed by the cloud screening. The error bars represent the triplet variability (AOD Max – AOD Min) divided by 2 so the full range represents the AOD triplet variability. The large triplet variability occurs mainly at shorter wavelengths than 675nm. **(b)** Data affected by large triplet variability (i.e., AOD 380nm, AOD 440nm, and AOD 675nm) are removed by using the Level 1.5 large triplet variability quality controls.







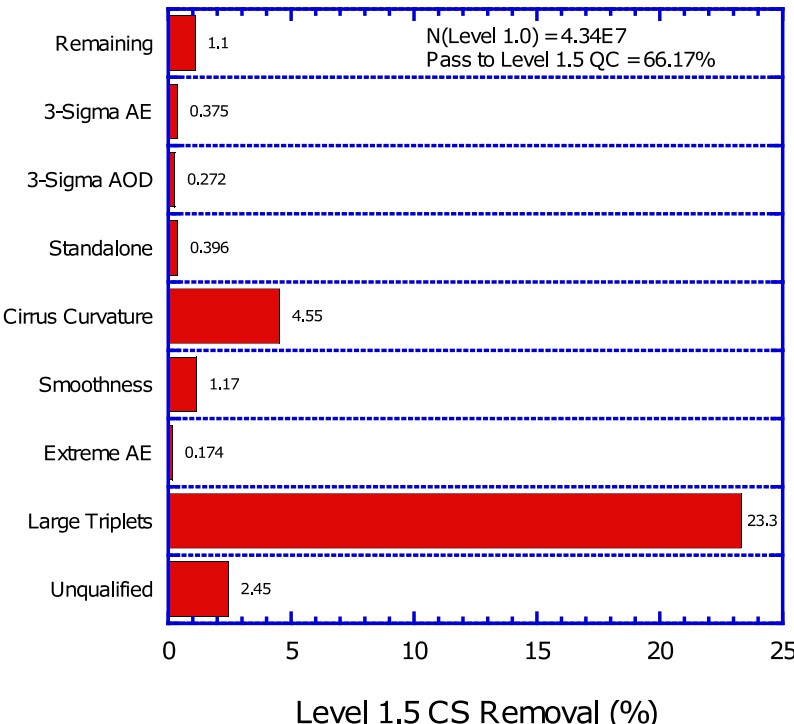

Level 1.5 CS Removal (%)


**Figure 16**. The Level 1.0 AOD measurement removal by the Level 1.5 cloud screening algorithm from 1993 to 2018. The plot
shows the impact of the major cloud screening steps in the Level 1.5 cloud screening algorithm and removal of these data applies
to all wavelengths. The triplet criterion removes more than 23% of the Level 1.0 data. Nearly 5% of the Level 1.0 data are
removed due to cirrus cloud contamination. The "Remaining" category indicates the check performed after each cloud screening
step to determine if enough measurements are available and do not meet the high AOD retention criteria. The "Unqualified"
category indicates data that are negative or lack sufficient channels to participate in the cloud screening algorithm.





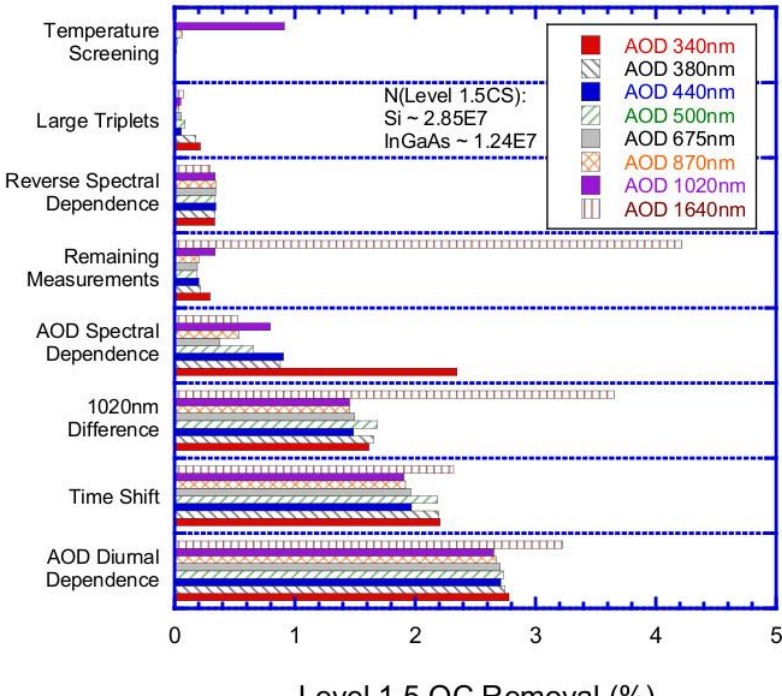


**Figure 17.** Level 1.5 quality control algorithm wavelength dependent impacts for each major step for the period analyzed from 1993–2018. The most significant removal for most channels is due to AOD diurnal dependence, time shift, and difference between AOD 1020nm on the Silicon and InGaAs detectors (resulting from collimator inconsistency). The AOD 340nm has significant removal of AOD spectral dependence. The 1640nm InGaAs channel has significant removal by "Remaining Measurements" since this wavelength cannot be checked for quality when the Silicon channels are not available. Temperature screening mostly applies to the 1020nm Silicon wavelength due to its strong temperature dependence near the edge of the signal sensitivity of the Silicon photodiode detector.

1698



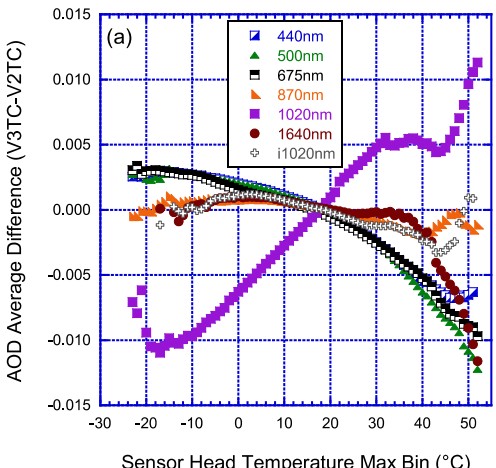

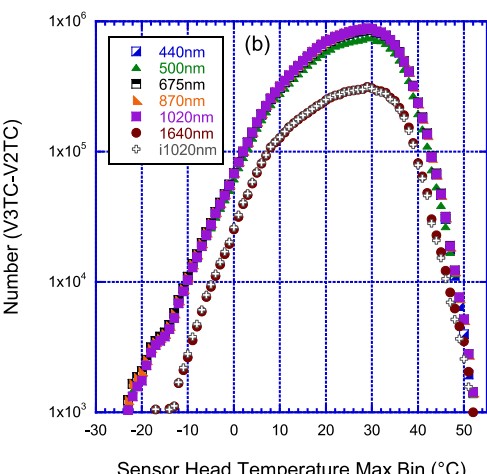

**Figure 18**. Difference in AOD response between Version 3 and Version 2 temperature correction applied to Version 3 AOD data based on the sensor head temperature from 1993–2018. The Version 2 temperature correction assumes temperature ranges for 1020nm and no temperature correction for all other wavelengths, while Version 3 temperature correction characterizes the temperature response for each filter or set of default filters for each instrument for wavelengths ≥400nm. **(a)** The AOD average difference plotted for each 1°C temperature bin from −25°C to +55°C. The AOD 1020nm exhibits an opposite trend compared to the other wavelengths varying from −0.01 at low temperatures and up to +0.01 at high temperatures. Other wavelengths have slight differences at cold temperatures but apparent dependencies at high temperatures greater than 40°C possibly due to extrapolation of the temperature coefficients to higher temperatures. **(b)** The number of measurements plotted for each 1°C temperature bin with a minimum of 1000 observations.





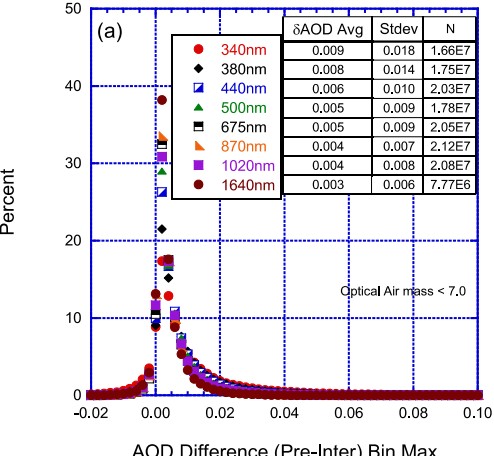

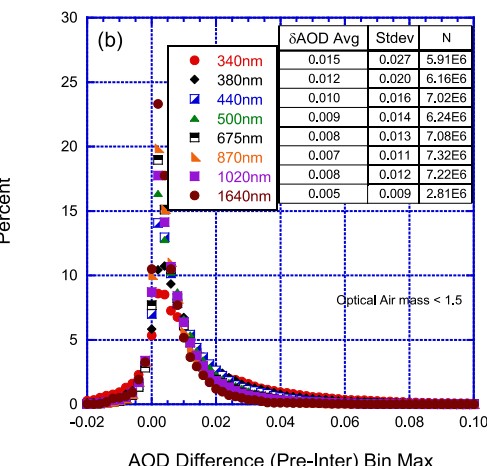


**Figure 19.** Using data qualified as Version 3, Level 2.0, aerosol optical depth (AOD) average difference comparing
measurements only with the pre-field calibration applied versus instruments with both the pre-field and post-field calibrations
applied from 1993–2018. Calibration sites are excluded from the analysis. The histogram of AOD differences is provided for the
optical air mass $1.0 \leq m < 7.0$ range in panel **(a)** and $1.0 \leq m < 1.5$ range in panel **(b)**. The average difference is largest for the UV
wavelengths and smallest for the longer wavelengths.





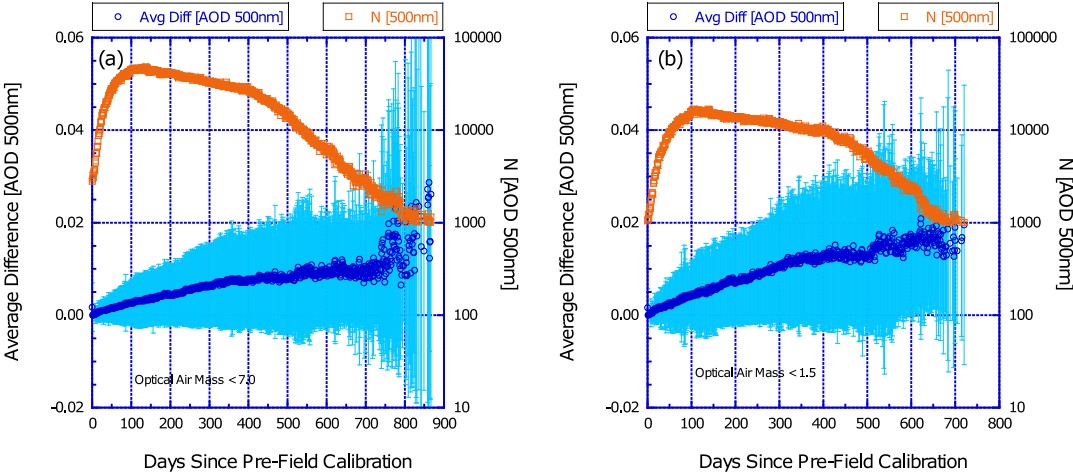


**Figure 20.** Using data qualified as Version 3 Level 2.0 aerosol optical depth (AOD) 500nm average difference comparing
measurements only with the pre-field calibration applied versus instruments with both the pre-field and post-field calibrations
applied from 1993–2018. The AOD average differences are provided for the optical air mass 1.0≤m<7.0 range in panel **(a)** and
1.0≤m<1.5 range in panel **(b).** Vertical bars represent the standard deviation for each day bin. The secondary y-axis in
logarithmic scale represents the number of measurements of AOD 500nm for each day bin.






**Figure 21**. Comparison of Version 3 and Version 2 Level 2.0 multi-year monthly average data sets. **(a)** The aerosol optical depth
(AOD) interpolated to 500nm to include data from instruments without 500nm. **(b)** The Ångstrom exponent (AE) is calculated
utilizing the inclusive ordinary least squares regression fit from 440–870nm. **(c)** The precipitable water in cm is derived from the
935nm water vapor channel. **(d)** The difference in the number of days is determined for each monthly long-term average.








**Figure 22.** Comparison of Version 3 and Version 2 Level 2.0 multi-year monthly average data sets for time matched instantaneous observations in both data sets. The panels are similar to those in Fig. 21.







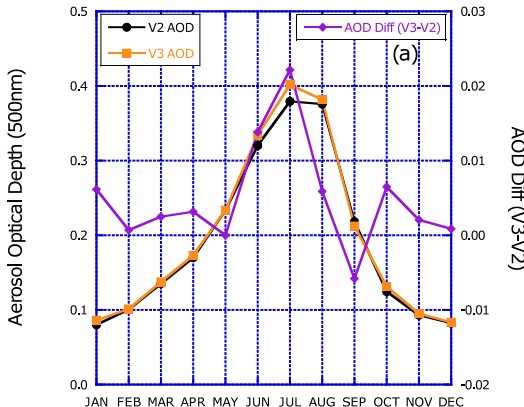

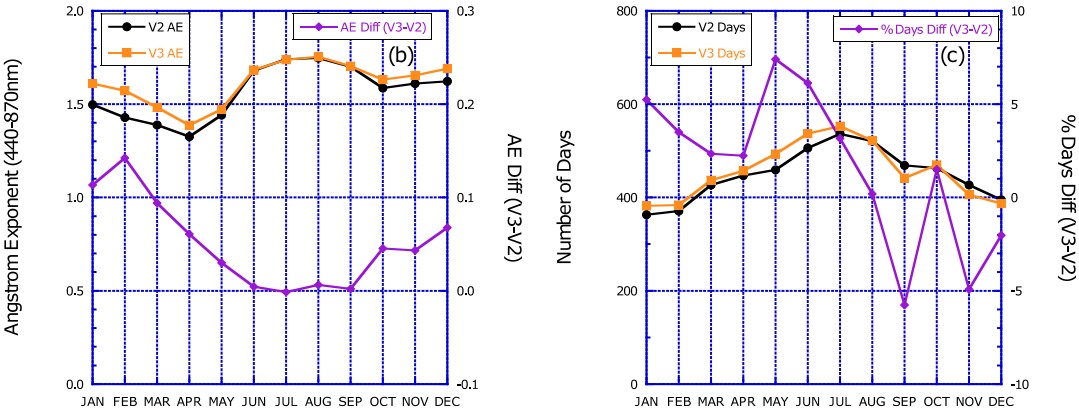


**Figure 23**. Long-term multi-year (1993–2016) monthly average comparisons of the Version 3 and Version 2 Level 2.0 data sets at the NASA Goddard Space Flight Center (GSFC), Maryland, USA. The panel **(a)** provides the AOD interpolated to 500nm for each version on the primary y-axis and differences on the secondary y-axis. The panels **(b)** and **(c)** are plotted similarly for the AE$_{440–870nm}$ and the number of days in the multi-year monthly average, respectively.




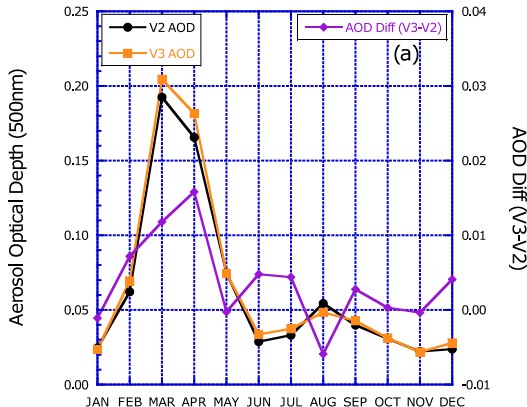

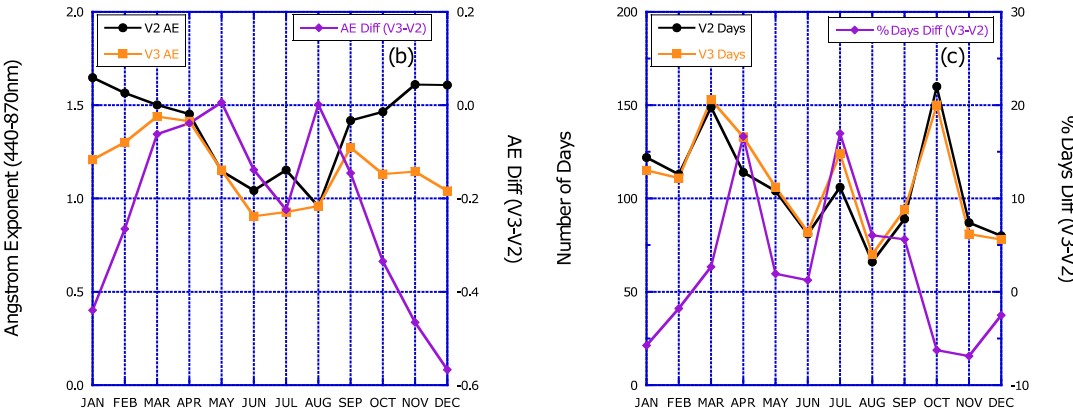

**Figure 24.** Similar to Fig. 23 except for Lulin, Taiwan (23.47° N, 120.87° E) from 2006–2017.





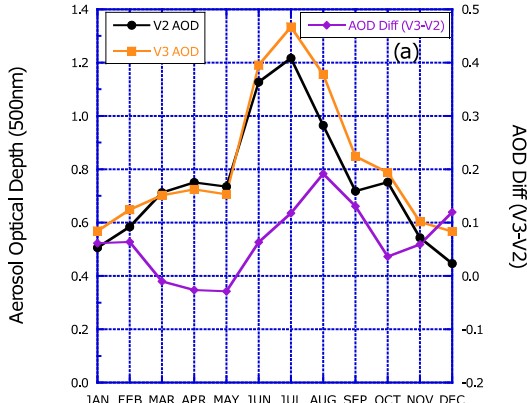

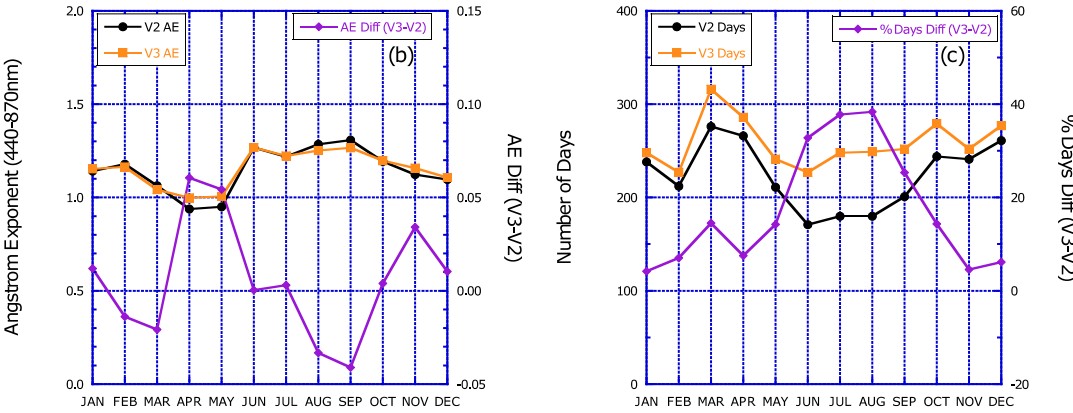


**Figure 25.** Similar to Fig. 23 except for XiangHe, China (39.75° N, 116.96° E) from 2001–2017, except 2009.





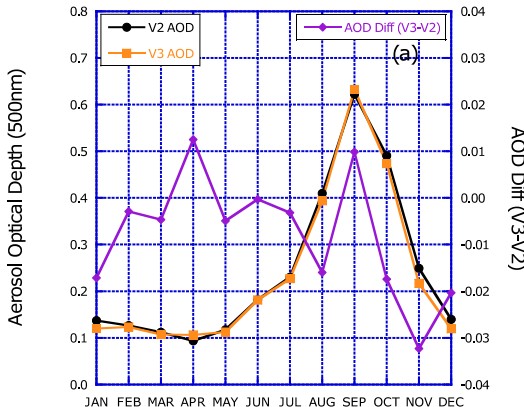

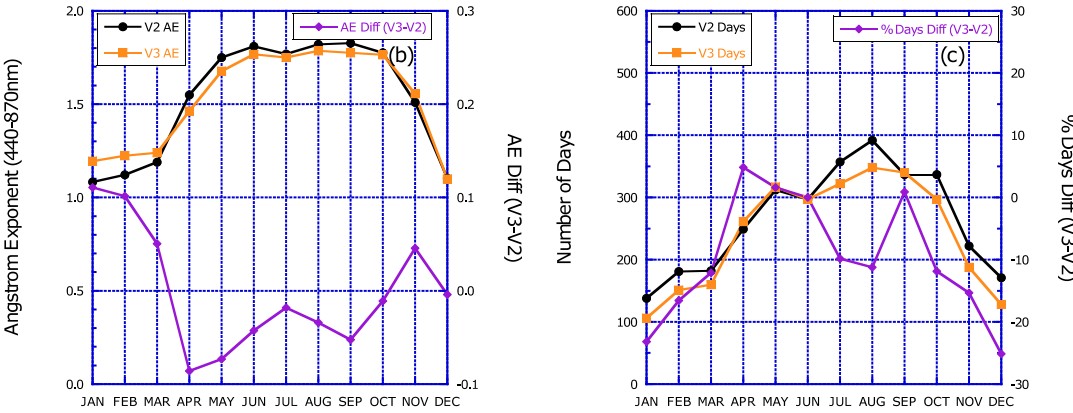


**Figure 26.** Similar to Fig. 23 except for Mongu, Zambia (15.25° S, 23.15° E) from 1997–2010.




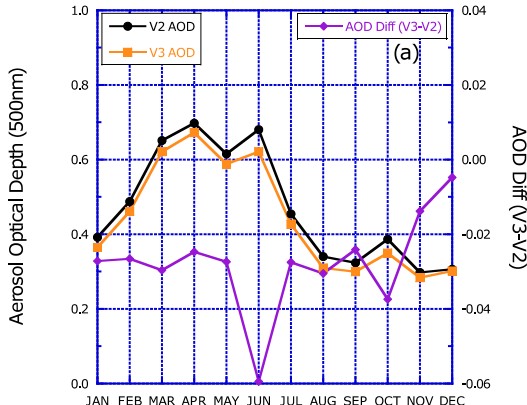

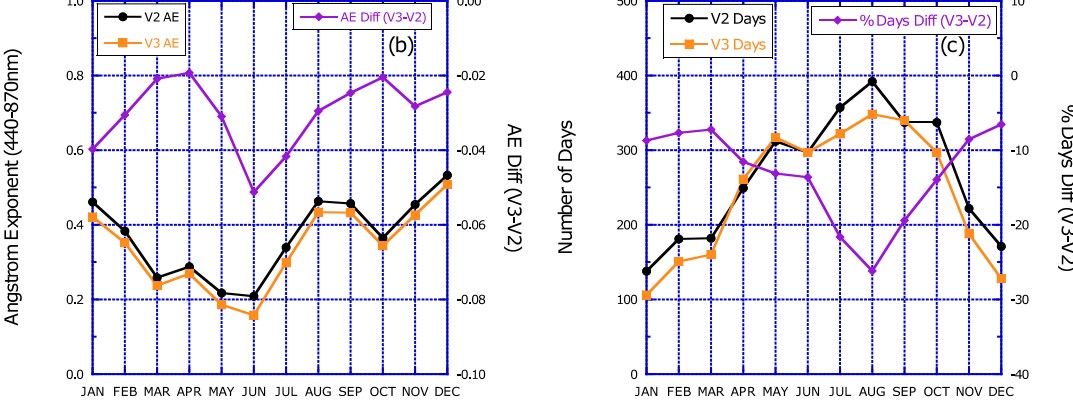


**Figure 27.** Similar to Fig. 23, except for IER-Cinzana, Mali (13.28° N, 5.93° W) from 2004-2017.