# Peer review of "Advancements in the Aerosol Robotic Network (AERONET)"

_Atmospheric Measurement Techniques, 2018_

## Short Comment (SC1) · 16 Sep 2018

Thank you for publishing this interesting paper on AERONET Version 3 database.

In page 4 lines 83-85 I miss the important following reference that is the first paper on the new AERONET standard Cimel photometer:

Barreto, Á., Cuevas, E., Granados-Muñoz, M.-J., Alados-Arboledas, L., Romero, P.

M., Gröbner, J., Kouremeti, N., Almansa, A. F., Stone, T., Toledano, C., Román, R., Sorokin, M., Holben, B., Canini, M., and Yela, M.: The new sun-sky-lunar Cimel CE318-T multiband photometer – a comprehensive performance evaluation, Atmos. Meas. Tech., 9, 631-654, https://doi.org/10.5194/amt-9-631-2016, 2016.

In addition, this paper was co-authored by two members of the AERONET team (Brent and Mikhail)

As for antecedents on lunar AOD, I highlight the omission of the papers by Barreto et al.: the previous one, in which an extensive characterization of lunar AOD is performed with the new CE318-T, and the following, using the classical CE318 with some modifications to track the moon.

Barreto, A., Cuevas, E., Damiri, B., Guirado, C., Berkoff, T., Berjón, A. J., Hernández, Y., Almansa, F., and Gil, M.: A new method for nocturnal aerosol measurements with a lunar photometer prototype, Atmos. Meas. Tech., 6, 585–598, doi:10.5194/amt-6-585-2013, 2013.

At least one of them should be referenced.

Thank you in advance for considering these suggestions, Emilio Cuevas (ecuevasa@aemet.es)

---

## Author Comment (AC1) · 21 Sep 2018

Thank you for your comments in regards to references. We have added Barreto et al. (2013, 2016) references to the manuscript.

In addition, we have added the following reference with respect to lunar measurements:

Li, Z., Li, K., Li, D., Yang, J., Xu, H., Goloub, P., and Victori, S.: Simple transfer calibration method for a Cimel Sun–Moon photometer: calculating lunar calibration coefficients from Sun calibration constants, Appl. Opt. 55, 7624-7630, https://doi.org/10.1364/AO.55.007624, 2016.

---

## Referee Comment (RC1) · Anonymous Referee #2 · 7 Nov 2018

This is a seminal work providing in depth analysis of the new algorithms used by AERONET/CIMEL aerosol network.

Here are some comments that hopefully improve the already high standards article.

L68 PHOTONS (PHOtométrie pour le Traitement Opérationnel de Normalisation Satellitaire) and (RIMA) Red Ibérica de medida Fotométrica de Aerosoles.

[Figure]

L134-135 This is true only if V(lamda) error is independent of air mass. E.g. not true if theoretically signal non linearity errors are involved. Have the instruments been tested for non linearity issues ?

L160 what is the impact in AOD due to the pressure uncertainty?

A table summarizing the uncertainty budget that is related to all factors included in section 2 for V2 and V3 pointing out in which aspects the V2 uncertainty has been improved based on the new V3 QA/QC procedures presented here, would give more value to the whole manuscript.

L59-63 It would be informative to mention other less dense existing surface based networks e.g. SKYNET, GAW-PFR.

L173-174 what happens with these (A and B) calculations due to the change of the 935nm filter over time?

L199 & 205 can O3 and NO2 climatology be found somewhere on the AERONET web page?

L206-214 Interesting results on this aspect can be found in Arola and Koskela, 2004

L245-253 it would be interesting to report a summary of the results found for the temperature dependence on AOD based on the characterization of almost all instruments, as mentioned. Is this temperature dependence more or less similar for all instruments / Are all instruments characterized for temperature dependence ?

L310 it has to be explained why this formula is used. Is it purely empirical?

L409 I think (whichever is greater) is not correct and is not needed. It is sufficient: less than 3 or less than 10%.

Cloud screening algorithms: A common issue is the AOD rejection from the cloud algorithms for dust related high and with high temporal variability, AOD cases. As accurate and also complete dust aerosol series is extremely important for aerosol-
radiative forcing global studies; are there any aspects of this new algorithm that are applied in such cases ? Could the authors include an example or an assessment of such cases ? (such as the very nice examples they already have for other cases)

Very high AOD events could impact the sun-photometer's AOD retrieval based on the diffuse / forward scattered light entering the field of view of the instrument resulting an underestimation of the calculated AOD. Are there any control/corrective measures for such cases ?

It is unclear to me why figure 20 cases (in XX') axis differ for the two figures. Aren't we talking about the same data only analyzed for different air masses ? If not it can be clarified.

There is a number of publications pointing out small but systematic differences in wv retrievals from AERONET and the ones from other instruments/methods (GPS, microwave radiometers) is V3 results lead to better results for such comparisons?

From the point of view of the AERONET user. There are a lot of issues presented here that contribute to the better interpretation of case studies, climatologies related with AERONET/AOD data. For such cases (lets say for example an analysis of a high AOD affected area with frequent dust events that is unknown how many of them have been captured and how many of them have been rejected by the cloud algorithm) the AERONET user could have two options:

a. To cite this paper speculating that part of his/her results could be related with various QA/QC definitions presented here.

b. To actually use and/or modify accordingly such QA/QC algorithms in order to have more solid conclusions. So the question is: will these algorithms be available to the AERONET users ?

Again, this is a very high quality paper and the authors have done a terrific job in terms of analysis, interpretation and presentation of sun-photometric quality control

and assurance procedures for AOD and WV retrievals.

---

## Referee Comment (RC2) · Anonymous Referee #1 · 13 Nov 2018

It is a great effort in transparency of the quality control procedures, as well as automation and objectivity. However it can be sometimes too specific and the applicability to other AOD networks difficult. The novel approach in Section 3.2.2 is of special interest and would deserve a paper by itself. Similarly, the efforts in section 3.3 are a valuable contribution to any AOD network, although sometimes too specific approach reduces the applicability of the method. I recommend the paper is published after minor revision.

[Figure]

Specific comments:

I miss some references in the introduction, for instance articles or web links to the list of contributing networks (lines 65-68), or some references about the Cimel instrument, especially the new 318 Model T (lines 83-84).

L103: Why is level 2.0 data provided within 1 month after calibration post deployment? Is there still any manual supervision?

L135: this is not completely true in all cases. For instance, at very low solar elevations, the air mass uncertainty and the contribution of aureole light within the instrument field of view can be very large and impose limitation to the application of the Beer law and the Kasten formula. This is actually discussed in line 335. Where can these effects be identified in the uncertainty equation 2?

L141, eq 3: is this the actual way of computing AOD, or do you use independent air mass for each component in the Beer law (eq. 1)? There is a contradiction with line 199, where the use of specific ozone air mass is indicated.

L160: this is a vague assessment of the pressure value used for Rayleigh correction. Do you have some reference where actual pressure and NCEP pressure are compared for different locations, seasons, elevations, etc.? Or maybe the effect is not so critical?

L191: there are more recent comparisons of AERONET and GPS-based water vapor retrievals.

L202: please provide quantification of the uncertainty: what spectral channels, what AOD uncertainty.

L226: please provide uncertainty of the calibration factors for the reference instruments and the field instruments.

L249-252: the discussion about various Cimel models is difficult to follow for a non-specialized reader. I would suggest adding some reference or providing the information

in a more general way.

L353. Isn't this test redundant with the usual temporal filter in the cloud-screening algorithm?

L362: check nr. 2 is missing?

L395: AOD1020>0.0: is this correct?

L415: Holben is misspelled.

L467: AERONET database comprises much more than AOD. Maybe saying "AERONET AOD database" is more precise.

L557: almucantars use fix set of azimuth angles, not scattering angles, therefore catching the halo or sun dogs is rather difficult. Why not using scattering angles instead? Maybe this is possible in the new Model T.

L566: what is hybrid scan?

L583: why may some angles not be available? Do you use right-left average value in almucantars or some right-left symmetry threshold to accept the angles?

L600: unnecessary "(" ?

L610: did you check how the fitting could be affected by incorrect pointing? Could cloud inhomogeneity yield to incorrect Sun tracking and incorrect aureole slope or curvature evaluation?

L759: do you remove all data affected by clock shifts or only data at large air masses? Have you quantified this effect for slow changing air mass (e.g. around noon or at high latitudes)? Maybe it's possible to retain some data within the prescribed AERONET AOD uncertainty of 0.01-0.02.

L1030: multi-day?

L1064: change "arctic" to "polar"

L1254: GAW-PFR network could be listed here

For the future, do you plan to apply any similar method for quality control of sky radiances and inversion products in V3?

---

## Author Comment (AC2) · 8 Dec 2018

Referee #2: This is a seminal work providing in depth analysis of the new algorithms used by AERONET/CIMEL aerosol network.

*Author Response: The Authors thank you for spending time in reviewing the manuscript and providing constructive comments to further improve the manuscript.*

Referee #2: Here are some comments that hopefully improve the already high standards article.

Referee #2: L68 PHOTONS (PHOtométrie pour le Traitement Opérationnel de Normalisation Satellitaire) and (RIMA) Red Ibérica de medida Fotométrica de Aerosoles.

*Author Response: Referee #1 had a similar comment and the changes are listed below.*

*Author Changes (in bold):*

> *Standardization of Sun photometer instrumentation, calibration, and freely available data dissemination of AOD and related aerosol databases highlights the success of the federated AERONET. For more than 25 years, the AERONET federation has expanded due to the investments and efforts of NASA (**Goddard Space Flight Center**, GSFC) (**Holben et al. 1998**), University of Lille (**PHOtométrie pour le Traitement Opérationnel de Normalisation Satellitaire (PHOTONS)**) (**Goloub et al., 2007**), University of Valladolid (**Red Ibérica de medida Fotométrica de Aerosoles (RIMA)**) (**Toledano et al., 2011**), other subnetworks (**e.g., AEROCAN (Bokoye et al., 2001**), AeroSpan (**Mitchell et al., 2017**), AeroSibnet (**Sakerin et al., 2005**), CARSNET (**Che et al., 2015**))**, and collaborators at agencies, institutes, universities, and individual scientists worldwide.*

Referee #2: L134-135 This is true only if V(lamda) error is independent of air mass. E.g. not true if theoretically signal non linearity errors are involved. Have the instruments been tested for non linearity issues?

*Author Response: The statement has been clarified for generalization as this applies to Equation 2.*

> *Author Changes (in bold):*
>
> *For the Cimel Sun photometer, the voltage signal is expressed as integer digital counts or digital number (DN). The error in the $\tau(\lambda)_{Total}$ is **generally** dependent on the optical air mass (m) by $\delta\tau$ proportional to $m^{-1}$ and hence the AOD computation error will **tend** be maximum at m=1 (Hamonou et al., 1999). **Cimel***

*instrument repeatability is tested during calibration procedures by comparing voltage ratios between the field instrument and reference instrument to be less than ±1% (Holben et al., 1998).*

Referee #2: L160 what is the impact in AOD due to the pressure uncertainty?

*Author Response: The AOD uncertainty due to pressure uncertainty (±2hPa) is expected to be less than 0.002 for the UV and insignificant for visible and near-infrared..*

Referee #2: A table summarizing the uncertainty budget that is related to all factors included in section 2 for V2 and V3 pointing out in which aspects the V2 uncertainty has been improved based on the new V3 QA/QC procedures presented here, would give more value to the whole manuscript.

*Author Response: Similar answer to Referee #1's question. Eck et al. 1999 stated estimated one sigma uncertainty to be for reference and field instruments. The AOD uncertainty estimates did not change at any significant digits spectrally for Version 3 since it is dominated by calibration uncertainty. The total estimated AOD uncertainty does not change since the Rayleigh optical depth uncertainty for Version 2 and 3 is now considered insignificant with use of station pressure and the inclusion of the NO2 optical depth uncertainty from 340nm to 500nm.*

Referee #2: L59-63 It would be informative to mention other less dense existing surface based networks e.g. SKYNET, GAW-PFR.

*Author Response: Thank you. We have included additional citations as suggested.*

*Author Changes (in bold):*

*Ground-based Sun photometry, a passive remote sensing technique, is robust in measuring collimated direct sunlight routinely during the daytime in mainly cloud-free conditions (Shaw 1983; Holben et al., 1998;* **Takamura and Nakajima 2004, Smirnov et al., 2009; Kazadzis et al., 2018**). *While these surface-based measurements are only point measurements, the federated AERONET provides measurements of columnar AOD and aerosol characteristics over an expansive and diverse geographic area of the Earth's surface at high temporal resolution.*

Referee #2: L173-174 what happens with these (A and B) calculations due to the change of the 935nm filter over time?

*Author Response: The filter degradation is monitored for all channels during the calibration process. If a filter is degraded, then it is replaced. As a result, the temporal variation in the change of the 935nm filter is considered insignificant.*

Referee #2: L199 & 205 can O3 and NO2 climatology be found somewhere on the AERONET web page?

*Author Response: The optical depth associated with the O3 and NO2 for specific wavelengths can be downloaded using the AERONET download tool product option "Aerosol Optical Depth" or "Total Optical Depth" at https://aeronet.gsfc.nasa.gov/cgi-bin/webtool_aod_v3. The file also includes the "climatological" values used for O3 and NO2 in Dobson units. The "Total Optical Depth" file will also contain the spectral optical depths of all components.*

Referee #2: L206-214 Interesting results on this aspect can be found in Arola and Koskela, 2004

*Author Response: Thank you for bringing the attention to this study. We have included in the text as shown below.*

> *Author Changes (in bold):*
>
> *Tropospheric NO2 is highly variable spatially due to various source emissions and stratospheric NO2 concentrations are more stable spatially than the tropospheric NO2 **and can bias the calculation of AOD if neglected (Arola and Koskela 2004;** Boersma et al., 2004). Therefore, regions with high tropospheric NO2 emission will tend to have greater proclivity for deviating from climatological means. Further, NO2 can vary significantly on the diurnal scale (Boersma et al., 2008).*

**Arola, A. and Koskela T.: On the sources of bias in aerosol optical depth retrieval in the UV range, J. Geophys. Res., 109, D08209, https://doi.org/10.1029/2003JD004375, 2004.**

Referee #2: L245-253 it would be interesting to report a summary of the results found for the temperature dependence on AOD based on the characterization of almost all instruments, as mentioned. Is this temperature dependence more or less similar for all instruments/ Are all instruments characterized for temperature dependence?

*Author Response: The temperature coefficient is applied to the digital number before the calculation of the AOD. All calibrated instruments have been temperature characterized. However, the temperature response depends on the sensor head optics including the detector and filters. Some instruments have had some or all filters replaced prior to the temperature characterization activity started in 2007. As a result, those instruments may not have the full characterization for historical data and rely on defaults computed based on the filter manufacturer. Figure 18 shows the average difference in AOD between Version 2 and Version 3 as a function of temperature for all instruments. Clarification in regards to the temperature correction (that is, digital number per degree C rather than AOD per degree C) has been added to the text.*

*Author Changes (in bold):*

*Beginning in 2010, the temperature sensitivity was characterized for almost all wavelengths uniquely for each Cimel instrument. The temperature effect on signal (**i.e., digital number per °C**) is a function of the combined sensitivity of the detector and the filter material itself.*

Referee #2: L310 it has to be explained why this formula is used. Is it purely empirical?

*Author Response: The amplifier do not affect the mid−visible wavelengths; however, the amplifier in can cause very large values in the UV. As a result, a conservative threshold of two times the sum of the mid−visible wavelength AOD and the sum of the UV wavelengths is empirically determined to be adequate to identify amplifier issues. The sums are used to prevent removal of aerosol data with large spectral dependence.*

Referee #2: L409 I think (whichever is greater) is not correct and is not needed. It is sufficient: less than 3 or less than 10%.

*Author Response: If the total number of measurements for the day is 10, we cannot have one measurement remaining based on less than three condition, but this measurement would stay if we keep only less than 10% condition. Alternatively, if we have 100 measurements, we will allow 10 points to remain but not nine. Therefore, "whichever is greater" is needed in this case. We have made the change below to clarify "less than 10%" below.*

> *Author Changes (in bold):    If the number of remaining measurements after all screening steps in Sect. 3.2 are performed is less than three measurements or **less than** 10% of the potential measurements (whichever is greater), then the algorithm will remove the remaining measurements.*

Referee #2: Cloud screening algorithms: A common issue is the AOD rejection from the cloud algorithms for dust related high and with high temporal variability, AOD cases. As accurate and also complete dust aerosol series is extremely important for aerosol radiative forcing global studies; are there any aspects of this new algorithm that are applied in such cases?

*Author Response: As you have noted, cloud screening procedures can inadvertently identify dust as clouds. The revised criteria for the AOD triplet variability (AOD Max minus AOD Min) exceeds 0.01 or 0.015*AOD (whichever is greater) still can potentially limit the number of dust measurements passing the algorithm if strong aerosol inhomogeneity occurs within the one minute triplet. Also, the cloud screening algorithm utilizes a new smoothness approach based on the rate of change of 0.01 apparent AOD in one minute (please see Section 3.1). While the new smoothness approach tends to preserve more data than Version 2, the Version 3 smoothness procedure could be affected by extreme changes in AOD due to anomalous aerosol plumes (e.g., biomass burning or desert dust plumes) where a strong gradient exists.*

Referee #2: Could the authors include an example or an assessment of such cases? (such as the very nice examples they already have for other cases)

*Author Response: The AERONET site IER−Cinzana (Mali) (13.28° N, 5.93° W) is already presented in Figure 27. This site located very near the south Saharan Desert boundary and frequently experiences dust events and tropical convection cirrus outflow. Figure 27 shows the Angstrom Exponent is typically between 0.2 and 0.5 indicating predominantly large particles associated with dust aerosols. The algorithm versions can be compared for AOD and Number of Days in panels a and c. In most months, Version 3 has lower AOD and fewer number of days than Version 2 suggesting the possibility of removal of cloud contamination or extreme dust events. Checking some specific periods of high AOD days removed in Version 3 and remaining in Version 2, the Version 3 algorithm properly removed data when cloud contamination was a high likelihood even in the potential presence of dust as on the following dates: 1 June 2010, 7 August 2010, 25 October 2011, and 3 June 2012. Figures 1 and 2 below provide an example of the case on 3 June 2012.*

[Figure]

**Figure 1 IER-Cinzana affected by cirrus clouds on 3 June 2012. These data were available in Version 2 Level 2.0 but are now removed in Version 3 Level 1.5 by the quality control algorithm. Terra image was collected 10:35 UTC and the Aqua image was collected at 13:40 UTC. The data were provided by NASA Worldview via the AERONET Data Synergy tool (https://aeronet.gsfc.nasa.gov/cgi-bin/bamgomas_interactive)**

[Figure]

**Figure 2** IER-Cinzana site on 3 June 2012 has high optical depth for Version 2 Level 2.0 but these data are affected by optically thin cirrus clouds which are removed by the Version 3 Level 1.5 quality control algorithm.

Referee #2: Very high AOD events could impact the sun−photometer's AOD retrieval based on the diffuse / forward scattered light entering the field of view of the instrument resulting an underestimation of the calculated AOD. Are there any control/corrective measures for such cases?

*Author Response: Section 3.1.2 provides explanation in detail the correction procedure regarding light entering the field of view of the instrument.*

Referee #2: It is unclear to me why figure 20 cases (in XX') axis differ for the two figures. Aren't we talking about the same data only analyzed for different air masses? If not it can be clarified.

*Author Response: Thank you for your comment. Yes, the axis should be the same. The Figure 20 has been modified as shown below.*

*Author Changes:*

[Figure]

**Figure 20.** Using data qualified as Version 3 Level 2.0 aerosol optical depth (AOD) 500nm average difference comparing measurements only with the pre-field calibration applied versus instruments with both the pre-field and post-field calibrations applied from 1993–2018. The AOD average differences are provided for the optical air mass 1.0≤m<7.0 range in panel **(a)** and 1.0≤m<1.5 range in panel **(b)**. Vertical bars represent the standard deviation for each day bin. The secondary y-axis in logarithmic scale represents the number of measurements of AOD 500nm for each day bin.

Referee #2: There is a number of publications pointing out small but systematic differences in wv retrievals from AERONET and the ones from other instruments/methods (GPS, microwave radiometers) is V3 results lead to better results for such comparisons?

*Author Response: The precipitable water (PW) is currently provided with the AOD product. The uncertainty for PW is expected to be within 10%. The uncertainty in PW does not cause a significant change in the uncertainty of AOD. The extrapolation of AOD to the 935nm channel may be improved in a few circumstances due to improved temperature characterization of the instrument.*

Referee #2: From the point of view of the AERONET user. There are a lot of issues presented here that contribute to the better interpretation of case studies, climatologies related with AERONET/AOD data. For such cases (lets say for example an analysis of a high AOD affected area with frequent dust events that is unknown how many of them have been captured and how many of them have been rejected by the cloud algorithm) the AERONET user could have two options:

a.        To cite this paper speculating that part of his/her results could be related with various QA/QC definitions presented here.

b.        To actually use and/or modify accordingly such QA/QC algorithms in order to have more solid conclusions.

*Author Response: The Level 1.0 data will have all of the data that may or may not have been rejected at the Level 1.5 stage and these data are readily available from the AERONET web site: https://aeronet.gsfc.nasa.gov.*

Referee #2: So the question is: will these algorithms be available to the AERONET users?

*Author Response: The methods of the algorithm are provided in this paper. Currently, the algorithm code is currently not available for public release; however, AERONET users are encouraged to contact the authors for pertinent modifications to the algorithm to include in future versions or adapting the algorithm for their applications.*

Referee #2: Again, this is a very high quality paper and the authors have done a terrific job in terms of analysis, interpretation and presentation of sun−photometric quality control and assurance procedures for AOD and WV retrievals.

[revised manuscript text omitted]

---

## Author Comment (AC3) · 8 Dec 2018

Referee #1:It is a great effort in transparency of the quality control procedures, as well as automation and objectivity. However it can be sometimes too specific and the applicability to other AOD networks difficult. The novel approach in Section 3.2.2 is of special interest and would deserve a paper by itself. Similarly, the efforts in section 3.3 are a valuable contribution to any AOD network, although sometimes too specific approach reduces the applicability of the method. I recommend the paper is published after minor revision.

Author Response: The Authors thank you for spending time in reviewing the manuscript and providing constructive comments to further improve the manuscript. The algorithm in its entirety may not apply to other networks directly, however, specific algorithm methods could be adopted by other AOD networks. The optically thin cirrus cloud screening approach presented in Section 3.2.2 was developed along with the other elements of the Version 3 algorithm and the authors deemed it more appropriate to include this section within the encompassing manuscript. While the efforts Section 3.3 are explained in detail, without it, the Version 3 data set could not be reliably quality controlled as some of these tasks were performed in Version 2 manually.

**Referee #1:Specific comments:**

I miss some references in the introduction, for instance articles or web links to the list of contributing networks (lines 65-68)

Author Response: We have modified the manuscript as shown below and added appropriate references.

Author Changes (in bold):

Standardization of Sun photometer instrumentation, calibration, and freely available data dissemination of AOD and related aerosol databases highlights the success of the federated AERONET. For more than 25 years, the AERONET federation has expanded due to the investments and efforts of NASA (Goddard Space Flight Center, GSFC) (Holben et al. 1998), University of Lille (PHOtométrie pour le Traitement Opérationnel de Normalisation Satellitaire (PHOTONS)) (Goloub et al., 2007), University of Valladolid (Red Ibérica de medida Fotométrica de Aerosoles (RIMA)) (Toledano et al., 2011), other subnetworks (e.g., AEROCAN (Bokoye et al., 2001), AeroSpan (Mitchell et al., 2017), AeroSibnet (Sakerin et al., 2005), CARSNET (Che et al., 2015)), and collaborators at agencies, institutes, universities, and individual scientists worldwide.

Bokoye, A. I., Royer, A., O'Neill, N. T., Cliche, P., Fedosejevs, G., Teillet, P. M., and McArthur, L. J. B.: Characterization of atmospheric aerosols across Canada from a ground-based sunphotometer network: AEROCAN, Atmosphere-Ocean, 39:4, 429-456,: https://doi.org/10.1080/07055900.2001.9649687, 2001.

Che, H., Zhang, X.-Y., Xia, X., Goloub, P., Holben, B., Zhao, H., Wang, Y., Zhang, X.-C., Wang, H., Blarel, L., Damiri, B., Zhang, R., Deng, X., Ma, Y., Wang, T., Geng, F., Qi, B., Zhu, J., Yu, J., Chen, Q., and Shi, G.: Ground-based aerosol climatology of China: aerosol optical depths from the China Aerosol Remote Sensing Network (CARSNET) 2002–2013, Atmos. Chem. Phys., 15, 7619-7652, https://doi.org/10.5194/acp-15-7619-2015, 2015.

Goloub, P., Li, Z., Dubovik, O., Blarel, L., Podvin, T., Jankowiak, I., Lecoq, R., Deroo, C., Chatenet, B., Morel, J. P., Cuevas, E., and Ramos, R.: PHOTONS/AERONET sunphotometer network overview: description, activities, results, Proc. SPIE, 6936, 69360V, https://doi.org/10.1117/12.783171, 2008.

Mitchell, R. M., Forgan, B. W., and Campbell, S. K.: The Climatology of Australian Aerosol, Atmos. Chem. Phys., 17, 5131-5154, https://doi.org/10.5194/acp-17-5131-2017, 2017.

Sakerin S.M., Kabanov D.M., Panchenko M.V., Pol'kin V.V., Holben B.N., Smirnov A.V., Beresnev S.A., Gorda S.Yu., Kornienko G.I., Nikolashkin S.V., Poddubnyi V.A., Tashchilin M.A: Monitoring of atmospheric aerosol in the Asian part of Russia in 2004 within the framework of AEROSIBNET program, Atmospheric and oceanic optics, 18, 11, 871–878, 2005.

Referee #1: or some references about the Cimel instrument, especially the new 318 Model T (lines 83-84).

Author Response: Based on earlier Short Comment from Emilio Cuevas, we have added a reference to Barreto et al., 2016 that describes main functionality of the CE318 Model T instruments. The Holben et al., 1998 describes the prior instrument Models as well as the AERONET web site (https://aeronet.gsfc.nasa.gov).

Author Changes (in bold): AERONET is a network of autonomously operated Cimel Electronique Sun/sky photometers used to measure Sun collimated direct beam irradiance and directional sky radiance and provide scientific quality column integrated aerosol properties of AOD and aerosol microphysical and radiative properties (Holben et al., 1998; https://aeronet.gsfc.nasa.gov/). The development and growth of the program relies on imposing standardization of instrumentation, measurement protocols, calibration, data distribution and processing algorithms derived from the best scientific knowledge available. This instrument network design has led to a growth from two instruments in 1993 to over 600 in 2018. During that time, improvements were made to the Cimel instruments to provide weather-hardy, robust measurements in a variety of extreme conditions. While the basic optical technology has evolved progressively from analog to

digital processing over the past 25 years, the most recent Sun/sky/lunar CE318 Model T instruments provide a number of new capabilities in measurement protocols, integrity, and customizability (**Barreto et al., 2016**).

Referee #1:L103: Why is level 2.0 data provided within 1 month after calibration post deployment? Is there still any manual supervision?

Author Response: The Level 2.0 data are provided within 1 month of calibration to ensure ancillary NCEP data is received and applied correctly to the entire instrument deployment. Most of the time, these data are readily available, however, some lapse in data transfer are rare but can occur due to issues such as server or network outages. The pre-field and post-field calibration steps still require manual analysis.

Referee #1:L135: this is not completely true in all cases. For instance, at very low solar elevations, the air mass uncertainty and the contribution of aureole light within the instrument field of view can be very large and impose limitation to the application of the Beer law and the Kasten formula. This is actually discussed in line 335. Where can these effects be identified in the uncertainty equation 2?

Author Response: The term ( $\tau * \delta m$ ) would be the term representing the optical air mass uncertainty. As described in Section 3.1.2, the limitation imposed spectrally on the maximum attenuation minimizes the impact of this uncertainty at large optical air mass (or low solar elevations). In this radiometer sensitivity evaluation, AERONET processing removes data having impacts of stray light at large optical air mass and thus the term ( $\tau * \delta m$ ) remains negligible.

Referee #1:L141, eq 3: is this the actual way of computing AOD, or do you use independent air mass for each component in the Beer law (eq. 1)? There is a contradiction with line 199, where the use of specific ozone air mass is indicated.

Author Response: The description of the calculation of the individual components is not explained in detail. The Authors have updated the document as shown below in Author Changes to clarify the component calculations.

Author Changes (in bold):

The spectral aerosol optical depth (AOD;  $\tau(\lambda)$ Aerosol) should be computed from the cloud-free spectral total optical depth ( $\tau(\lambda)$ Total) and the subtraction of the contributions of Rayleigh scattering optical depth and spectrally dependent atmospheric trace gases as shown in Eq. (1).

$$\tau(\lambda)_{Aerosol} = \tau(\lambda)_{Total} - \tau(\lambda)_{Rayleigh} - \tau(\lambda)_{H_2O} - \tau(\lambda)_{O_3} - \tau(\lambda)_{NO_2} - \tau(\lambda)_{CO_2} - \tau(\lambda)_{CH_4}$$
(1)

The Rayleigh optical depth ( $\tau_{Rayleigh}$ ) is calculated based on the assumptions defined in Holben et al. (1998), optical air mass (Kasten and Young 1989), and formula by Bodhaine et al. (1999), except correcting the result based on the NCEP derived station pressure. The ozone (O3) optical depth ( $\tau_{03}$ ) is dependent on the O3 absorption coefficient (ao3) for the specific wavelength, the geographic and temporally dependent multi-year monthly climatological Total Ozone Mapping Spectrometer (TOMS) O3 concentration (Co3), and the O3 optical air mass (mo3) (Komhyr et al., 1989) using the following formulation:  $\tau_{03} = a_{03} * Co_3$ \* mo3/m. Similarly, nitrogen dioxide (NO2) optical depth ( $\tau_{NO_2}$ ) is computed using absorption coefficient (aNO2) and geographic and temporally dependent multi-year monthly climatological Ozone Monitoring Instrument (OMI) NO2 concentration (CNO2) assuming NO2 scale height is equal to aerosol:  $\tau_{NO2} = a_{NO2}$ \* CNO2. The water vapor optical depth ( $\tau_{H2O}$ ) is calculated based filter dependent (e.g., 1020nm and 1640nm) A and B coefficients (discussed further below) and precipitable water in cm (u) using the following linear formulation:  $\tau_{H2O} = A + Bu$ . The carbon dioxide (CO2) optical depth ( $\tau_{CO2}$ ) and methane ( $\tau_{CH4}$ ) use station elevation dependent formulations:  $\tau_{CO2} = 0.0087 * P/P_0$  and  $\tau_{CH4} = 0.0047 * P/P_0$ , assuming the U.S. standard atmosphere (1976) and absorption constants derived from HITRAN. Further descriptions of these calculations are provided below.

Referee #1:L160: this is a vague assessment of the pressure value used for Rayleigh correction. Do you have some reference where actual pressure and NCEP pressure are compared for different locations, seasons, elevations, etc.? Or maybe the effect is not so critical?

Author Response: The Version 1 AOD algorithm used the U.S. Standard Atmosphere (1976). However, issues were detected at Mauna Loa in Langley measurements due to the significant deviation of assumed station pressure based on elevation. The calculated station pressure from the NCEP/NCAR reanalysis geopotential height and surface pressure fields was found to be a uniform method to apply to globally distributed sites as station pressure is not available at all sites. These derived NCEP/NCAR station pressure estimation follows the same procedure as Version 2. Various AERONET sites at high elevation were evaluated to be generally within 2 hPa of the local station pressure. Cimel Model T instruments have pressure sensors connected to the control unit and report the station pressure. For example, at Mauna Loa Observatory with an elevation of (3402 meters), the in mean station pressure retrieved from NCEP (679.9±1.5 hPa) subtracted by the Model T measured station pressure (681.5±1.6 hPa, where the Model T has an uncertainty of 1hPa stated by Cimel Electronique web site (https://www.cimel.fr)) based on 80,895 measurements from 2015 to 2016 is -1.6 hPa and this result is consistent within the expected range of uncertainty for the derived NCEP pressure.

Referee #1:L191: there are more recent comparisons of AERONET and GPS-based water vapor retrievals.

Author Response: Thank you for identifying this omission. Several references to comparisons of AERONET to GPS measurements have been added.

Author Changes (in bold): The one sigma uncertainty in the calculation of PW in cm is expected to be less than 10% compared to GPS precipitable water retrievals (Halthore et al., 1997; **Bokoye et al., 2003; Sapucci** et al., 2007; Alexandrov et al., 2009; Prasad et al. 2009; Bock et al., 2013; Van Malderen et al., 2014; Pérez-Ramírez et al., 2014; Campenelli et al., 2018).

- Alexandrov, M. D., Schmid, B., Turner, D. D., Cairns, B., Oinas, V., Lacis, A. A., Gutman, S. I., Westwater, E. R., Smirnov, A., and Eilers, J.: Columnar water vapor from multifilter rotating shadowband radiometer data, J. Geophys, Res., 114, D02306, https://doi.org/10.1029/2008JD010543, 2009.
- Bock, O., Bosser, P., Bourcy, T., David, L., Goutail, F., Hoareau, C., Keckhut, P., Legain, D., Pazmino, A., Pelon, J., Pipis, K., Poujol, G., Sarkissian, A., Thom, C., Tournois, G., and Tzanos, D.: Accuracy assessment of water vapour measurements from in situ and remote sensing techniques during the DEMEVAP 2011 campaign at OHP, Atmos. Meas. Tech., 6, 2777-2802, https://doi.org/10.5194/amt-6-2777-2013, 2013.
- Bokoye, A. I., Royer, A., O'Neill, N. T., Cliché, P., McArthur, L. J. B., Teillet, P. M., Fedosejevs, G., and Thériault, J.-M.: Multisensor analysis of integrated atmospheric water vapor over Canada and Alaska, J. Geophys. Res., 108, 4480, doi: 10.1029/2002JD002721, D15, 2003.
- Campanelli, M., Mascitelli, A., Sanò, P., Diémoz, H., Estellés, V., Federico, S., Iannarelli, A. M., Fratarcangeli, F., Mazzoni, A., Realini, E., Crespi, M., Bock, O., Martínez-Lozano, J. A., and Dietrich, S.: Precipitable water vapour content from ESR/SKYNET sun–sky radiometers: validation against GNSS/GPS and AERONET over three different sites in Europe, Atmos. Meas. Tech., 11, 81-94, https://doi.org/10.5194/amt-11-81-2018, 2018.
- Prasad, A. K. and Singh R. P.: Validation of MODIS Terra, AIRS, NCEP/DOE AMIP-II Reanalysis-2, and AERONET Sun photometer derived integrated precipitable water vapor using ground-based GPS receivers over India, J. Geophys. Res., 114, D05107, doi: 10.1029/2008JD011230, 2009.
- Pérez-Ramírez, D., Whiteman, D. N., Smirnov, A., Lyamani, H., Holben, B. N., Pinker, R., Andrade, M., and Alados-Arboledas, L.: Evaluation of AERONET precipitable water vapor versus microwave radiometry, GPS, and radiosondes at ARM sites, J. Geophys. Res. Atmos., 119, 9596–9613, https://doi.org/10.1002/2014JD021730, 2014.
- Sapucci, L.F., Machado, L.A., Monico, J.F., and Plana-Fattori, A.: Intercomparison of Integrated Water Vapor Estimates from Multisensors in the Amazonian Region. J. Atmos. Oceanic Technol., 24, 1880–1894, https://doi.org/10.1175/JTECH2090.1, 2007.
- Van Malderen, R., Brenot, H., Pottiaux, E., Beirle, S., Hermans, C., De Mazière, M., Wagner, T., De Backer, H., and Bruyninx, C.: A multi-site intercomparison of integrated water vapour observations for climate change analysis, Atmos. Meas. Tech., 7, 2487-2512, https://doi.org/10.5194/amt-7-2487-2014, 2014.

Referee #1:L202: please provide quantification of the uncertainty: what spectral channels, what AOD uncertainty.

Author Response: Eck et al. 1999 stated estimated one sigma uncertainty to be for reference and field instruments. The AOD uncertainty estimates did not change at any significant digits spectrally for Version 3, since total AOD uncertainty is dominated by calibration uncertainty. The total estimated AOD uncertainty does not change since the Rayleigh optical depth uncertainty for Version 2 and 3 is now considered insignificant with use of station pressure. NO2 optical depth uncertainty is now included from 340nm to 500nm and it is generally considered  $\leq 0.003$ .

Referee #1:L226: please provide uncertainty of the calibration factors for the reference instruments and the field instruments.

Author Response: Eck et al., 1999 provides information on the AOD uncertainty. We will provide clarification as shown below.

Author Changes (in bold):

The calibration of the AOD measurements is traced to a Langley measurement performed by a reference instrument (Shaw 1983; Holben et al., 1998). The reference instruments obtain a calibration based on the Langley method morning only analyses based on typically 4 to 20 days of data performed at a mountaintop calibration sites. The primary mountaintop calibration sites in AERONET are located at Mauna Loa Observatory (19.536° N, 155.576° W, 3402 m) on the island of Hawaii and Izana Observatory (28.309° N, 16.499° W, 2401 m) on the island of Tenerife in the Canary Islands (Toledano et al., 2018). These reference instruments are routinely monitored for stability and typically recalibrated every three to eight months. Reference instruments rotate between mountaintop calibration sites and inter-calibration facilities at NASA GSFC (38.993° N, 76.839° W, 87 m) in Maryland, Carpentras (44.083° N, 5.058° E, 107 m) in France, and Valladolid (41.664° N, 4.706° W, 705 m) in Spain, where reference instruments operate simultaneously with field instruments to obtain pre-field and post-field deployment calibrations. For periods when the AOD is low ( $\tau$ 440nm<0.2), optical air mass is low (m<2), and aerosol loading is stable, the reference Cimel calibration may be transferred to field instruments (Holben et al., 1998). Eck et al. 1999 estimates the reference instrument calibration uncertainty impact on AOD varies from 0.0025 to 0.0055 with the maximum representing uncertainty only in the UV channels (340nm and 380nm). In Version 3, the field instrument AOD uncertainty is still estimated to be from 0.01 to 0.02 with the maximum representing the uncertainty only in the UV channels (340nm and 380nm).

Referee #1:L249-252: the discussion about various Cimel models is difficult to follow for a non- specialized reader. I would suggest adding some reference or providing the information in a more general way.

Author Response: We have added a citation and link to the AERONET web site.

**Author Changes:**

For Cimel Model 4 and some Model 5 instruments with two Silicon photodiode detectors, the digital counts for solar aureole and sky instrument gains are used to determine temperature coefficients for each detector (Holben et al., 1998; https://aeronet.gsfc.nasa.gov). Some Model 5 and all Model T instruments perform the direct Sun and sky measurements on the same detector (Silicon or InGaAs) and typically utilize the solar aureole gain digital counts (Barreto et al., 2016; https://aeronet.gsfc.nasa.gov).

Referee #1:L353. Isn't this test redundant with the usual temporal filter in the cloud-screening algorithm?

Author Response: The digital voltage triplet various described in Section 3.1.3 utilizes the digital number (instrument raw signal) rather than the using the computed aerosol optical depth. As stated in Section 3.4, this filter comprises of the removal of 11% of the Level 1.0 database which is mainly attributed to large cloud spatio-temporal variability. We have attempted to clarify the use of the digital number (and not AOD) below.

Author Changes (in bold):

**3.1.3 Digital Number Triplet Variance**

As mentioned in Sect. 2, the Cimel instrument performs a direct Sun triplet measurement at regular intervals throughout the day. A variance threshold is applied based on the root mean square (RMS) differences of the triplet measurements relative to the mean of these three values. If the (RMS/mean)\*100% of the **digital number** triplet is greater than 16%, then these data are not qualified as Level 1.0 AOD (Eck et al., 2014). The **digital number** temporal variance threshold is sensitive to clouds with large spatial-temporal variance in cloud optical depth and optically thick clouds such as cumulus clouds as well as issues due to poor tracking of the instrument.

Referee #1: L362: check nr. 2 is missing?

Author Response: Please see changes below which properly orders the numbered list.

Author Changes (Changes in bold):

These potentially unphysical values of TS are evaluated by a number of algorithm steps such as checks for 1) constant TS values, 2) unphysical extreme high or low TS, 3) potentially physical yet anomalously low TS with respect to the NCEP/NCAR reanalysis ambient temperatures, and 4) unphysical TS decreases (dips) or increases (spikes).

Author Response: In Level 1.0 database, AOD 1020nm can be computed as zero or below zero (even if it is considered unphysical). From Level 1.0 AOD, we allow  $AOD \ge -0.01$  (based on AOD uncertainty of 0.01) to participate in the cloud screening algorithm for Level 1.5. After cloud screening, the wavelengths with  $AOD \le -0.01$  do not advance further into the Level 1.5 algorithm.

In light of this question, it brought to our attention the omission data using this threshold as it is a legacy quality control step occurring at the end of the cloud screening module. The percentage of the Level 1.5 AOD removed by wavelength is generally less than 0.5% of the total Level 1.5 data set after all other Level 1.5 cloud screening data sets have been applied. The author changes also include the changes for this omission.

Author Changes (in bold):

**Further**, daily averaged data are evaluated for temporal stability using the AOD stability during the day at 500nm (or 440nm) and daily outlier triplets using the 3-sigma check for AOD at 500nm (or 440nm) and AE440–870nm to be within  $\pm 3$  standard deviations (Smirnov et al. 2000). Finally, each wavelength is evaluated to be greater than or equal to -0.01 (based on uncertainty of 0.01; Eck et al., 1999). At this point in the quality control algorithm, the remaining triplet measurements are not expected to have a major component of  $\tau$ cloud or  $\tau$ cirrus.

Nearly 5% of the removal of the Level 1.0 data was due to the presence of cirrus clouds as detected by the solar aureole curvature algorithm and is significant since a cirrus contamination bias is evident in the AOD in Version 2 Level 2.0 data set. The "Unqualified" category indicates data that are not triplets or lack the sufficient channels to participate in the cloud screening part of the algorithm and these measurements are rejected from Level 1.5. Finally, spectral AOD removed due to too low negative values (AOD<-0.01) has maximum removal of approximately 0.5% for 380nm and 1% for 340nm of the total Level 1.5 AOD measurements due to 0.02 uncertainty in the UV at very low optical depths, while other AOD wavelengths have generally much less than 0.5% removal. After all of the data are cloud screened, about 66% of the Level 1.0 data are passed to the second part of the Level 1.5 instrument quality control algorithm for examination of the instrument anomalies and other spurious clouds and artifacts.

Referee #1: L415: Holben is misspelled.

Author Response: Thank you. The citation has been corrected.

Author Changes (in bold):

The basic Cimel Sun photometer Sun and sky measurement protocols were specified to NASA requirements in **Holben** et al. (1992, 1998, and 2006), and have only been slightly modified since that time for improved measurement capability of the Model 5 and Model T instruments.

Referee #1: L467: AERONET database comprises much more than AOD. Maybe saying "AERONET AOD database" is more precise.

Author Response: Thank you. "AOD" has been added for emphasis as shown below.

Author Changes (in bold):

As a result, the following sections will describe the mechanisms in which these additional cloud and anomaly components are automatically eliminated or reduced as close to zero as possible to provide a quality assured AOD (taerosol) after final calibration is applied (see Sect. 4) across the global AERONET AOD database.

Referee #1: L557: almucantars use fix set of azimuth angles, not scattering angles, therefore catching the halo or sun dogs is rather difficult. Why not using scattering angles instead? Maybe this is possible in the new Model T.

Author Response: Yes, almucantars measure at azimuth angles; however, they are converted to scattering angles for data processing. The principal plane and hybrid (principal plane like and almucantar like sky scan) measure the angular distribution of sky radiances at discrete scattering angles. These measurement methods do not have enough scattering angle resolution to fully capture magnitude change of the halo or sun dog in scattering angle range between 6 and 35 degrees. To pursue this further, the instrument would need to perform high scattering angle resolution measurements in this scattering angle range by either creating a new measurement scenario or changing the hybrid scenario. However, implications in the time it takes to perform measurements need to be considered when changing instrument measurement procedures.

Referee #1: L566: what is hybrid scan?

Author Response: The Cimel Model T instrument has the capability to measure symmetric scattering angles of sky radiances in which the number of scattering angles measured is maximized for the solar zenith angle. The hybrid resembles a principal plane near solar noon but it still provides symmetry for cloud clearing capability. As the solar zenith angle increases, the hybrid measurement scan resembles an almucantar measurement. More information can be found on the AERONET web site (https://aeronet.gsfc.nasa.gov).

Referee #1: L583: why may some angles not be available? Do you use right-left average value in almucantars or some right-left symmetry threshold to accept the angles?

Author Response: Some radiance data collected at the solar aureole scattering angles can be saturated (or unavailable) sometimes for instruments prior to the Cimel Model T instrument. Each right and left scan is evaluated separately for presence of cirrus.

Referee #1: L600: unnecessary "("?

Author Response: Thank you, the parentheses has been corrected when referring to equation 11.

Author Changes (in bold): Therefore, we derive the Eq. Error! Reference source not found. to determine the slope of curvature dependent only on the slope of the linear regression fit of LA and  $\varphi$  on logarithmic scale as follows:

Referee #1: L610: did you check how the fitting could be affected by incorrect pointing? Could cloud inhomogeneity yield to incorrect Sun tracking and incorrect aureole slope or curvature evaluation?

Author Response: The correlation coefficient is utilized to reduce the effect of incorrect pointing or inhomogeneity in the solar aureole radiance measurements.

Referee #1: L759: do you remove all data affected by clock shifts or only data at large air masses? Have you quantified this effect for slow changing air mass (e.g. around noon or at high latitudes)? Maybe it's possible to retain some data within the prescribed AERONET AOD uncertainty of 0.01-0.02.

Author Response: In Version 3, the clock shifts are identified and data are removed for the entire day. Data near solar noon may have less influence of the clock shift; however, the AOD uncertainty is also a maximum at solar noon so any deviation for clock shift may draw these data into question. As a result, the entire day is removed when a clock shift is identified. For slowly varying optical air mass, the effectiveness of air mass influenced methods is more difficult to determine and likely requires a large perturbation to be detected.

Referee #1: L1030: multi-day?

Author Response: Thank you. We have clarified the point below.

Author Changes (in bold):

Once the above conditions are met, these data are considered to reach Level 2.0. These Level 2.0 data are recommended for publication and use in various atmospheric applications. The automated **quality control** algorithm attempts to preserve aerosol data while removing data artifacts.

Referee #1: L1064: change "arctic" to "polar"

Author Response: Thank you. We have made the change below.

Author Changes (in bold):

Temperature characterization has proven to be small yet necessary adjustment to the AOD computation and this improvement is especially exhibited in **polar** regions or sites with very low aerosol loading in which the Version 3 AOD spectra have much less crossover allowing for the computation of more accurate Ångstrom exponents than in the Version 2 data set.

Referee #1: L1254: GAW-PFR network could be listed here

Author Response: Thank you. We have made the change below.

Author Changes (in bold):

Other surface-based remote sensing networks such as MAN (Smirnov et al., 2009), SKYNET (Takamura and Nakajima 2004), **GAW-PFR** (Kazadzis et al., 2018), and PANDORA (Herman et al., 2009) may benefit by implementing applicable quality control methods established by AERONET.

Kazadzis, S., Kouremeti, N., Nyeki, S., Gröbner, J., and Wehrli, C.: The World Optical Depth Research and Calibration Center (WORCC) quality assurance and quality control of GAW-PFR AOD measurements, Geosci. Instrum. Method. Data Syst., 7, 39-53, https://doi.org/10.5194/gi-7-39-2018, 2018.

Referee #1: For the future, do you plan to apply any similar method for quality control of sky radiances and inversion products in V3?

Author Response: The Version 3 inversions depend on the AOD input and hence almucantar and hybrid inversions are directly impacted by changes to the AOD quality control algorithm. The Version 3 inversion quality assurance follows the Holben et al., 2006 quality controls. Holben et al. 2006: http://dx.doi.org/10.1117/12.706524.

- Advancements in the Aerosol Robotic Network (AERONET) 1
- Version 3 Database Automated Near Real-Time Quality 2
- Control Algorithm with Improved Cloud Screening for Sun 3
- Photometer Aerosol Optical Depth (AOD) Measurements 4
- David M. Giles1,2, Alexander Sinyuk1,2, Mikhail SG. Sorokin1,2, Joel S. Schafer1,2, Alexander Smirnov1,2, Ilya Slutsker1,2, Thomas F. Eck2,3, Brent N. Holben2, Jasper Lewis2,4, James Campbell5, Ellsworth J. Welton2, Sergey Korkin2,3, and Alexei Lyapustin2 5 6
- 7

[revised manuscript text omitted]